

# Antarctic clouds, supercooled liquid water and mixed-phase investigated with DARDAR : geographical and seasonal variations

Constantino Listowski[1], Julien Delanoë[1], Amélie Kirchgaessner[2], Tom Lachlan-Cope[2], and John King[2]

[1]LATMOS/IPSL, UVSQ Université Paris-Saclay, UPMC Univ. Paris 06 Sorbonne Universités , CNRS, Guyancourt, France
[2]British Antarctic Survey, National Environment Research Council, High Cross, Madingley Road, Cambridge, England, CB3 OET, UK

**Correspondence:** C. Listowski (constantino.listowski@latmos.ipsl.fr)

**Abstract.** Antarctic tropospheric clouds are investigated using the DARDAR (raDAR/liDAR)-MASK products. The cloud fraction is divided into the supercooled liquid water (SLW)-containing clouds and the all-ice clouds. The low-level SLW fraction varies according to temperature and sea ice fraction. It is the largest over water. In East Antarctica, the SLW fraction decreases sharply polewards. It is two to three times higher in West Antarctica. The all-ice cloud geographical distribution is shaped by the interaction of the main low-pressure systems surrounding the continent and the orography, with little links with sea ice fraction. We demonstrate the largest impact of sea ice on SLW (mostly mixed-phase clouds, MPC) in autumn and winter, while it is almost null in summer and intermediate in spring. Monthly variability of MPC shows a maximum fraction at the end of summer and minimum in winter. Conversely, the unglaciated (pure) SLW (USLW) fraction has a maximum at the beginning of summer. Monthly evolutions of MPC and USLW fractions do not differ on the continent. This demonstrates a seasonality in the glaciation process in marine liquid-bearing clouds. From the literature, we identify the pattern of the monthly evolution of the MPC fraction as being similar to the one of the aerosols, which is related to marine biological activity. Marine bioaerosols are known to be efficient Ice Nucleating Particles (INPs). The emission of these INPs into the atmosphere from open waters would come on top of the temperature and sea ice fraction seasonalities as factors explaining the mixed-phase clouds monthly evolution.

## 1 Introduction

Antarctic clouds need to be correctly represented in regional, and global atmospheric models, to improve daily operational forecast as well as future global climate predictions. Clouds' contributions to Antarctica's ice mass balance via precipitation, and to the Antarctic surface energy budget are poorly constrained. They exert a warming effect on the ice sheet (Scott et al., 2017; Nicolas et al., 2017). The microphysical properties of clouds can also affect circulation at much lower latitudes due to the changes they induce in the energy budget and the meridional temperature gradients (Lubin et al., 1998). In the Southern Ocean (SO), clouds cause major radiative biases in climate prediction models, down to the Antarctic seas (Flato et al., 2013; Bodas-Salcedo et al., 2014; Hyder et al., 2018). The supercooled liquid water causes major difficulties in cloud microphysics modelling over the Southern Ocean (Bodas-Salcedo et al., 2016). It is difficult to conceive that the SO energy budget long-standing dilemma will be solved without paying close attention to clouds in the Antarctic region (60° S-90° S), which is





the southern boundary of the SO and more generally the cold sink of our planet. In Antarctica, surface radiation biases of several tens of watt per square metres are derived from mesoscale high-resolution models, which point to major problems in the simulation of the cloud cover (Bromwich et al., 2013a) and of the cloud thermodynamic phase, and more particularly of the supercooled liquid water (SLW, the water staying in the liquid phase below the freezing point) (Lawson and Gettelman,

2014; King et al., 2015; Listowski and Lachlan-Cope, 2017). Ultimately, the right balance of ice versus liquid mass in antarctic clouds in high resolution models will largely depend on the way the ice microphysics is implemented and the way it leaves or remove the formed SLW (Listowski and Lachlan-Cope, 2017), and how processes like secondary ice multiplication, observed in that region (Grosvenor et al., 2012; Lachlan-Cope et al., 2016; O'Shea et al., 2017) can be correctly accounted for. This balance will in turn determine the ability of the model to minimise the surface radiative biases. Improving the modeling of

the SLW over the region may induce drastic changes in the simulations of clouds in the SO (Lawson and Gettelman, 2014), without being certain that any improvement for one part of the region will also lead to the improvement of cloud modelling over the rest of the Antarctic. Hence, being able to track the formation of SLW and the mixed-phase Antarctic-wide, appears as a necessary step to improve cloud microphysics modeling in the Antarctic, for lowering the surface radiative biases across the region that are observed in the climate prediction models (Lenaerts et al., 2017).

Because of the remoteness of the continent and the harsh environment to which every ground or aircraft operation is exposed, and due to the inaccessibility of most Antarctica to in-situ cloud science, satellite observations appear as a welcome if not crucial complement. As a matter of fact Palerme et al. (2014) used satellite radar products to build the first climatology of snowfall rates across the Antarctic continent. Nonetheless a few airborne and ground-based campaigns took place in the last ten years, presenting new cloud and precipitation studies that unveiled cloud or precipitation properties in different regions

like the Antarctic Peninsula (Grosvenor et al., 2012; Lachlan-Cope et al., 2016), the Weddell Sea (O'Shea et al., 2017), the West Antarctic Ice Sheet (Scott et al., 2017; Silber et al., 2018), Adélie Land (Grazioli et al., 2017a, b), Dronning Maud Land (Gorodetskaya et al., 2015), or South Pole (Lawson and Gettelman, 2014). In order to get the needed wider perspective on the geographical distribution and seasonal variation of the cloud thermodynamic phase and the SLW Antarctic-wide, we make use of the synergetic DARDAR (raDAR/liDAR) products (Delanoë and Hogan, 2010; Ceccaldi et al., 2013), which were recently

used for mapping the Arctic mixed-phase clouds (Mioche et al., 2015). Bromwich et al. (2012), who gave a review on all aspects of tropospheric Antarctic clouds, illustrated the strength of using active remote sensing over passive remote sensing systems to correctly capture cloud cover over icy terrains and especially over the Antarctic continent. Previous studies used other radar-lidar satellite products to describe the horizontal and vertical distribution of clouds over the Southern Ocean and Antarctica during the 2007-2009 period (Verlinden et al., 2011), the ice microphysical properties and cloud distribution over Antarctica

during the 2007-2010 period (Adhikari et al., 2012). Jolly et al. (2018) described the cloud and phase distribution over the Ross Sea and the Ross ice shelf according to a classification of dynamical regimes evidenced in previous works. However, it is the first time that the DARDAR products are used in that region. More particularly, we aim at describing the cloud geographical and seasonal variation antarctic-wide on a monthly to seasonal scale with a specific focus on the supercooled liquid water (SLW). In section 2 we recall the main features of the Antarctic atmosphere, and in section 3 we present the data and the method we

use. In section 4 we present results on the seasonal, geographical and vertical variations of the different thermodynamic phases,





as well as the monthly variations over specific regions. We also present comparisons made with ground-based measurements performed over our period of interest. Finally, we investigate links with the sea ice fraction for the different cloud phases. In section 5, we discuss the results and, importantly, discuss the link between the seasonality of mixed-phase clouds and the sea ice fraction, and provide with an explanation of the observed monthly time-series for these clouds. Section 6 concludes and

recall our main results.

## 2   The Antarctic environment

We recall the salient features of the Antarctic environment, focusing on our four years of interest (2007-2010, see section 3). The continent is characterised by a very contrasted topography illustrated in Figure 1. West Antarctica (WA) has the lowest average surface height, with a peak altitude at around 2.5 km above sea level (asl) in the interior of the West Antarctic Ice Sheet

(WAIS). East Antarctica (EA) hosts the Antarctic Plateau, which reaches altitudes of 4 km asl.

Antarctica is surrounded by an uninterrupted stream of westerlies favoured by the lack of land as illustrated by the isobars of the Mean Sea Level Pressure (MSLP) depicted in Figure 2a. We used ERA-Interim reanalysis monthly average products (Dee et al., 2011), averaged over 2007-2010 for each season. There are three permanent (climatic) low-pressure systems (King and Turner, 1997), whose average positions are essentially determined by the topography of the continent (Baines and Fraedrich,

1989). The most obvious of these lows is the Amundsen Sea low (ASL) to the west of the continent, across the Amundsen and the Ross seas around 140° W (Figure 2a-d). The two other lows are located around 100° E (see in Figures 2b and c), and around 30° E (see in Figures 2a and c). Along the coastline, an easterly circulation prevails, fueled by the above-mentioned lows and also by the regime of katabatic winds, which characterise Antarctica (see e.g. King and Turner, 1997). These downslope winds are induced by the strong cooling of the atmosphere over the high-altitude icy terrains in the interior of the continent, and their

deviation to the west while reaching the coast - due to the Coriolis force - contributes to the coastal easterly circulation. Hence, a weak anticyclonic regime prevails in the interior of the continent in East Antarctica, where air subsidence contributes to the outward surface flow of the katabatic wind regimes (James, 1989). A cyclonic circulation dominates above the surface, with a strong permanent low above the Ross Ice Shelf (RIS) area as illustrated in Figures 2e-h.

Finally, the sea ice exerts control over the moisture and heat transported into the lower atmosphere, and therefore will affect

the cloud cover and their properties, as evidenced in the Arctic (Kay and Gettelman, 2009; Taylor et al., 2015; Morrison et al., 2018), and over the Southern Ocean in winter (Wall et al., 2017), and spring and summer (Frey et al., 2018). The largest extent of sea ice occurs in September, and the smallest in February. The westernmost part of the Weddell sea shows a persistent, and dense, sea ice coverage throughout the year. Figure 2i-l show the average seasonal sea ice fraction over 2007-2010 plotted using the passive microwave sea ice concentration data record (Cavalieri and Zwally, 1996, updated yearly) archived by the

National Snow and Ice Data Center (NSIDC), and projected onto the grid used to map the cloud fraction (see section 3.2).



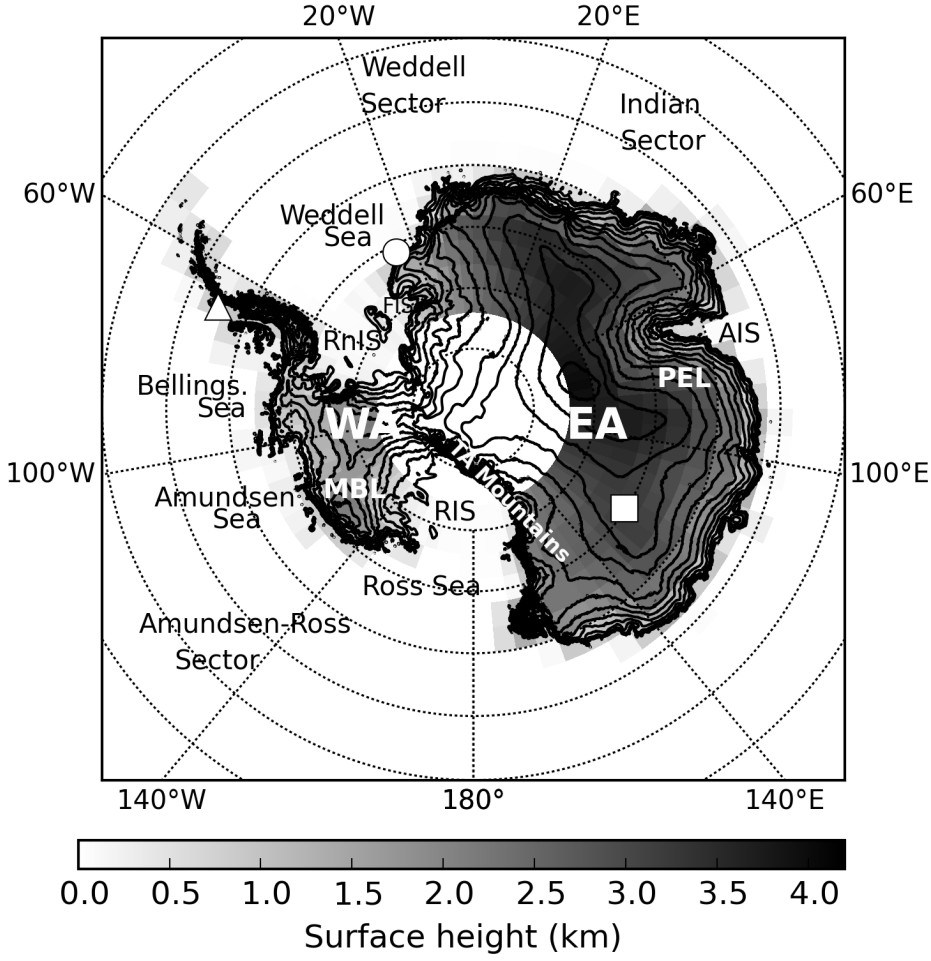

**Figure 1.** The Antarctic continent and its topography (Fretwell et al., 2013), along with names of places mentioned in this study. The Transantarctic (TA) Mountains separates West Antarctica (WA) from East Antarctica (EA). Several Ice Shelves are also reported : the Ross Ice Shelf (RIS), the Ronne Ice Shelf (RnIS) and the Amery Ice Shelf (AIS). MBL stands for Marie Byrd Land and PEL for Princess Elisabeth Land. The location of two UK and one French-Italian Antarctic stations are indicated with markers: Rothera (triangle), Halley VI (circle) and Dome C (square)





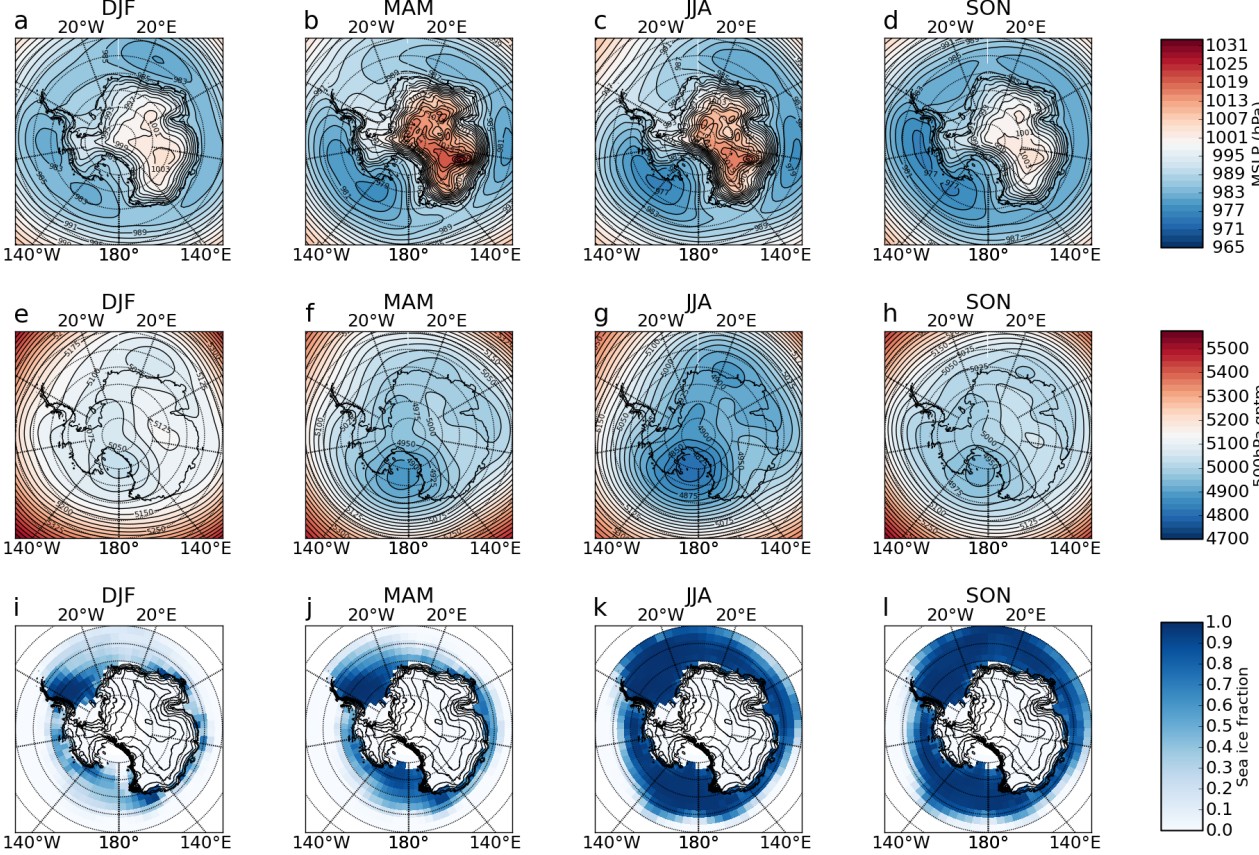

**Figure 2.** Four-year (2007-2010) seasonal averages of (a-d) the Mean Sea Level Pressure (MSLP, in hPa), (e-h) the 500 hPa geopotential height (m). (i-l) show the four-year seasonal average of the sea ice fraction obtained from the National Snow and Ice Data Center, and projected onto the grid used to map the cloud fraction (see section 3.2). The topography contours are also indicated.

## 3 Data and method

### 3.1 The DARDAR-MASK version 2 products

The DARDAR products were developed in order to use the complementarity of the CALIOP (Cloud Aerosol Lidar with Orthogonal Polarization) lidar onboard CALIPSO (Cloud Aerosol Lidar and Infrared Pathfinder Satellite Observations, Winker et al., 2010) and the Cloud Profiling Radar (CPR) onboard CloudSat (Stephens et al., 2002). Both satellites are part of the A-Train constellation (Stephens et al., 2002). A seamless retrieval algorithm uses both signals to obtain two products, namely the DARDAR-MASK (Delanoë and Hogan, 2010; Ceccaldi et al., 2013) and the DARDAR-CLOUD (Delanoë and Hogan, 2008). Due to their different wavelengths, the radar and the lidar are not sensitive to the same part of the hydrometeors size distribution. The cloud radar will be more sensitive to the large particle and will miss very small droplets or ice crystals. In contrast the lidar



is very sensitive to the concentration of hydrometeors and can detect optically thin cirrus and supercooled water but suffers from strong attenuation effect. The lidar signal is almost fully extinguished in a cloud with an optical thickness larger than 3. This synergy provides the unique opportunity to vertically describe the interior of clouds across the entire Antarctic. In this study we only make use of the DARDAR-MASK product which contains information on the three-dimensional cloud thermodynamic

phase classification at the vertical resolution of the lidar (60 m) and at the horizontal resolution of the CPR 1.7 km (along-track) x 1.4 km (cross-track). We use the most recent version 2 of those products recently made available by the AERIS/ICARE Datacenter that are introduced in Ceccaldi et al. (2013). The DARDAR-MASK v2 is built from the lidar attenuated backscatter coefficient at 532 nm (CALIPSO Level 1 products, version 4-1), the Vertical Feature Mask (VFM, CALIPSO Level 2 products, version 4-1) and the 94 GHz radar reflectivity (CloudSat 2B GEOPROF, version 4). The ECMWF-AUX (version R04) products

provide thermodynamic state variables stored in the DARDAR-MASK. They are analysis products provided by the European Centre for Medium-Range Weather Forecasts (ECMWF) that are interpolated on the CloudSat grid by the CloudSat team. The DARDAR-CLOUD products which give access to the ice microphysical properties like the ice water content, and the ice effective radius, will be investigated in a separate work (their version 2 were not yet available at the time of writing of this work).

Substantial improvements were made for the DARDAR-MASK v2 in comparison to v1 (Ceccaldi et al., 2013). The main features are a better assessment of the higher cloud cover (above 5 km height), which was overestimated in v1 due to a block effect present in the CALIOP Vertical Feature Mask (VFM) (An overcounting of cloud occurrences, due to a coarser resolution of the VFM, projected on the DARDAR 60m vertical resolution grid). As version 2 relies now directly on the original 60m-resolution lidar signal, it does not suffer from this effect. The other significant improvement is a better categorization of

supercooled water pixels, that were overestimated in v1 in the lowest atmospheric layers (Ceccaldi et al., 2013). Two examples of typical Antarctic DARDAR scenes are shown in Figure 3. The topography shows up as the brown colour in the colour-coded DARDAR-MASK transects. These two examples of transects illustrate the different categories of the mask introduced in Ceccaldi et al. (2013), along with some common features of the cloud phase and its vertical distribution observed in the Antarctic region. The summer transect (top) shows the high occurrences of supercooled liquid water with (light green colour)

or without (red colour) ice, allowing to differentiate between mixed- phase and unglaciated layers, respectively. The early spring transect (bottom) is an example of intrusion of large synoptic scale systems that can happen over the Antarctic plateau, to the east of the continent, with no or very little occurrences of a mixed-phase. As we are interested in mapping the occurrences of the liquid and the ice phase, we do not use the distinct categories developed for the ice phase but we consider all the ice categories together (namely the 'ice clouds', the 'highly concentrated ice', and the 'spherical or 2D ice"). Hence we track the

general ice phase occurrence (light blue, dark blue, purple colours of the colour-coded mask), the occurrence of supercooled liquid water (SLW) wether it is mixed (light green colour of the colour-coded mask) or not (red colour of the colour-coded mask) with ice. The vast majority of Antarctic tropospheric clouds occurs in these categories. Warm liquid clouds occurrence are observed on the margins of the domain (60° S-62° S;100° W-180° W) in very negligible amounts compared to the rest of the investigated cloud phases. Finally, note that the category "multiple scattering due to SLW" was not introduced in Ceccaldi

et al. (2013), and was subsequently added by (Ceccaldi, 2014, their section III.3.5). As explained in that work, this corresponds





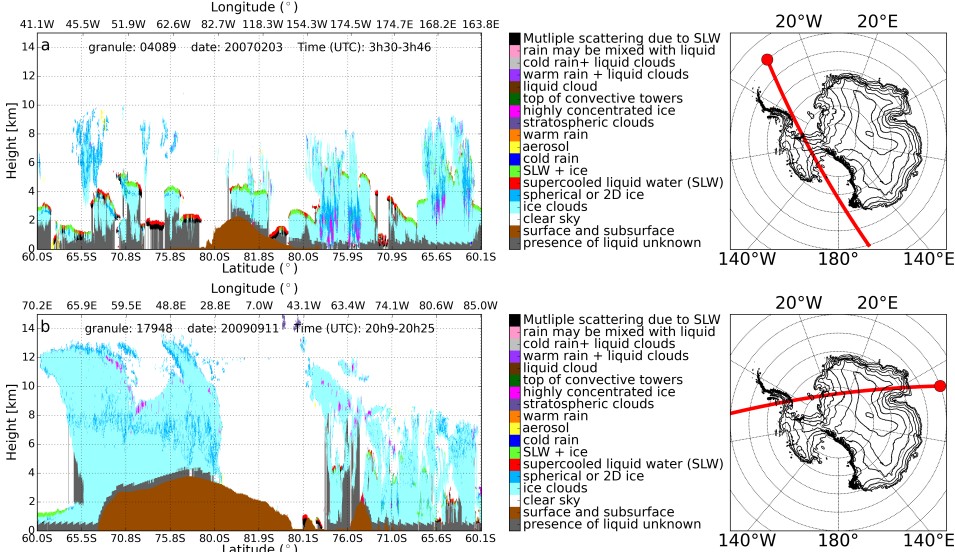

**Figure 3.** Two examples of DARDAR-MASK transects (altitude versus latitude and longitude) illustrating the various categorisation included in the DARDAR-MASK version 2. The summer transect (a) occurred on the 3rd of February 2007, while the early spring one (b) occurred on the 11th of September 2009. The small map next to each transect show the satellite track (red solid line) projected over the Antarctic region. The circle indicates the beginning of the tracks across the Antarctic.

to the detection of a backscatter signal from below the SLW layer, which is still important despite a strong attenuation of the lidar signal there. If the radar does not detect any ice, this signal has to be caused by the multiple scattering in the layer of supercooled droplets above.

## 3.2 Methodology

### 3.2.1 Statistics

About fifteen orverpasses occur each day over the Antarctic region (Fig. 4a), which we define as the region poleward of 60° S. Following Adhikari et al. (2012) and Mioche et al. (2015), we divide the area in gridboxes of 2° in latitude and 5° in longitude, which correspond approximately to a gridbox of 280 km by 220 km at 60° S, and of 280 km by 80 km at 82°S (the southernmost latitude observed by the satellites). The grid, on which the overpasses are combined to derive the occurrence frequency of the clouds, appears in the maps of the Figures 4a and c. The shaded areas in Figures 4d delimit the investigated Antarctic region in this study located between 60° S and 82° S and the three different latitudinal bands used to derived latitudinally averaged vertical transects: the Southern Ocean transect (60° S-65° S), the Coastal transect (65° S-75° S) and the Interior transect (75° S-82° S).

The sun synchronous polar orbit of the satellites results in an exponentially increased sampling of the continent as we observe closer to the pole (Figure 4b). The Southern Ocean limit at 60° S shows one overpass every two days (∼45 per season), while



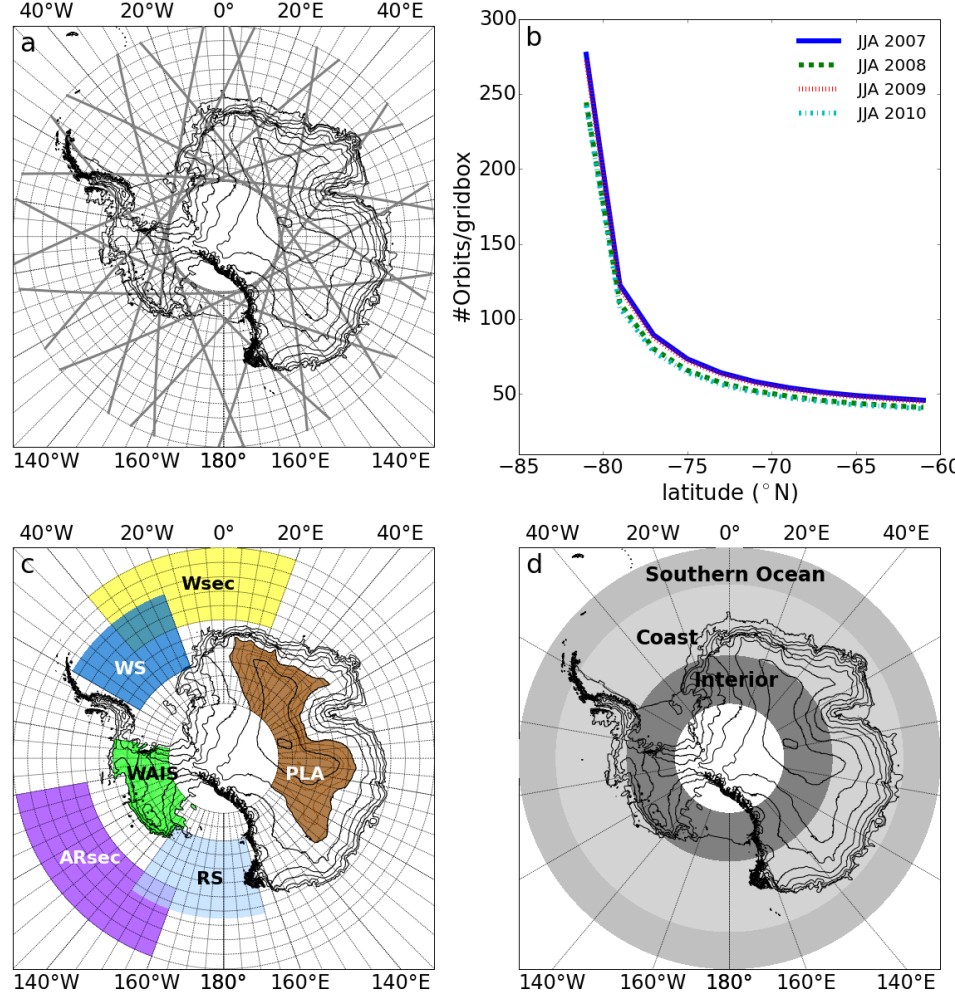

**Figure 4.** (a) The Cloudsat tracks across the Antarctic on the 1st of January 2007, and the grid (gridboxes of 2° in latitude and 5° in longitude) used to derive the geographical distribution of cloud occurrences and extending between 60° S and 82° S (b) The zonally averaged number of satellite overpasses per gridbox as a function of latitude, for the whole winter season each year. (c) Areas of interest used in the study, and introduced in section 4.3 for investigating the monthly evolution of cloud occurrences. They are called: Wsec (in the Weddell Sector), ARsec (for the Amundsen-Ross sector), WS (in the Weddell Sea), RS (in the Ross Sea), the WAIS (on the West Antarctic Ice Sheet), and PLA (part of the Plateau) (d) The three latitudinal bands used for the average vertical transects presented in section 4.2.

the southern most limit at 82° S shows on average more than 2.5 overpasses per day (∼250 per season).The sharp increase of the statistics towards the South pole is welcome as it is the area where the cloud cover is the lowest (e.g. Bromwich et al., 2012). The measurement statistics hardly change along a given parallel due to the symmetry of the polar orbiters' trajectories in relation to the south pole (hence the zonal average presented in Figure 4b). The measurement statistics are very similar from
5   one season and one year to the next. JJA is shown as an example in Figure 4b. Only DJF 2010 shows a significant reduction





(by 40 %) in the number of available DARDAR products. We use the four years 2007-2010 as they are the only four full years with nighttime observations for the CPR, which works only by daylight due to a battery failure from 2011 onwards.

Note that despite the different times of satellite overpasses over the different Antarctic areas, we do not expect any diurnal cycle to bias our observations and conclusions. For instance, the local (UTC) times of the overpasses above the gridbox

including Rothera are 2am (5am), and 5pm (8pm), while for Halley they are on average at 1.30am (1.30am) and 6pm (6pm). The morning, and evening times correspond to the descending, and ascending nodes of the satellite overpasses, respectively. Cloud cover varies diurnally as a result of the development of convection, but in and around Antarctica this will be weak at all times of year. Over the ocean, diurnal variation in surface temperature is small. Even over the ice sheets, diurnally varying convective boundary layers develop in summer at locations like Dome C, but the layer are very shallow and do not

generate convective cloud (King et al., 2006). Moreover, the ceilometer data introduced in 3.3 and used in 4.4.1 confirm the low amplitude of the cloud occurrence diurnal cycle (not shown) at Halley (2.5 % absolute variation), and Rothera (6 % absolute variation) compared to the average amplitude of the seasonal cycle (>20 %).

### 3.2.2 Cloud and phase fraction mapping

Following Mioche et al. (2015) in their study of Arctic clouds with DARDAR v1 products, we derive a (temporal absolute)

cloud fraction $F_{CLD}$ as a cloud occurrence frequency. $F_{CLD}$ is the ratio of the number of cloud occurrences $N_{CLD}$ per gridbox over the number of observations (footprints) $N_{footprints}$ in that gridbox: $F_{CLD} = N_{CLD}/N_{footprints}$. A valid cloud occurrence is an occurrence of at least three adjacent vertical pixels flagged with the same condensed phase. We do not distinguish between precipitable and non-precipitable frozen hydrometeors as the ice phase includes both cloud ice and snow in the DARDAR products. We focus on tropospheric clouds, so that stratospheric features are not accounted for in the derived horizontal or

vertical distributions of the cloud fraction. We use the tropopause height provided by the CALIOP product, and stored in the DARDAR product. The tropopause lies at ∼9 km in the summer, and at ∼12 km in the winter. The same method is used to derive the occurrence frequency $F_X$ of a given thermodynamic phase $X$. The occurrence frequency of SLW will be called the SLW fraction ($F_{SLW}$). The relative fraction (as opposed to the previously defined absolute fraction) can also be computed for the different phases, where the number of observations $N_{footprints}$ is replaced by the number of cloud occurrences $N_{CLD}$ in the

ratio. This technique is applied for every month to derive a monthly averaged fraction in every single lat-lon gridbox. To obtain the cloud (or any phase) seasonal fraction, the number of total occurrences of clouds (or any phase) over the three months of interest is divided by the total number of footprints in each gridbox over these months. In DJF (austral summer) of a given year, the December month is the one of the previous year. For instance, DJF 2007 uses December 2006. Thus, we obtain maps of the geographical distribution of the fraction of clouds or phase types. The vertical distribution of the cloud fraction is also

computed by deriving the ratios as explained above, but for each of the 60m vertical pixel.

The DARDAR-MASK includes a mixed-phase category (first type), and we extend this category by adding the clouds where a supercooled liquid water (SLW) layer is detected with at least three adjacent vertical pixels containing ice below (second type), following Mioche et al. (2015). In practice, most of the detected mixed-phase clouds (MPC) are of the first type, but SLW layers with an ice phase immediately below are clearly detected. We interpret these as occurrences of a mixed-phase



since the ice below is immediately in contact with the liquid layer; their microphysics must be interacting. Also, cloud where ice crystals are too small and/or too few to be detected by the radar in the top liquid layer of the cloud is also possible (in the upper atmosphere, for instance, the CPR cannot detect thin cirrus). SLW that is not part of any mixed-phase cloud as defined above (hence being pure liquid) is categorised as unglaciated supercooled liquid water (USLW). It is liquid where no glaciation

process has occurred (see for example Figure3a: the layer appearing in red colour around 2 km altitude at longitudes between 51° W and 82° W). We designate by "glaciation process" the processes by which a pure liquid layer becomes a mixed-phase layer. SLW will refer to any detection of supercooled liquid water (whether involved in a mixed layer or not). Adding the USLW fraction and the MPC fraction gives the SLW fraction. An all-ice category is defined and accounts for occurrences of the ice phase when no supercooled liquid at all is present in the investigated part of the troposphere (the whole of it, or the

low, mid, or high part of it). This is proven to be useful to investigate occurrences of strict ice-only processes, in order to put the behavior of these clouds in perspective with the SLW layers. The abbreviation CLD will designate any cloud occurrence. Importantly, all-ice and SLW are complementary by definition. Thus, we can summarise all the fractions we are interested in and their relationship, by writing:

$$F_{\text{CLD}} = F_{\text{all-ice}} + F_{\text{SLW}}$$
$$F_{\text{SLW}} = F_{\text{MPC}} + F_{\text{USLW}}$$

A distinction is made between low-level clouds (at altitudes between 500 m, and 3 km above the surface), mid-level clouds

(3 km-6 km above the surface), and high-level clouds (more than 6 km above surface). When no restriction to a particular altitude level is considered, we will speak about the total cloud (phase) fraction.

### 3.2.3   Limitations of the products

Finally, two main limitations have to be considered when using spaceborne lidar and radar datasets. First, the strong extinction of the lidar meeting a supercooled liquid layer prevents it from detecting any other liquid layer that could exist below this one.

The lidar signal can also get extinguished closer to the surface because of optically thick ice clouds above. Figure 3a shows grey-shaded areas flagged "presence of liquid unknown", illustrative of the extinction below the detections of supercooled liquid layers (first part of the transect in Figure 3a), respectively. This category is flagged in the mask when the lidar signal is extinguished and when at the same time the radar does not detect any ice. Second, the surface clutter or blind-zone of the radar (Tanelli et al., 2008) (surface wave reflections blurring the signal) prevents it from detecting any ice cloud (or identifying the

ice part of the MPC) close to the surface. This can be clearly seen when the identification of the ice phase ceases when nearing the ground (at ∼500 m above the surface) in Figure 3a after the longitude 168° E, and in Figure 3b right a the beginning of the transect.

Practically, the clutter height is not constant, and it is flagged in the CloudSat products used to build the DARDAR-MASK products so that the latter do not take into account the radar signal in areas where the clutter is identified. This will result

effectively in a reduced statistics close to the surface. The lidar information, however, is not filtered out. Hence, detection of





supercooled liquid layers even in the blind zone of the radar can happen. To derive the geographical distribution of the CLD fraction we consider the atmosphere above 500m above the surface, ignoring a large part of the radar ground contamination, and in order not to work with the very reduced radar or lidar statistics too close to the surface, following (Mioche et al., 2015). Their statistics was approximately halved (∼60 % loss) between 500m and 1000m above the surface. In the appendix A we

show vertical transects of the occurrence frequency of the lidar extinctions and the radar contamination (Fig. A1). Additionally, the monthly time-series of the occurrences of radar contamination and lidar extinction or attenuation above 500m above the surface are shown in Figure A2. In our dataset, ∼80 % of the radar observations are still available at an altitude of 500m above the ground (Figure A2a), and almost no contamination occurs above 1 km. Importantly, there is almost no seasonality in the radar clutter occurrences. Seasonality in the radar signal reflection can occur because of the changing nature of the surface

(caused by, e.g., more waves at the sea surface at a given time of the year Tanelli et al., 2008). The statistics of lidar observations show a ∼55 % occurrence of signal extinction at 1 km above the surface, and ∼65 % at 500m above the surface (Figure A2b). Thus, the lidar statistics are more impacted than the radar statistics in terms of signal obstruction. The lower altitude cut-off set at 500m above the surface to derive the geographical distributions does not affect our conclusions, and this is discussed in appendix B. It is mainly the SLW monthly fraction that is affected by this cut-off but not its relative variations (Figure B1).

## 3.3   Ceilometers dataset

Vaisala CT25k ceilometers were installed at Halley and Rothera in 2003, their purpose being to support logistical and scientific aircraft operations. They operate on the LIDAR principle, with a laser at 905nm as light source. The maximum measurement range of the instruments is 25000ft (∼7.5 km) with a horizontal resolution of 50ft. In this study we use datasets from Rothera and Halley over 2007-2010 (section 4.4.1). We use the operational products from the internal software of the instruments,

providing the cloud base height. We do not use the complete backscatter signal. This requires specific processing (e.g. Tricht et al., 2014) which is out of scope of the present study. The ceilometer allows different recording intervals (1 measurement every 60, 30 or 15 seconds). For most of the time these settings were kept constant for years at one level or another, but there are also changes from one month to the next, or even from day to day. Since we are looking at the ratio of cloud observation over the number of total observations, this is not an issue. Several cloud base heights are recorded if the instruments detects

more than one cloud layer. However, the number of measurements when a clear second or third cloud layer is detected is negligible and we only used the first (lowest) detected cloud base height. For instance, at Rothera 897947 individual ceilometer measurements were recorded in 2007 and a first clear cloud base was detected in 400589 cases (45 %). A second and third cloud layer with a clearly defined base height was recorded in 35530 (4 %) and 1499 (0.2 %) cases, respectively.



## 4   Results

### 4.1   Geographical and seasonal distribution of clouds and supercooled liquid water

The geographical distributions of the CLD fraction and the SLW fraction are shown as seasonal averages derived over 2007-2010 in Figure 5. It clearly shows how the SLW fraction distribution (Figure 5e-h) differs from the CLD fraction distribution
5   (Figure 5a-d). We first comment on the CLD fraction and then on the SLW fraction.

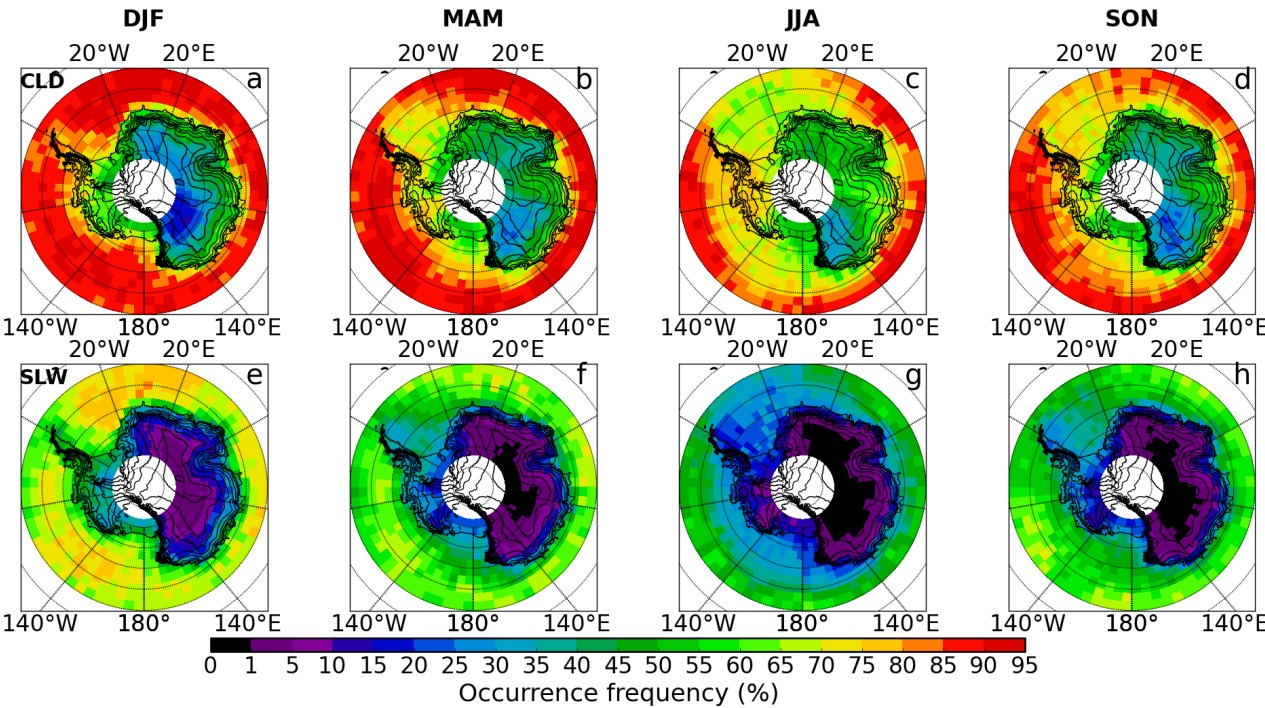

**Figure 5.** Geographical distribution of the total CLD fraction (a-d), and the total SLW fraction (e-h) for each of the four seasons based on 2007-2010 averages.

Figure 5a-d, shows that the SO and the Antarctic Seas have the largest CLD fractions as already demonstrated in previous studies using other synergetic A-Train products (Verlinden et al., 2011; Bromwich et al., 2012; Adhikari et al., 2012). There, the cloud fraction reaches values larger than 90%. From summer to winter the CLD fraction decreases the most in the Weddell Sea and the Weddell Sector, and in the Ross Sea (by an amplitude of ∼30 %). These are places where the sea ice formation extends
10   the most equatorward (Figure 2k). In the other places the winter CLD fraction remains as high as in summer. Observing the highest continental cloud fractions over the West Antarctic Ice Sheet (WAIS) is consistent with the presence of the Amundsen Sea Low (ASL) to the north of it, which brings moisture from lower latitudes to the slopes of West Antarctica's (WA) coasts. There, the orography induces adiabatic cooling and cloud formation (e.g. Scott et al., 2017). The deepening of the ASL in winter (Figure 2c) consistently leads to an increase in the CLD fraction over the WAIS (Figure 5c). A salient feature is the



minimum of the CLD fraction reached over the so-called Megadune region (75-82° S, 110-150° E) (e.g. Frezzotti, 2002), west of the Transantarctic Mountains. In fact, the minimum CLD fraction occurs around the 140° E longitude throughout the year. The largest value of this minimum occurs in winter (30-35 %). This region corresponds to the area with the largest subsidence of air on the Plateau, as emphasised by (Verlinden et al., 2011, their Figure 4). More generally, the lowest CLD fractions are

found across the high altitude terrains of the Antarctic Plateau, compared to the cloudier (by at least 20 %) WAIS. Outside of the Megadune region, the cloud fraction in East Antarctica (EA) increases from 30-35 % in summer to 60-65 % in winter.

The SLW fraction geographical distribution (Figure 5e-h) is in strong contrast to the one of the CLD fraction, especially over the continent. In EA, the SLW fraction decreases sharply polewards and away from the coast in all seasons (Figure 5e-h), following the increasing surface height. The SLW fraction is at most 10% in summer over the Plateau, decreasing to almost 0

% during other seasons, and especially in winter. WA shows, in comparisons to EA, larger continental SLW fraction in summer (30-40 % vs 10-20 %). The SLW fractions are the largest over the ocean with an average value of ∼70 % (Figure 5e). As for the CLD fraction, the strongest decrease of offshore SLW fractions in winter occurs in regions where sea ice forms. In summer, the Eastern Weddell Sea and the Weddell sector have the largest SLW fractions (75-80%). The western (60° W-40° W) Weddell Sea show systematically lower SLW fractions than the eastern (40° W-20° W) Weddell Sea, and more particularly in summer

with a 12 % absolute difference in SLW fraction.

To further emphasise the difference between cloud phase fraction distributions, we show the seasonal geographical distribution of the all-ice, MPC and USLW fraction in Figure 6. Recall that all-ice, MPC and USLW fractions describe - when added up - the entire CLD fraction. By considering the all-ice fraction (Figure 6a-d) we clearly highlight the enhancement of ice clouds over the WAIS from the summer season (20-25 %) to the winter season (∼65-70 %). This is a feature explicable by the

deepening of the ASL (Figure 2c), and the upper level low pressure system in the Ross Sea region (Figure 2g). The increase in the all-ice fraction in winter on the WAIS and in EA over Princess Elisabeth Land (PEL, ∼100° E) near the coasts goes along with the strengthening of the ASL and the other climatic low pressure system located around 100° E, respectively (Figure 2c).

The all-ice fraction distribution in winter (Figure 6c) ranges between 45 % and 70 % across the whole continent, except west of the Transantarctic Mountains where it is around 30 %. It is interesting to note that the cloud-depleted area observed

in the CLD fraction over the Megadune region is also observed in the all-ice fraction throughout the year, but not in the SLW fraction. This area is located downwind of the ASL (the upwind area being MBL, Figure 4a) and of the upper level low pressure system of the RIS. The airstream of ASL will interact with the Transantarctic Mountains, and prevent moisture or cloudiness from progressing further. Note that west, and east of the Amery Ice Shelf (AIS, Figure 4a) the all-ice fraction is larger than over it. At the same time the SLW fraction gets actually larger than in the neighboring areas of similar latitude.

This can be seen in all seasons. This is consistent with the presence of the depression in the land south of the AIS, where the absence of sharp longitudinal gradient in the orography would allow (due to the lack of adiabatic cooling) for slower or delayed cooling, and the liquid phase not to freeze. The largest all-ice fractions each season happen to be where the orography is, and southwards of places where the three climatic low pressure systems are (see section 2, and Figure 2a-d). Hence, the all-ice fraction correspond to orographic clouds and less of those clouds occur over the large Weddell and Ross Seas embayments

(e.g. Figure 5e). The Antarctic Peninsula (∼65° W) also acts as barrier to the dominant westerlies, and triggers ice cloud





**Figure 6.** Geographical distribution of the total all-ice fraction (a-d), MPC fraction (e-h), and USLW fraction (i-l) for each of the four seasons based on 2007-2010 averages.

formation through interaction between the airflow and the orography. The all-ice fraction is larger in that region than in areas over water nearby at similar latitudes. It can be already noticed that the spatial pattern of the sea ice fraction spatial distribution in winter is not detected in the all-ice cloud fraction distribution (for instance, compare the winter patterns of sea ice in Figure 2o with the winter CLD distribution in Figure 5c and the winter all-ice distribution in Figure 6c).



The MPC fraction (Figure 6e-h) and he USLW fraction (Figure 6i-l) are the largest in summer. There is an area of concentrated higher USLW fraction in the eastern Weddell Sea in summer, which has no counterpart elsewhere in the Antarctic region (e.g. in the Ross Sea). Over the SO and the seas, the absolute difference between the average MPC fraction and USLW fraction is the largest in Autumn (33 % vs 20 %, Figure 6f and j), while it is the smallest in spring (26 % vs 23 %, Figure 6h and l).

This difference is 8 % in summer, and 4 % in winter. As for the SLW fraction, the MPC and USLW fractions are lower on the continent and particularly more in EA, where they decrease polewards. They show no difference between each other, in contrast to what is observed over seas.

## 4.2   Vertical distribution of clouds and supercooled liquid water

As a complement to the geographical distribution of clouds, we now investigate their vertical distribution with a focus on the
supercooled liquid. We show the four-year average transects (at the 60 m vertical resolution) in the three latitudinal bands defined in Figure 4d, aimed at roughly describing the SO (60° S-65° S), the coastal areas (65° S-75° S), and the interior of the continent (75° S-82° S). Transects were built for the CLD fraction (Figure 7A), the SLW fraction (Figure 7B), the MPC fraction (Figure 8A), and the USLW fraction (Figure 8B). In Figure 8, isothermes built using the ECMWF temperatures stored in the DARDAR product indicate the average temperature at which MPC and USLW layers form. Similar transects for all
clouds as the ones shown in Figure 7 are discussed in Adhikari et al. (2012) for 2007-2010, and in Verlinden et al. (2011) for 2007-2009. However we show the four-years average for the CLD fraction to put the other transects into context. We limit ourselves to the low and mid altitudes as this is where all the supercooled liquid water occurs. The averaged topography in each transect is indicated by the white solid line. Since the topography is not homogeneous along any given meridian within each latitudinal band, the number of effective footprints per altitude level will change along any meridian in the coastal and
the interior transects. In order to show a smooth pattern of cloud vertical distribution, we divide the number of occurrences of any cloud type in any three dimensional gridbox by the effective number of footprints in it (it equals the number of overpasses above that gridbox if the gridbox is above the surface, and zero, if it is below the surface). In doing so, when averaging to build the transect, we account for the actual reduction of footprints along each meridian at altitude levels that are partly above and partly below the surface. Note that since the fractions are derived in each of the 60 m vertical bin, they are lower than the
ones derived for the geographical distributions, for which the occurrences of clouds were derived over the tropospheric column whatever their altitude. The reduced statistics due to the radar blind zone lidar signal extinction and across the Antarctic clearly appears in the resulting transects for the CLD fraction at ∼500 m asl (e.g., Figure 7Aa). This is illustrated and discussed in Appendix A with Figure A1. There is a sharp reduction in the low-level CLD fraction below ∼500 m asl, which corresponds to the lesser ability to detect clouds because of the radar blind zone. Satisfyingly, despite a reduction by up to 40 % of the number
of valid radar observations from 1 km to 500 m asl (Figure A1a), no discontinuity appears in the vertical transects above 500 m asl (Figure 7Aa). This suggests that the vertical distribution of CLD fraction is well reproduced above this altitude, and that it is legitimate to use 500m as a low altitude cut off for the geographical distributions introduced in the previous section.





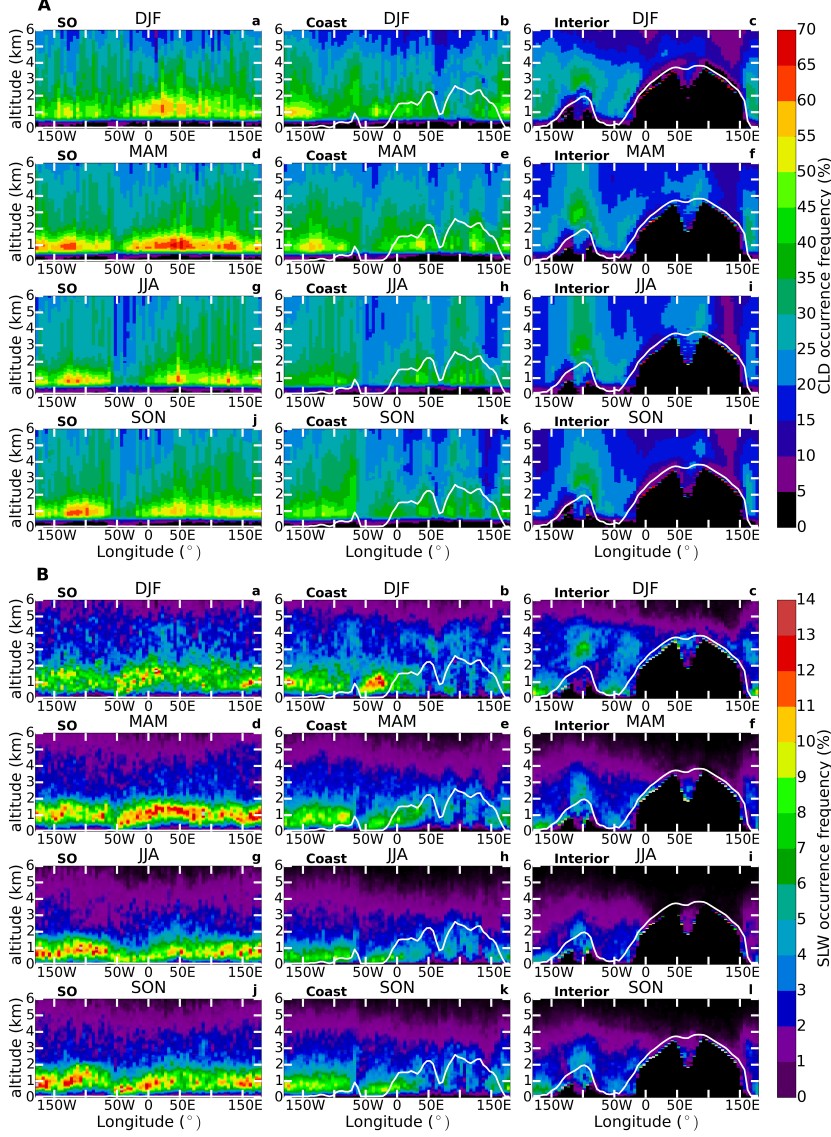

**Figure 7.** (A) Four-year (2007-2010) average seasonal vertical transects of the CLD fraction, spatially averaged over three latitudinal bands defined in Figure 4d (SO stands for Southern Ocean). One column corresponds to one latitudinal band, showing the four seasons. Each line corresponds to one season. (B) Same as (A) for the SLW fraction. The white line is the average surface elevation in the latitudinal band.

### 4.2.1 Vertical CLD fractions

The highest vertical CLD fractions (70 %) occur at low altitudes in the SO transects (Figure 7Aa, d, g and j). The maximum of the summer CLD fraction occurs across the boundary (20° E) between the Weddell Sector and the Indian Sector, and in autumn in the Indian sector and the Amundsen-Ross Sector (Figure 7A.a and d). In spring, it is the latter which has the highest





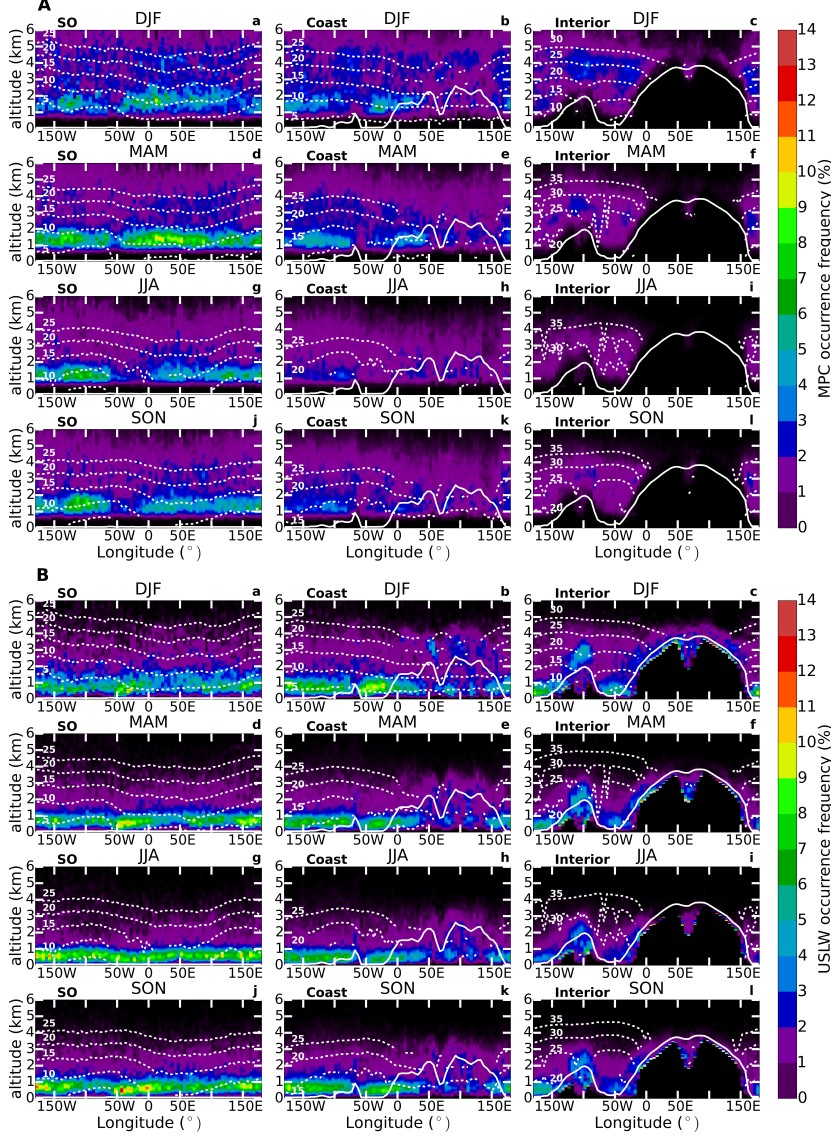

**Figure 8.** (A) Same as Figure 7A but for MPC fraction. (B) Same as (A) but for USLW fraction. Additionally, isothermes are shown every 5°C (white dotted lines). The warmest temperature shown in all subplots is -5°C. In plots Aa to Ak and Ba to Bk (SO and Coastal transects) isothermes are shown down to -25°C only, while for the interior transects they are shown down to -35°C.

occurrences of low-level clouds. To the east of the Antarctic Peninsula (AP, ∼65° W), north of and in the Weddell Sea (60° W-25° W), the CLD fraction is halved at each altitude in winter (20-30 % Figure 7Ag and h) compared to summer (40-60 % Figure 7Aa and b) in the altitude range 0.5 km-2 km asl. This reduction is less pronounced above 2 km asl. To the west of the AP, the CLD fraction is hardly changed at similar altitudes. Hence, this drop in the CLD fraction induces a dramatic difference





in the winter and spring longitudinal distribution of low-level clouds across the AP, between the west (45-55 %) and the east (20-30 %) of the mountains. Additionally and whatever the season (Figure 7b, e, h and k), there is a ∼20 % absolute difference in the mid-level CLD fractions between both sides of the AP i.e. well above the highest peak of the AP (∼2.5 km). East of the AP, the lowest low-level CLD fractions in winter and spring coincide with the largest sea ice fractions (Figure 4k and l).

In EA, east of the AIS (around 90-100° E, Figure 7Ah) a local increase in the vertical extension of coastal CLD fraction occurs in winter, at altitudes up to 6 km asl. This feature in the vertical distribution of clouds occurs while the climatic low-pressure system located off the coast is strengthened (Figure 4c). This low pressure system is the weakest in summer (Figure 4a), and the CLD fraction is also at its lowest (∼25 %) (Figure 7Ab). This seasonal variation is seen during each year taken separately. South of the AIS (∼70° E), the CLD fraction is lower than immediately to the east and to the west of the AIS
(Figure 7Ae, h and k). This is the effect of the land depression there, preventing the orographically induced cloud formation from occurring.

Generally, the CLD vertical extension follows the airmass interactions with the coastal topography. This is also clearly visible in the interior transects (Figure 7Ac, f, i and l) around 100° W, on the WAIS. There, the ASL brings moisture from lower latitudes, triggering cloud formation through adiabatic cooling on the steep coasts. In winter the vertical extension of clouds
lead to values of 45 % and 30 % at 2 km and 6 km asl, respectively, against 40 % and 10 % in summer. The CLD occurrences extend to higher altitudes in winter because of the deeper ASL and the contraction of the westerly circulations towards the coast (Figure 4c), In EA, the area of lowest CLD fraction (∼5-10 %) is visible around 140° E consistently with observations made by Verlinden et al. (2011) over 2007-2009. It is the area of largest subsidence, and also the region immediately west of the Transantarctic Mountains which prevents moisture or cloudiness from WA to enter EA. The land depression extending
poleward from the AIS is the area of maximum CLD fraction in the interior in winter (at ∼80° E Figure 7Ac, f, i).

### 4.2.2   Vertical SLW fractions

The largest SLW fractions are consistently found in the lowest (warmest) atmospheric layers, below 2 km altitude in the Southern Ocean transect (Figure 7Ba, d, g, and j) and the coastal regions (Figure 7Bb, e, h, and k). The largest SLW fractions over the largest oceanic area are found in autumn in the SO, across the Weddell and Indian sectors (0° E-70° E) at 1-1.5 km
altitude (Figure 7Bd). This corresponds to an area of preferred MPC formation compared to USLW (compare transects 8Ad and Bd). In the coastal transect the Weddell Sea is an area of enhanced SLW formation (60° W-25° W, Figure 7Bb). This maximum is clearly due to the increase of the USLW fraction there (Figure 8Bb) and not to the MPC summer fraction (Figure 8Ab). This already appeared in the USLW fraction horizontal distribution (Figure 6i). This suggests that is is an area more prone to maintain layers of supercooled liquid with no significant glaciation process. There is a clear cut in the SO zonal distribution
of the SLW fractions where the northern tip of the AP is (∼60° W) causing an asymmetry in this distribution. Lower altitudes (asl) are reached by SLW layers to the east of the AP compared to the west (particularly visible outside summer months, in Figure 7Bd, g, and j). This can be explained by the lower surface temperatures on the eastern side of the AP, which is well documented in the literature (e. g. Morris and Vaughan, 2003). This is also visible in the isothermes, whose altitudes gets lower to the east of the AP ( (Figure 8Aa, d, g and j).





Over water, the highest USLW fractions generally occur between 0.5km and 1km asl, and the largest vertical extent of the largest USLW fractions occurs in the Weddell Sea (Figure 8Bb). The maximum MPC fractions are located between 1km and 1.5km asl with no MPC detected below 500m asl. The isothermes indicate the average temperatures at which the MPC and USLW layers form. In the SO transect and in the Coastal transect, the highest MPC fractions occur between -15°C and -5°C

and more particularly between -15°C and -10°C. In the SO transect, USLW layers occur at temperatures above -5°C in summer (Figure 8Ba) and between -10°C and -5°C in other seasons (Figure 8Bd, g and j). The high USLW fractions in the Weddell Sea in summer between 0.5km and 1km asl occur at temperatures between -10°C and -5°C (Figure 8Bb)

In the interior transects, the SLW fraction is the largest above the WAIS (100° W) and above the Ross Ice Shelf (RIS, 170° E-150° W) throughout the year (Figure 7Bc, f, i, and l). In EA, on the Plateau, SLW layers occur only in summer at

temperatures down to -35°C. These fractions maximise in summer at 3km asl over the WAIS and below 500 m asl above the RIS (Figure 7Bc). For both locations these SLW occurrences are mainly in the form of USLW layers (compare Figure 8Bc and Ac). On the WAIS, the feature occurs at average temperatures between -23°C and -20°C (Figure 8Bc), and around -25°C in other seasons. It is reminiscent of quasi steady-state mountain-wave orographic clouds displaying supercooled droplets down to -33°C with no ice (Heymsfield and Miloshevich, 1993). However satellite observations do not allow to state on the lifetime of

such a feature here. Note that low-level SLW layers that are categorised as USLW layers in the radar blind zone above the RIS (below 500m asl) could actually be MPC layers. Silber et al. (2018), who investigate liquid-bearing clouds with ground-based measurements at McMurdo Station (167°E) at the edge of the RIS throughout the year 2016 did not differentiate between pure and mixed SLW layers. Thus, we cannot determine the preferred formation of USLW or MPC layers at very low altitudes there. In EA, some occurrences of SLW are evidenced (Figure 7Bc) while no SLW is detected over the Plateau (Figure 7Bi) in winter,

except from the depression of the land south of the AIS. There, intrusion of moisture from the ocean allows for enhanced SLW formation (Figure 7Ai).

Unlike for the CLD fraction, no discontinuity occurs in the SLW fraction vertical distribution close to the surface, especially over seas. This can be seen in Figure 7Bd, for example, which shows the autumnal SLW fraction at ∼50° W. This suggests that the statistics of the SLW fraction vertical distribution close to the surface is not much affected by the reduced statistics due to

the lidar extinctions (∼80 % near the surface, Figure A1d). Above lands, some spurious SLW fraction enhancements appear at the surface, though (e.g. Figure 8Ac, at ∼50° E). It is also interesting to note that the maximum in the vertical MPC fraction occurs above 1km height in the transects (Figure 8A), suggesting that the decrease of their occurrences at 1km and below, is rather real and not an artifact of the 40 % reduced statistics of the radar observations by 500m asl.The consequence of this is that the picture given by the DARDAR-MASK products of the MPC occurrences and the USLW occurrences is representative

of their actual averaged distribution down to 500 m asl, and possibly down to the surface for the SLW fraction at least (where the radar is not needed). Hence using a 500m low cut-off altitude for deriving the horizontal distributions of MPC and USLW fractions seems legitimate despite the reduced statistics.



### 4.3 Monthly time-series over specific Antarctic regions

#### 4.3.1 Total cloud and phase fractions

We now spatially average the geographical distribution of total CLD fractions presented in section 4.1, over distinct areas defined in Figure 4c. Doing so, we increase the statistics compared to a single gridbox, while we pin down the monthly

evolution in these regions. The regions investigated are called Wsec (in the Weddell Sector), ARsec (in the Amundsen-Ross sector), WS (in the Weddell Sea), RS (in the Ross Sea), PLA (the Antarctic Plateau) and the WAIS (Figure 4c). We also show the monthly time-series over the whole Antarctic region. Note that ARsec and Wsec are of similar sizes, as WS and RS. Figure 9 shows the monthly evolution of several total fractions : CLD (a), SLW (b), all-ice (c), MPC (d) and USLW (e). The shaded areas indicate the four-year maximum and minimum monthly average values, as an indication of the amplitude of interannual

variability.

A striking feature is the constant average CLD fraction throughout the year for the Antarctic as a whole, around 68 % (black lines) (Figure 9a). When considering specific regions, different patterns appear. Generally, over continental regions the maximum in CLD fractions occur in winter, and the minimum in summer, while it is the opposite over oceanic regions. The CLD fraction derived for ARsec shows the lowest amplitude of variation. It decreases from 90-92% in mid-Autumn, throughout

the winter and reaches a minimum of 78 % by the end of it. It increases again, reaching a second maximum around 90-92 % in late spring. A similar pattern appears for Wsec, with a stronger decrease throughout winter, down to 65 %. This is consistent with the larger sea ice fractions observed in that area in JJA (Figure 2o), and SON (Figure 2p) and can be related to the likely reduced moisture flux into the atmosphere. CLD fractions in WS and RS show the same pattern of a decreasing CLD fraction starting from a maximum in summer. However, the values are on average lower in winter over RS ($\sim$60 %) than over WS ($\sim$70

%). On the continent, the WAIS shows a slight increase in CLD fraction from summer (60%) to winter (75%) before decreasing abruptly from September to October. A much clearer trend emerges over the Plateau with a steady increase in cloud cover from summer to winter and a maximum in July. It is the area where the seasonal cycle has the largest amplitude of variation (as already noted by Verlinden et al., 2011, using vertical transects). The same abrupt decrease of the CLD fraction as over the WAIS is noticeable between September and October.

The monthly evolution of the SLW fraction (Figure 9b) is a general decrease of SLW from summer to winter with a minimum reached in August, before increasing again. This seasonal cycle is not biased by the one of the lidar signal extinctions, whose fractions are equal or lower than the SLW fractions and follow the same pattern (Appendix A, Figure A2). As a lidar signal extinction will happen below a SLW layer detection, this is expected. Some of the SLW layers may be detected just above the 500m lower altitude cut-off, so that the SLW occurrence is then counted in, but the extinguished area below is missed in the

statistics. Extinction or attenuation of the lidar signal can also happen because of optically thick ice clouds, and this is why the occurrences of extinctions and attenuations are as important in the summer as they are in the winter over the WAIS (Figure A2)). Overall, the seasonal cycle for SLW fraction above 500m above the surface is not biased by the lidar extinctions. The largest SLW fractions occur in ARsec and WSec (both 75 %) against 40 % and 30 %, respectively, in the winter. The lowest values of the SLW fraction are observed above the continental areas. The SLW fraction in the WAIS is 40 % in summer and 10




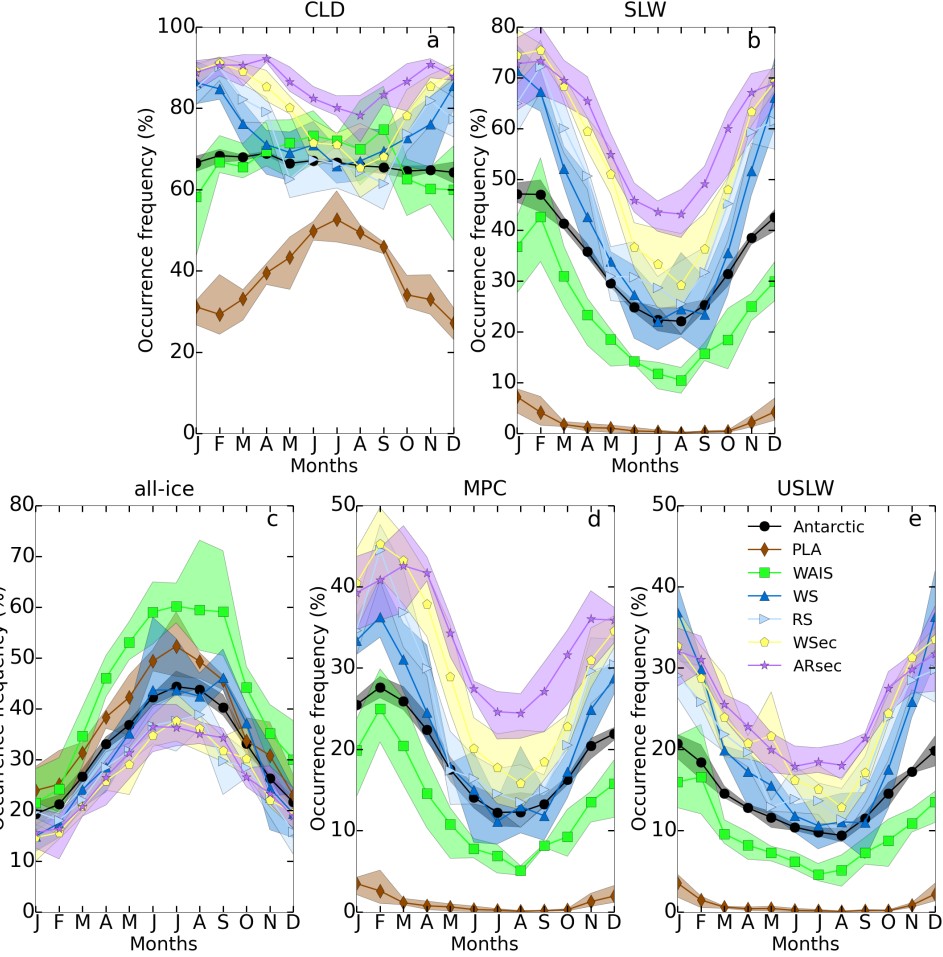

**Figure 9.** Four-year (2007-2010) average monthly time series of the total CLD fraction (a), SLW fraction (b), all-ice fraction (c), MPC fraction (d), and USLW fraction (e). See section 3.2 for the definition of categories and Figure 4c for the definition of regions. The shaded areas indicate the amplitude between the monthly minimum and maximum encountered over the years from 2007 to 2010.

% in winter. The Plateau has little but non-negligible SLW occurrences, with ∼10 % in January, and none by the end of winter and early spring. The relatively simple SLW fraction seasonal cycle points to the temperature seasonality as being one of the main driver everywhere in the Antarctic (colder temperatures favouring more glaciation in clouds).

Contrary to the CLD fraction evolution, the all-ice cloud fraction shows the same evolution over each area, increasing from
5   the end of the summer to the winter (Figure 9c). ARsec and Wsec show similar values ranging from 15 % to 35 %, while the WAIS reaches the largest four-year average of 60 %. The Plateau has the second largest values of all-ice fractions in Winter (50%). This lower fraction than the WAIS' can be explained by the ASL located off the WAIS coast, contributing to a direct inflow of moisture (and cloudiness) towards it. Note the almost identical evolution for the CLD and all-ice fractions over the





Antarctic Plateau, showing the almost exclusive presence of ice clouds there. These fractions only differ during the summer months, when SLW is also observed in non-negligible amounts (∼10 %, Figure 9b).

We differentiate between the mixed phase cloud layers (MPC, Figure 9d) and the unglaciated ones (USLW, Figure 9e). A striking difference appears when considering the transition from beginning of summer to autumn. All regions – with the exception of the Plateau – show a local maximum of MPC fraction in late summer or early autumn before a decrease in the following months, with a minimum reached around August. Conversely, the USLW fraction shows a steep decrease over the same period, which starts in January. This difference suggests that the glaciation process converting the supercooled liquid to a mixed-phase follow a distinctly different cycle from the one describing the mere occurrence of supercooled liquid (although the former is obviously related to the latter). These differing behaviours are readily observed by comparing the Antarctic averages (the black solid lines) in Figure 9d and e. The differences of MPC fractions between marine areas (ARsec, Wsec, WS and RS) are larger than the differences of USLW fraction between the same areas. For instance the USLW fractions are within a 5 % occurrence range in winter, while the MPC fractions can differ by more than 15 %. This points to larger regional differences in the glaciation process and occurrences of MPC layers than in the mere occurrences of the USLW.

### 4.3.2   Cloud and phase fraction at low, mid and high levels

To look further into the details of the monthly evolution of the cloud and phase fractions, we divide them into low-level, mid-level, and high-level fractions, as defined in section 3.2. Figure 10 shows the all-ice fractions (a-c), the SLW fractions (d-f) and the MPC fraction (g-i). Since the addition of the all-ice and the SLW fractions give the CLD fraction it is easy to infer what the dominant component of the CLD fraction is and we do not show the CLD fraction here, although we will still refer to it.

Over the continent, the monthly variability of the CLD fraction is primarily driven by the mid-level and high-level all-ice clouds (Figure 10b and c). The monthly variability is within a 30 % range and 35 % range for mid- and high-level all-ice fractions, respectively on the WAIS, and 30 % and 15 % in PLA. Regarding PLA, the mid-level clouds can virtually be considered as high clouds (in comparison to the oceanic regions) since the average altitude of the Plateau is 3 km above sea level. The evolution of the continental mid-level and high-level all-ice fractions appear to be the same in both WAIS and PLA, changing from a minimum in summer to a maximum in winter. This is consistent with the increases in cyclogenesis, and depressions off shore in that season (e.g. King and Turner, 1997), leading to more intrusions of weather systems over the continent. The monthly evolution of continental clouds is essentially driven by ice clouds, notably large frontal systems devoid of SLW, as shown in the example in Figure 3b. The mid- and high-level clouds detected over the WAIS and the Plateau are almost exclusively of the all-ice type given the much smaller mid-level SLW fractions (0 % and ≤10 % for PLA and WAIS, respectively) compared to the mid-level all-ice fractions (20-50 % and 10-40 % for PLA and WAIS, respectively) on one side, and the almost null high-level SLW fraction (except over ARsec) compared to the all-ice fractions on the other side.

Interestingly, over water (WS, RS, Wsec and ARsec regions) the mid-level CLD fraction show almost no monthly variability compared to the low-level CLD fraction and the high-level CLD fraction (not shown). Mid-level CLD fractions are always within a 10 % range of values in ARsec, Wsec, RS and WS. We can understand the absence of monthly variation for mid-level CLD fraction over marine areas, since the ∼13 % increase in mid-level all-ice fraction (Figure 10b) is almost compensated by



a similar decrease in SLW fraction (∼10 % decrease, Figure 10e). This may be explained by the mid-level liquid phase being more often converted or replaced by ice in the winter season. Over water, the low-level CLD fractions are within a ∼40 % range of values in WS, ∼30 % in RS, ∼35 % in Wsec and ∼20 % in ARsec and this variability is caused by the SLW fraction (Figure 10d). High-level CLD fractions, driven by the all-ice fraction, are within a ∼25 % range of values in WS, ∼15 % in ARsec and

5    Wsec, and ∼10 % in RS This demonstrates that the variability of the CLD fractions over water is firstly due to the low-level liquid-bearing clouds, which dominate the CLD fraction, and secondly to the high-level all-ice clouds, while mid-level clouds have little influence. Over marine regions (WS, RS, Wsec, ARsec), the monthly variability of the all-ice fraction (Figure 10a-c) is largely driven by the mid- and high-level all-ice clouds (Figure 9b and c), pointing to the increased cyclonic activity and number of frontal systems in winter (as for the general CLD fraction).

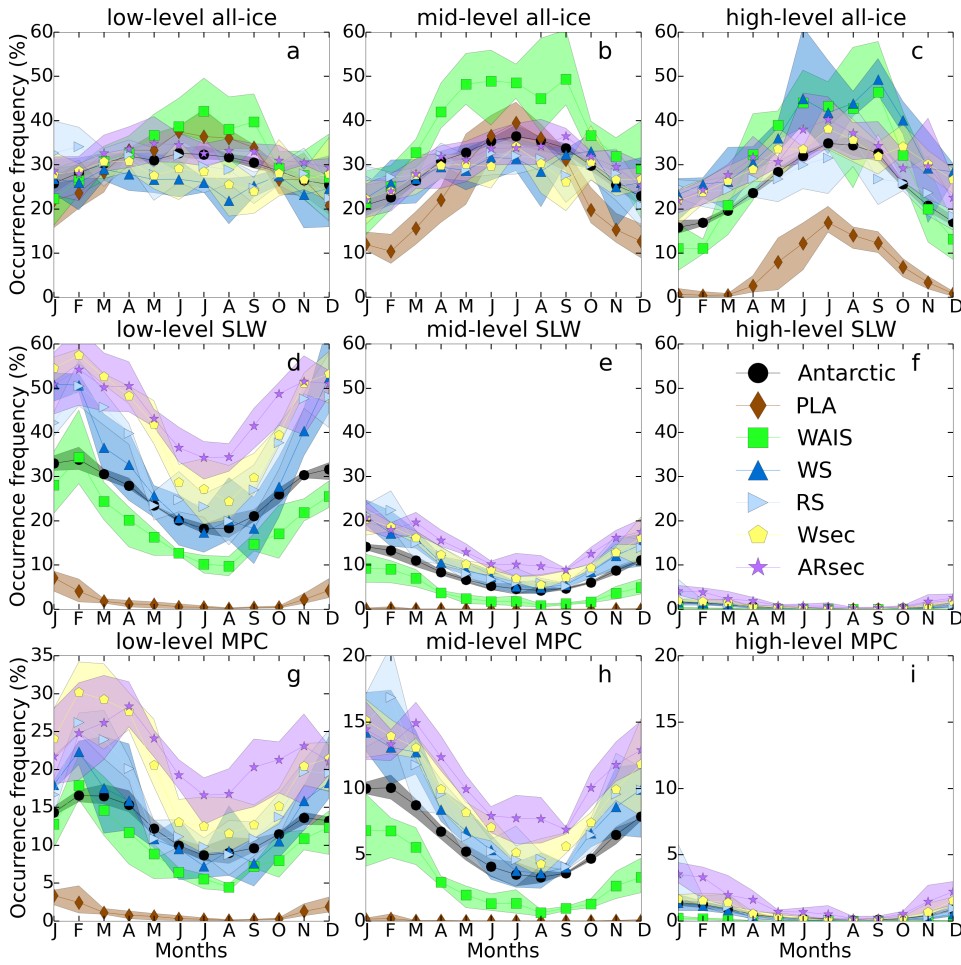

**Figure 10.** Top row: four-year (2007-2010) average monthly time series of the low (a), mid (b), and high-level (c) all-ice fraction for the different antarctic regions defined in Figure 4c. Middle row: same as top row but for the SLW fraction. Bottom row: same as top row but for the MPC fraction.





The monthly evolution of the low-level MPC fraction clearly differs from the one of the low-level all-ice fraction, but also from the low-level SLW fraction. Over marine areas, little monthly variations of the low-level all-ice fraction occur throughout the year in comparison to the low-level MPC fraction, suggesting that different factors affect their respective formation and evolution. More particularly, the monthly variation observed for the low-level all-ice fraction in WS, Wsec and ARsec is in a

range of values of 5 % (10 % for RS), while the monthly variations in low-level MPC fractions are within a larger range of values, i.e. 20 % for Wsec, 15 % for WS and RS, and 10 % for ARsec. The largest part of the USLW fraction is driven by the low-level USLW layers (not shown), which does not show a local maximum at the end of summer-beginning of autum, explaining the different patterns between the SLW (Figure 10d) and MPC (Figure 10g) seasonal cycles. Finally, Figure 10g demonstrates that the singular evolution of the MPC fraction from summer to autumn (Figure 9d) is due to the low-level MPC.

The mid-level MPC fraction does not display any similar local maximum in autumn. It rather shows a monthly variation similar to the low-level USLW layers (not shown). The particular monthly variation of the low-level MPC fractions points to a seasonal cycle of the glaciation process, involving interactions with the surface and/or the lowest layers of the troposphere. Note that, given the absence of seasonality in the radar clutter occurrences (Figure A2), the identification of ice in SLW layers to assess the existence of a mixed-phase cloud seasonality is not biased.

Figure 11 shows the monthly time series of the in-cloud temperature and water vapour mixing ratio at the top (Ttop and Qtop) of the low-level MPC and USLW layers, as well as the ones at the surface (T2m and Q2m) below where these clouds occur. The seasonal cycle of Ttop and T2m show a similar pattern to the one of the SLW fraction suggesting the temperature as being the main driver of the SLW fraction evolution. The decrease of Qtop and Q2m is a direct consequence of the formation of sea ice and the reduction in moisture coming from the sea surface. Ttop temperatures of USLW layers are larger than those

of the higher MPC layers as the latter are higher and have active glaciation processes suggestive of lower temperatures. Note that in the DARDAR-MASK, the two criteria using temperature in the identification of supercooled liquid is -40°C, taken as the homogeneous nucleation temperature, below which the lidar backscatter will be considered as coming from highly concentrated small ice crystals, and 0°C, above which the liquid layer will be considered as warm liquid (Ceccaldi et al., 2013). Apart from that it is the combination of lidar and radar observations that determines whether or not liquid and ice

are simultaneously present. Hence, our observations of a systematic higher average temperatures (and lower altitude - from section 4.2) of the USLW are, while being independent, in line with the identification of these layers. Marine supercooled liquid layers top temperatures range between -22°C and -10°C. Continental supercooled liquid layers temperatures range between -38°C and -22°C. The average lowest temperature reached by supercooled liquid water layers occur on the Plateau (-38°). The statistics based on the highest number of samples, ie those for the entire Antarctic region (black lines in Figure 11a) give a

1.5-2 °C warmer Ttop for USLW than MPC. This temperature difference is significant at the 99.9% level (using a t-test), while the differences between Qtop for MPC and USLW is significant at the 90 % level only. There are no statistically significant differences Antarctic-wide in the near-surface temperature and water vapour mixing ratio between the MPC and USLW layers (Figures 11c and d). This shows that the average near-surface conditions are the same for both types of SLW occurrences, and more particularly over water. The only exception is the winter near-surface temperature on the Plateau, which correspond to

extremely low and almost null SLW occurrences (10g and j)





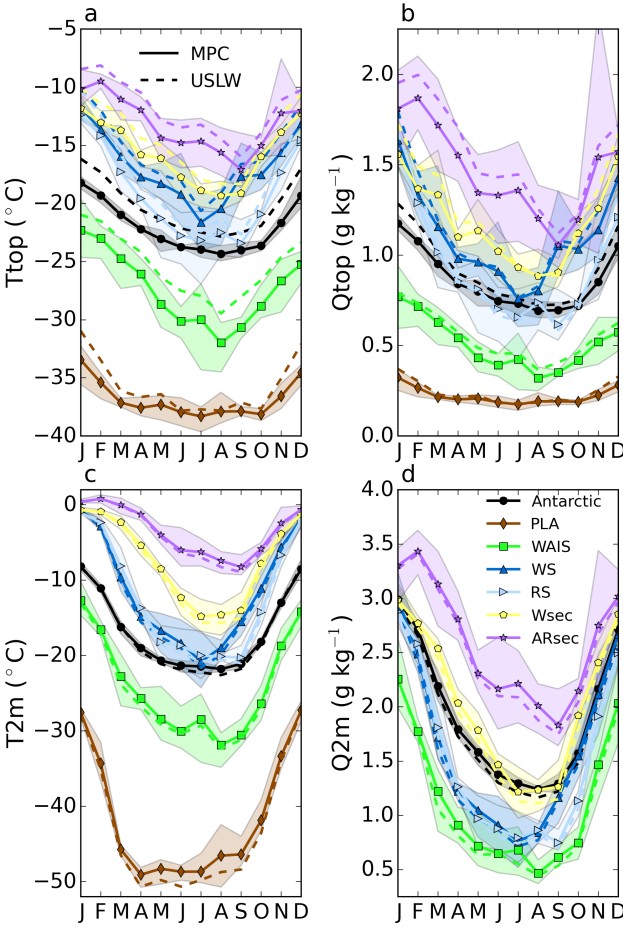

**Figure 11.** Top row: monthly time series of (a) the in-cloud Temperature (Ttop), and of (b) the in-cloud water vapour mixing ratio (Qtop), at the top of the low-level MPC and USLW layers, for the different regions of interest. Bottom row: near-surface (two-meters) temperature (c) and water vapour mass mixing ratio (d) below these layers. The shaded areas indicate the interanual variability for the MPC layers. For readability, the ones for the USLW layers are not shown, but they are of similar amplitude.

## 4.4 Comparison with ground-based measurements: from the coast to the interior

The DARDAR products were validated in the Arctic by Mioche et al. (2015) using comparisons with a ground-based Micropulse Lidar. In this section we use the geographical cloud fraction distributions derived above to make comparisons with ground-based measurements of different sorts (cloud fraction, precipitation, SLW fraction) performed over 2007-2010 in
5  Antarctica.



### 4.4.1 Ceilometer cloud base observations at Rothera and Halley between 2007 and 2010

In order to get a better perspective of the monthly evolution of cloud fraction illustrated by Figure 9a, we performed qualitative comparisons with ceilometer data collected at the British Antarctic Survey's stations Rothera and Halley for the period 2007-2010. These were introduced in section 3.3. We compare these with our low-level CLD fraction as the average height of cloud

bases detected by the ceilometers is ∼1600 m at Rothera and ∼1000 m at Halley. When using the data from the ceilometers, we plot cloud bases occurrences as measured starting from above the surface (>0 m) and from above 500 m above the surface (>500 m). In order to have enough monthly statistics from the satellite overpasses we extend the analyses to larger regions than the gridbox containing the respective station (see Figure 4a). Hence, in addition to the stations gridboxes we derive the low-level CLD fraction using - for Rothera station - the Bellignshausen sea (i.e. upwind of the station) and - for Halley - the

Weddell Sea (off the Brunt Ice Shelf where Halley sits). From Figures 5a-d, both station gridboxes experience similar seasonal cloud fractions to the one from these nearby areas. Thus it is legitimate to use those larger areas as proxies for both stations. Rothera is in the lee side of Adelaide island's mountains, though, meaning that part of the clouds observed over Rothera are orographic in nature and that local effects should be more pregnant than at Halley.

Figure 12a shows the monthly evolution of the low-level CLD fraction derived for the gridboxes corresponding to Rothera

(triangles), and Halley (circles), as well as for the Bellingshausen Sea and the Weddell Sea. Figure 12b shows the cloud fraction restrained to, and not restrained to ceilometer detections above 500m above the surface for both stations (using the same distinct markers as for Figure 12a). In each Figure the shaded area indicates the maximum, and minimum monthly average over the four years (the interannual variability). The monthly evolution of fog occurrences (reported in the ceilometers dataset as "Full obscuration but no cloud base detected"), is also reported for both stations.

The ceilometer cloud fraction (Figure 12b) are systematically lower than those derived from DARDAR products (Figure 12a). This has two potential reasons. First, ceilometers record at one single point thousands of observations per day, while the satellite has two observations per gridbox per day in the best case at these latitudes and the cloud cover over the gridbox may not be uniform. Also, the ceilometer has a much smaller footprint than the satellite and it samples the "patchiness" of the cloud on small scales. At least we found that, using only the ceilometers observations corresponding to the satellite overpasses, no

difference appears in the annual cycle (not shown). This is consistent with the fact that the diurnal cycles amplitude at both stations is negligible, and therefore does not bias our study. Nonetheless the problem of detecting much finer structures in the cloud cover with the ceilometer, that the satellite cannot resolve, spatially or temporally, is still a likely cause for mismatches. The second explanation is that fog, and blowing snow - particularly at Halley for the latter - can lead to signal obscuration and prevent the ceilometer from observing the clouds from the surface, thus lowering the number of observations. This is discussed

below.

Consistently, the cloud fraction at Halley is lower than at Rothera for both the DARDAR and the ceilometers datasets. A similar pattern appears between the ceilometers' cloud fractions at Halley (>0 m), and the DARDAR cloud fraction across the Weddell Sea: a decrease of the CLD fraction with a minimum in September, followed by a steeper increase. However, this feature is much dampened in the ceilometer's dataset when restricting detections to altitudes above 500m above the surface.



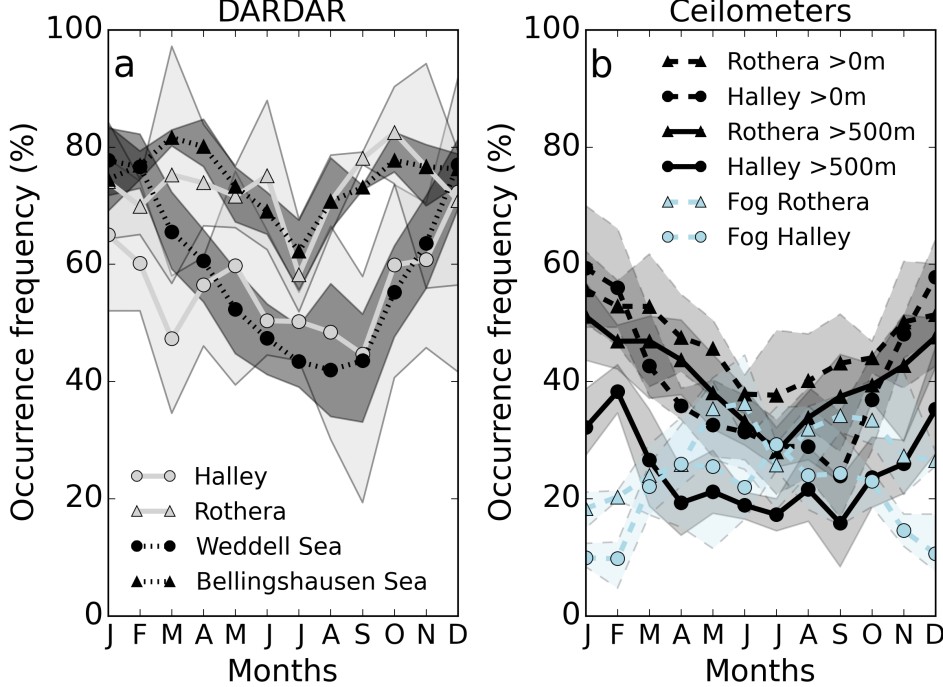

**Figure 12.** (a) Four-year (2007-2010) average time-series of the monthly mean low-level CLD fraction plotted for the gridbox corresponding to Rothera (triangle, solid line) and Halley station (circle, solid line), and the Bellingshausen Sea (triangle, dotted line) and the Weddell Sea (circle, dotted line). (b) Four-year (2007-2010) average time-series of the monthly mean of the cloud base detections by the ceilometers at Rothera (triangle) and Halley (circle), for detections above 0 m (dashed line) and 500 m (solid line) above surface. Light blue dashed lines show the fog detections for Rothera (triangle), and Halley (circle).

A similar seasonal evolution is detected at Rothera for both the ceilometers and the DARDAR products. For both datasets the minimum in cloud fractions at Rothera occurs in July. This minimum is 60 % for the DARDAR products, and 40 % (30 %) for the >0m (>500m) ceilometer detections, respectively. Also, for both datasets the maximum cloud fraction values occur in summer (75 % with DARDAR, and 50 % with the ceilometers). Note that the local maximum observed in March with the

5    DARDAR dataset is not observed with the ceilometers. Absolute differences in cloud fractions between ceilometer detections above 0 m and above 500 m at Halley (more than 20 % in summer, and down to 10 % in the winter) are larger than at Rothera (less than 10 %). This suggests a less vertically homogeneous distribution of hydrometeors at Halley.

Focusing on the detection of fog at both stations, we find 10 to 30% occurrences of fog at Halley with the maximum reached in July, and 20 to 37 % at Rothera, with a maximum at the beginning of winter (Figure 12b). Interestingly the average difference

10    between the cloud fractions from DARDAR and from the ceilometers ranges between 22 % and 43 % at Rothera, and between 32 % and 40 % at Halley. Hence, the fog occurrences can possibly explain a large part of the lower cloud fraction seen by the ceilometers, and help reconcile both data sets, particularly at Rothera. At Halley, however, blowing snow events (Mann et al.,



2000) are an additional likely source of ceilometer obscuration. It is probable that the signal of the seasonal cycle in cloud fraction seen in the DARDAR dataset is masked in the ceilometer dataset because of the seasonal cycle in fog occurrences and blowing snow events. This would explain the reduced seasonal cycle at Halley when restricting the ceilometer observations to altitudes above 500m.

Finally, one should also note, that the ceilometer detects a cloud base and we work with cloud phase fraction on a vertical grid from the DARDAR, and do not derive cloud base values here. Our low-level CLD fraction statistics could include clouds with a base below 500m, which are counted in the >0m statistics, but not in the >500m statistics. It is difficult to assess the bias induced by this difference since the cloud base of low-level clouds detected with the DARDAR products is difficult to determine because of the lidar signal extinction or the radar ground clutter. We have shown qualitative agreements between

DARDAR and ceilometer observations at both stations. However, there is a need for more systematic comparisons of ground-based measurements of cloud occurrences with combinations of lidars and cloud profiling radars in Antarctica. This will be the topic of future study using the DARDAR products for more recent years, when such ground instruments were deployed.

### 4.4.2   Precipitation measurements at Dome C in 2009 and 2010

A study of in-situ precipitation measurements over the Plateau showed that snowfall over winter at Dome C (75.1° S;123° E,

Figure 1) was about five times less important in the winter 2010 (∼1 mm water equivalent - w.e.) than in the winter 2009 (∼5 mm w.e.) (Schlosser et al., 2016, their Figure 4). Schlosser et al. (2016) related this change to the changing strength of the westerly winds belt around Antarctic coasts quantified by the Southern Annular Mode (SAM) index (Marshall, 2003): winter 2009 was a low SAM-index season, allowing more intrusions of moisture, while winter 2010 was a high SAM-index season.

We do not investigate interannual variability here, but these measurements are an opportunity to assess the consistency of

our CLD fraction variability with changes in winter precipitation measured from the ground, over 2009 and 2010. Recall that the ice phase in the DARDAR-MASK products include both, cloud ice and precipitating ice, so that increased precipitation is expected to cause an increase in our low-level CLD fraction. We subtract from the winter four-year average CLD fraction introduced in section 4.1 the 2009 and 2010 winter average, respectively. Thus, we derive an anomaly for the gridbox centered on Dome C (120°E-125° E and 74° S-76° S). Given that synoptic scale systems are the ones causing the substantial increases

in precipitations from one year to the next in these high altitude regions of the continent (Schlosser et al., 2016), it is reasonable to think that, given the absence of any topographical feature for hundreds of kilometers around Dome C station, the gridbox of size 280x100 km is representative for the location.

The CLD fraction anomalies in winter 2009 and 2010 are +5 % and 0 %, respectively. The relative increase in winter 2009 (ratio of the winter 2009 value to the winter four-year average) is 15 %. Looking at different levels, the anomaly (relative

increase) in low-level CLD fraction is +7 % (+32 %) in JJA 2009, against -4 % (-20 %) in JJA 2010; for the high-level clouds it is +5 % (+74 %) in JJA 2009, against +0.2 % (+4 %) in JJA 2010. The picture differ only for the mid-level clouds with -0.3 % (-2 %) in JJA 2009, and +4 % (+20 %) in JJA 2010. The increase in occurrences of low-level CLD fraction in JJA 2009 is consistent with the increased snow fall observed on the ground. The simultaneous increase in high-level CLD fractions illustrates the more numerous deep (thick) clouds reaching Dome C in JJA 2009. The decrease in low-level clouds in winter





2010 consistently shows the less numerous precipitating clouds, in agreement with the lower precipitation measured at the surface that year. No supercooled liquid is involved in this change of cloud fractions, as the SLW occurrences are null above Dome C during winter (Figure 5g). Overall, a 15 % relative increase in the all-ice fractions in winter 2009 (+32 % for low-level all-ice, and +75 % for high-level all-ice) is consistent with the increased snowfall measured on the ground by Schlosser et al.

(2016) during this winter.

### 4.4.3    Supercooled liquid water observations at South Pole in 2009

On the continent the DARDAR data are limited to latitudes lower than 82° S. To this respect, measurements of supercooled liquid water (mixed-phase clouds) like the ones done by Lawson and Gettelman (2014) at South Pole Station (SPS) are essential to better constrain the distribution of supercooled liquid water. During the summer 2009, they used a tethered balloon to

calibrate the mixed-phase clouds detection of a micropulsed lidar (MPL), and to subsequently deduce the number of mixed-phase clouds detection in comparison to the ice cloud detections throughout the year. From the Figure 2D of (Lawson and Gettelman, 2014), we can derive the ratio of mixed-phase cloud occurrences over the total number of cloud occurrences. The authors show the number of 10-min detections of mixed-phase clouds and all-ice clouds each month, respectively. We divide the number of mixed-phase clouds occurrences by the total number of cloud detections to build a relative fraction of mixed-

phase clouds. We attempt here a comparison with our low-level SLW relative fraction. We use our low-level fraction as the detections by the MPL are all below 3 km above the surface. The ground clutter prevents the CPR from correctly assessing the presence of ice mixed with SLW close to the surface. At the same time it is not clear in which case the strong backscatter signal of their MPL was indeed a signature of a MPC or just of a USLW layer, and the authors do not distinguish between both. Since we detect both MPC and USLW in the interior of the continent we consider both in our case. Thus, we have to use our

low-level SLW relative fraction. Additionnaly, the cloud detections by the MPL ranges between 200 m and 2200 m above the surface (Lawson and Gettelman, 2014, their Figure 2C). Only the lidar can detect the lowest layers because of the CPR blind zone.

On the continent, from the geographical distribution of the SLW fractions (Figure 5e-h) it is clear that it follows the topography as higher terrains experience lower temperatures and moisture (due to the distance to the coast), and hence lower

low-level SLW fractions. SPS is located at an altitude of 2840 m asl. It is on the slope of the ice sheet that extends northwards towards the Transantarctic mountains (Diamond-shape marker in Figure 13a and b). Relying on the idea that the SLW occurrences follow the changes in orography, the SLW fractions at South Pole should be close to the ones on the East Antarctica's side of the mountains, at similar altitudes (circle marker in Figure 13a and b). Thus, we extract values of the SLW relative fractions from the few gridboxes verifying 20° W<lon<20° E and 80°S<lat<82° S with a surface height between 2500 m asl

and 3000 m asl (called area 1; north of the circle marker in Figure 13a and b). As element of comparison we also extract the monthly time-series of the relative SLW fraction on the southernmost boundary of the WAIS (called area 2, 60° E<lon<140° E, 80°S<lat<82° S with a surface height between 2000 m asl and 2500 m asl ; north of the square marker in Figure 13a and b). Recall that our statistics is the best close to 82° S, with ∼2.5 overpasses per gridbox each day (Figure 4b).





Figure 13a and b show respectively the 4-year average summer and winter geographical distributions of the SLW relative fraction. Figure 13c shows the monthly time series of the SLW relative fraction for 2009 with (red solid line) and without (red dotted line) the 500 m lower altitude cut-off in area 1, and in area 2 without the cut-off (green dotted line). To give context, the interannual variability over 2007-2010 is also represented as shaded area for each extraction. SPS observations are shown in 5 black (diamond markers).

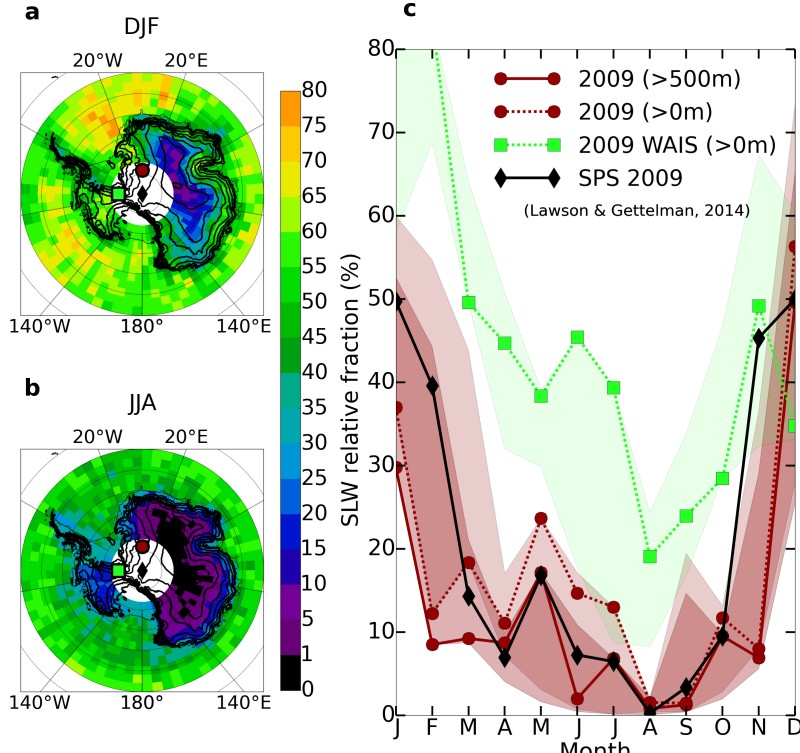

**Figure 13.** Left column: Geographical distribution of the 2007-2010 average of the SLW relative fraction in DJF (a) and JJA (b). The diamond marker indicates South Pole Station (SPS). The circle and square markers indicate the approximate locations of the gridboxes - north of the markers - used to extract the time-series in area 1 (circle) and area 2 (square), respectively (see text for exact definition of area 1 and 2). Right column: (c) Monthly time-series of the DARDAR SLW relative fraction in 2009 with (red solid line) and without (red dotted line) the lower altitude cut-off of 500m, extracted north of the circle marker shown in (a) and (b). The green dotted line shows the time-series extracted north of the square marker shown in (a) and (b). The shaded areas represent the corresponding interannual variabilities. The black diamond markers show the relative fraction of mixed-phase clouds observed with the MPL at South Pole (2850m asl) in 2009 (Lawson and Gettelman, 2014, extracted from their Figure 2D).

It is remarkable how the seasonal cycles from DARDAR observations with or without cut-off in area 1 show a similar pattern to the ground-based observations despite the different locations investigated. This is in spite of the much larger temporal resolution of the MPL, which continuously observes at a single point and can detected features missed by the satellites (as for





the ceilometers). It is also clear that area 1 is much more representative of SPS than area 2 where SLW relative fractions are by far larger than at SPS during all seasons but SON. Expectedly the SLW fractions in area 1 are larger without the altitude cut-off but only between March and July 2009. The 2009 DARDAR fraction is lower than the SPS observations by 10 % (absolute difference) and 20 % in January and February, and by 35 % in November. For the rest of the months, the difference is less

than 10 %. Importantly, though, the MPL observations lie in the interannual variability range of the DARDAR observations (with or without cut-off) throughout the year (except in November). These results suggest that cloud osbervations at South Pole are representative of lower latitudes on the Plateau and on its outskirts The contribution of SLW to cloud clearly maximise in summer with average values of 50 % in December and January for the MPL, and 40 % for DARDAR (45 % for the 2007-2010 average). The minimum of SLW relative fraction is in August for both datasets: 1 % for DARDAR (with and without the

altitude cut-off) and 0 % at SPS.

Finally, note that in 2009 the occurrences of mixed-phase clouds at South Pole were all below 5 km asl (Lawson and Gettelman, 2014). This is also the highest altitude at which we retrieve SLW above terrains as high as South Pole in summer (Figure 7Bc). The temperatures reported by (Lawson and Gettelman, 2014) in summer are between -28°C and -32°C and this is consistent with our values of -33°C to -31°C derived for January on the Plateau Figure (11a). This successful comparison,

although made between different locations, validates the ability of DARDAR to capture the SLW seasonal cycle, and more particularly in the southernmost regions probed by the satellites.

### 4.5    Sea ice, supercooled liquid water and mixed-phase clouds

The monthly evolution of the various cloud phase derived in section 4.3 showed distinct patterns. Here, we investigate the potential links between the low-level cloud and sea ice. Very recently, two studies investigated the impact of the sea ice on

the winter low-level clouds in the Southern Ocean and the Antarctic seas (Wall et al., 2017), and on the summer and spring low-level clouds (Frey et al., 2018), and we compare our results to theirs in section 5.2. For the four seasons we distinguish between the general low-level CLD fraction and SLW fraction. We also keep investigating the complementary all-ice fraction to contrast with the observations made for SLW. As we do not have the information at the DARDAR footprint about the presence or absence of sea ice at the sea surface we use monthly products of sea ice fraction provided by NSIDC at a resolution of 25 km

x 25 km, which we project on the grid we use to map the cloud occurrences (Figure 2i-l). Hence, we work with the distribution of sea ice (spatial) fraction derived at a monthly time-scale, and we investigate in all gridboxes how often a given type of cloud layer forms on a monthly-average basis given the monthly-average sea ice fraction in that gridbox. In Figure 14 we show the low-level CLD, SLW and all-ice fractions as a function of sea ice fraction. We also give the Spearman's rank correlation coefficient computed for each seasonal sample (over the four years). We also report the amplitude of change in the different

fractions between areas where sea ice fraction is < 5%, and areas where the sea ice fraction is ≥95 %. In the following we will refer to this quantity when speaking of difference in the average low-level fractions between open water and sea ice. All the changes of fraction between open water and sea ice were found to be satistically significant at the 99 % level (using a t-test).

A clear signal of decreasing low-level CLD fraction as the sea ice fraction increases is detected in autumn, winter and spring (Figure 14b, c and d), while it is much less in summer, for which the negative correlation is the weakest (-0.25) and the absolute




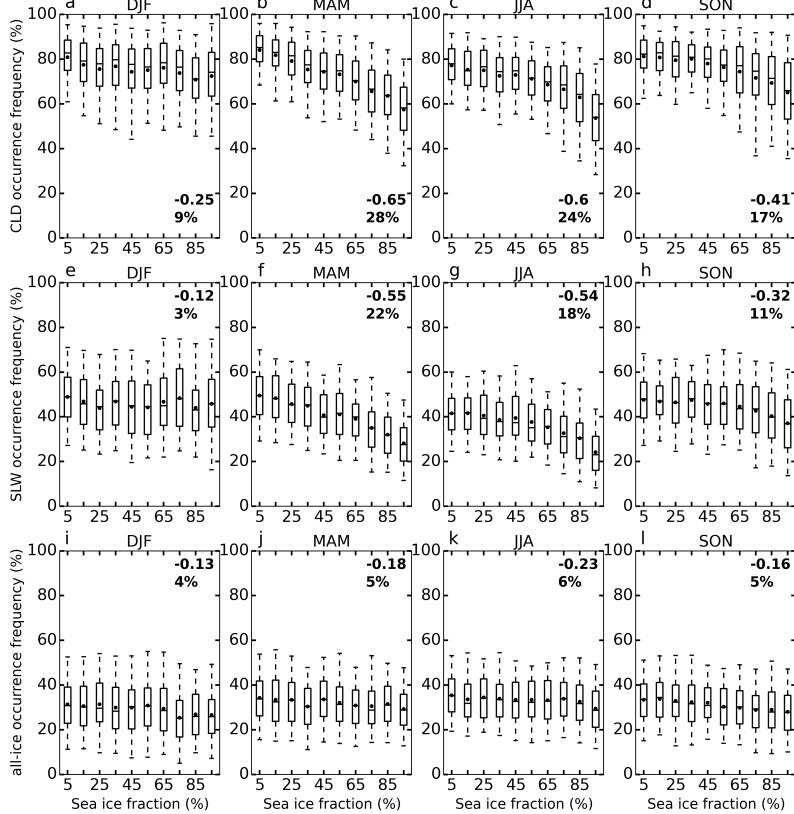

**Figure 14.** Whisker plots with the median (horisontal lines), the mean (circles), the first, and third quartiles, and the 5th and the 95th percentiles) showing - as a function of the sea ice fraction - the evolution of the low-level CLD fraction (top-row), the low-level SLW fraction (mid-row) and the low-level all-ice fraction (bottom row). The sea ice bin width is 10 %, and the center value of each bin is labelled on the x-axis. Each column represents one season. The Spearman's rank correlation coefficient between the low-level cloud fraction and the sea ice fraction is derived over the whole sample that is used to then compute the whisker plots, and it is indicated at the top right corner of each subplot. The p-values for the correlation coefficients are always <0.01. The absolute difference (in %) of occurrence frequency (fraction) of clouds between open water (sea ice fraction <5 %) and sea ice (sea ice fraction >95 %) is indicated below the correlation coefficient.

change in CLD fraction between open water and sea ice the smallest (9 % ). The largest anticorrelation occurs in autmn (-0.65) along with the largest difference in CLD fraction between open water and sea ice (28 %). Note that an equivalent signal is not found for the mid- and high-level CLD fractions (not shown). Comparing the seasonal plots for the SLW fraction (Figure 14e-h) and for the all-ice fraction (Figure 14i-l) we conclude that the correlations found between low-level CLD fractions and sea ice fraction are largely due to the SLW fraction changes, rather than to the all-ice fraction changes. The latter show the weakest correlation with sea ice fraction changes (-0.23 to -0.13) with a constant change in amplitude between open water and sea ice (∼5 %) throughout the year. In contrast, the SLW fraction shows the most pronounced anticorrelation with sea ice fraction in autumn (-0.55 and 22 % change) and winter (-0.60 and 18 %), while it is weaker in spring (-0.41 and 17 %). In





summer, the SLW fraction shows the weakest correlation coefficient of all coefficients derived here (-0.12) and the smallest seasonal change between open water and sea ice (3 %). The contrasting behaviour between the SLW clouds and all-ice clouds demonstrate that the latter are not driven by sea ice variability, at least to the point of inducing clear changes in their occurrence when sea ice changes.

5    In Figure 15 we further distinguish between low-level MPC and USLW layers since distinct monthly time-series prevailed for the low-level MPC fraction and USLW fraction (Figure 9d and e). The strongest anticorrelations with sea ice fraction occur for the low-level MPC, which also show the largest change in fraction between open water and sea ice. The strongest correlations for the MPC fractions is observed in autumn and winter with a correlation coefficient of -0.54 et -0.51, and 14 % and 11 % change in fraction between open water and sea ice, respectively. The difference between open water and sea ice, and the anticorrelation with sea ice for the USLW fraction is about twice lower than for the MPC fraction. In spring the anticorrelation weakens and so does the fraction change between open water and sea ice, especially for the USLW layers (1 % change against still 9 % for MPC). The summer months display strikingly different patterns. The USLW fractions are larger by 5-10 % than the MPC fractions, with the largest differences at larger sea ice fractions. The USLW fraction remains within 2 % and no correlation with sea ice fraction is detected (0.03). However, the MPC fractions remains weakly correlated to the sea ice fraction (-0.22) with a 6 % change in fraction between open water and sea ice. With respect to the anticorrelations between MPC or USLW fractions and sea ice fraction the spring months appear as intermediate between autumn and winter, and summer. The MPC display always some anticorrelation with sea ice fraction, while the USLW layers are not correlated to sea ice fraction in summer and less than MPC fractions in other seasons. These observations suggest a stronger link between MPC and sea ice, than between USLW and sea ice. More generally it shows that the low-level CLD fraction variability as a function of sea ice is more driven by the one of the low-level MPC. This strongly points to a link between glaciation process in clouds and sea ice variability.

In order to illustrate the way sea ice spatial variability affects the lower atmosphere, we derive the difference in potential temperature between the surface pressure level ($\theta_{\mathrm{SLP}}$) and the 850 hPa pressure level ($\theta_{850\mathrm{hPa}}$) at all seasons, for all years. This is a indicator of the coupling between the sea surface and the atmosphere (e.g. Klein and Hartmann, 1993; Kay and Gettelman, 2009). The pressure level 850 hPa roughly corresponds to a representative altitude level where low-level mixed phase clouds are found. A larger difference of potential temperature indicates a stronger surface static stabilty and a weaker coupling between the sea surface and the atmosphere. In this case, the exchange of heat and moisture between the sea surface and the atmosphere is not favoured. We use the ECMWF temperature and pressure profiles collocated with the satellite overpasses at the DARDAR footprint level and provided with the DARDAR products. The potential temperature difference is plotted as a function of sea ice fractions in Figure 16. We derive the Spearman's rank correlation coefficient between the potential temperature difference and the sea ice fraction, as well as the potential temperature difference between open water and sea ice. The largest near-surface static stability variations between open water and sea ice are found in autumn and winter (9K and 7K, Figure 16b and c respectively), and intermediate stability variation is found in spring (4K amplitude, Figure 16d), while no dependency of the sea surface-atmosphere coupling on sea ice fraction is observed in summer (Figure 16a). This is in agreement with the differences between open water and sea ice observed for the low-level SLW, MPC and USLW fractions. This difference gets smaller in



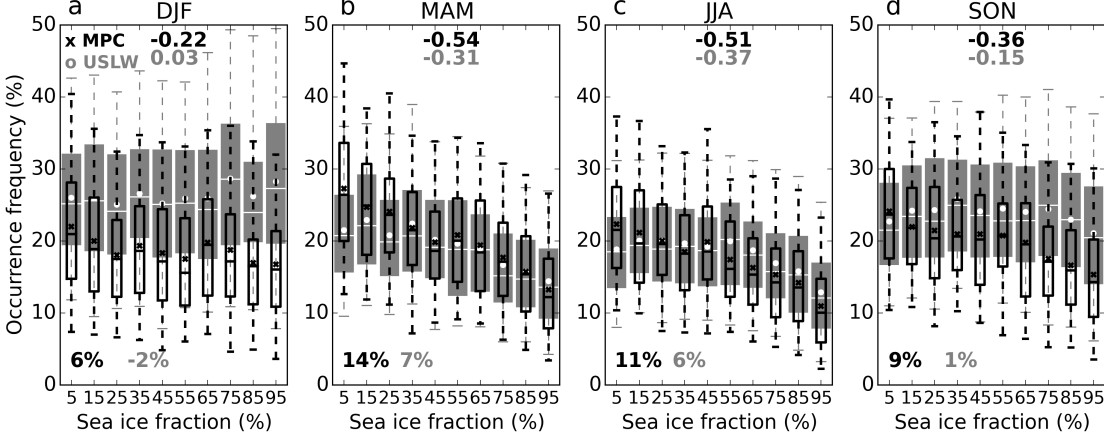

**Figure 15.** Same as Figure 14 for the low-level MPC (narrow empty black boxes, with a cross showing the average) and USLW (larger shaded grey boxes, with a circle for the average) fractions. The p-values for the correlation coefficients are always <0.01, except for the USWL fraction in summer (p-value=0.06).

autumn, winter, spring and summer, in that order. The same observation can be made for the strength of the correlation with the sea ice fraction, which is the largest for both the near-surface stability and the SLW (MPC) fraction in autumn, winter, spring and summer in that order. These results support the idea that the decrease in the SLW fraction (from which mainly the MPC fraction) with the increase of sea ice fractions is caused by a reduction in the coupling between the sea surface and the atmosphere. It is interesting to note that in summer, however, while the strength of the coupling between the sea surface and the atmosphere does not vary as a function of sea ice fraction, the MPC fraction still shows some variation. This will be discussed below.

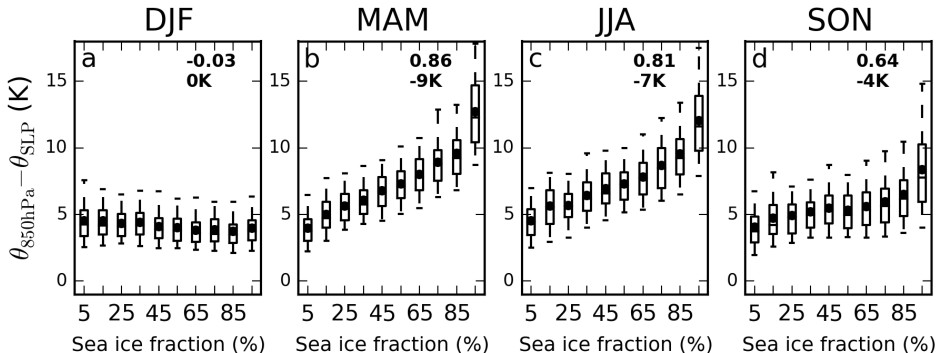

**Figure 16.** Whisker plots showing the potential temperature difference $\theta_{850hPa}$-$\theta_{SLP}$, as a function of the sea ice fraction for each season. The potential temperatures are derived from the ECMWF temperatures and pressure profiles collocated with the satellite overpasses, at the DARDAR footprint level.



## 5 Discussion

### 5.1 On the clear differences between SLW and all-ice clouds

The average total tropospheric CLD fraction in the Antarctic region is around 68 % anytime of the year each year (Figure
9a), demonstrating no Antarctic-wide seasonal or interannual variability. However the Antarctic-wide SLW fraction decreases

from ∼47 % in summer to ∼23 % (Figure 9b). The complementary all-ice fraction increases accordingly. The largest seasonal
variability for the CLD fraction is overall found on the Plateau as noted by previous studies (Verlinden et al., 2011; Adhikari
et al., 2012) and it is exclusively due to the all-ice fraction given the same values of CLD and all-ice fraction over the Plateau
(Figure 9a and c). The lowest seasonal CLD fractions are found in the eastern part of the Plateau in the so called Megadune
region (120° E-160° E;75°S-82° S) in summer (Figure 5). This minimum in the CLD and all-ice fractions is not detected

in the SLW fraction whose distribution is more zonally homogeneous in EA (Figure 5e-h). The Megadune region actually
witnesses the lowest occurrence of clouds at anytime of the year. Here, the weak anticyclonic continental circulation creates
a center of higher pressure (Figure 2a) associated with the strongest subsidence of air in that region (Verlinden et al., 2011).
Consistently, it also corresponds to an area where the lowest snowfall rates were mapped by Palerme et al. (2014), and more
particularly to the areas with the lowest contribution by snow to the overall accumulation (see their Figure 6). Interestingly, it

is also the area with the most blowing snow events reported from CALIPSO observations (Palm et al., 2017). This could be
partly explained by the higher ability to detect those blowing snow events in an area of minimum cloud fraction. The choice
of a lower altitude cut-off at 500 m above the surface does not bias our results and discussions as shown in the Appendix B. It
has the smallest impact on the all-ice fractions (Figure B1c). and the seasonality of the clouds and the different phases is not
altered. Suppressing this cut-off mainly changes the low-level USLW fractions (Figure B1e), because they can be detected by

the lidar down to the surface. However, since it is in the radar blind zone it is not possible to say wether these USLW fractions
are not actually MPC fractions. But suppressing the cut-off does not change the shape of the MPC monthly evolution in places
where the radar can still assess the presence of ice (Figure B1d). The ground clutter quickly reduces the number of available
observations between 500m and the surface but it is not zero (Figure A1a).

The changes in the SLW and all-ice fractions do not occur at similar altitudes and are not driven by similar mechanisms.

The monthly variability in all-ice fractions increases with altitude over marine regions (from 5 % at low-levels up to 30 %
at high-levels, Figure 10d-f). The all-ice fraction maximise in winter and autumn as a consequence of the increase in storm
activity and deep clouds (Adhikari et al., 2012). Over the continent the changes in all-ice fractions are large already at low
altitude levels, because of the interaction of airmasses with the orography, and especially where the permanent lows around
the continent are (Figure 6a-d). In these places the vertical extension of the CLD (or equivalently, all-ice) fraction is also the

largest, especially in winter (e.g. Figure 7h an di) with the strengthening of the lows and the increased cyclonic activity near the
coasts (Simmonds et al., 2003). The overall amplitude of change of the low-level all-ice fraction is larger on the WAIS (∼20
%) than on the Plateau (15 %). This can be explained by the location of the WAIS closer to the sea and to the ASL. At mid- and
high-levels the continental all-ice fractions increases by 30-35 % in winter as a consequence of synoptic scale systems made of
deep clouds, reaching the interior. The SLW fraction decreases poleward, reaching <1 % at 82° S in EA, compared to 35 % in



WA. This difference is due to the increasing surface height poleward, which reaches higher altitudes in EA where the Plateau lies, with the coldest temperatures year-long.

## 5.2 On the links with sea ice

In contrast to the all-ice fraction, the seasonality of the SLW fraction is largely driven by its low-level part. It is determined by
the tropospheric temperature and sea ice fraction seasonality. This result is in line with the observations made over the Ross Sea and Ice Shelf by Jolly et al. (2018), that the occurrences of liquid-containing clouds varied more as a function of seasons than of circulation regimes in that region. Our results suggest it is actually the case Antarctic-wide. The anticorrelation of low-level CLD fraction with sea ice fraction is clearly due to the SLW fraction and not to the all-ice fraction (Figure 14). This points to an exclusive link between liquid-containing cloud formation and sea ice fraction evolution. The largest anticorrelations between
the low-level SLW fraction and the sea ice occurs in autumn and winter (Figure 14f and g) when the sea surface-atmosphere coupling also shows the strongest anticorrelation with sea ice fraction (Figure 16b an c). No correlation occurs in summer (Figure 14e), when no variation of sea surface-atmosphere coupling with the sea ice fraction is observed (Figure 16a). The spring case 14h) is intermediate between winter and autumn, and summer. Sea ice extent is at its lowest in summer so, at this time of year, more sea ice points will be close to the ice edge than in other seasons and the vertical (potential) temperature
gradient will be the smallest. Hence, in summer, advection of cloud may be a more important control on cloudiness at a given location than local processes controlled by sea ice concentration. The lack of correlation between the SLW fraction and the sea ice fraction in summer (Figure 14e) is consistent with the lack of cloud cover differences between open water, and sea ice found by Frey et al. (2018) in the Southern Ocean, and with the similar findings in the Arctic (Kay and Gettelman, 2009; Taylor et al., 2015; Morrison et al., 2018).

The reduction in our winter SLW fraction between open water and sea ice (18 %) is strikingly similar to the reported value by Wall et al. (2017) (17 % between areas with sea ice fraction of 95 % and areas of sea ice fraction of 0 %). Wall et al. (2017) investigated the effect of advection of cold air off of the sea ice edge (low-level jets), which caused clouds to form right above the nearby open waters, in the Southern Ocean. This agreement occurs despite the fact that we use a combined radar-lidar product, while they use a lidar-only product ((GCM-oriented CALIOP products developed by Chepfer et al., 2010), and
despite that they work on the 2006-2014 period, while we investigate the 2007-2010 period. This is consistent with the fact that it is our SLW part of the CLD fraction which anticorrelates with sea ice fraction (Wall et al. (2017) do not differentiate between SLW and CLD). Our low-level CLD fractions are 78 % and 54 % over open water and sea-ice, respectively. Wall et al. (2017)'s cloud fractions are ∼70 %, and ∼50 (Their Figure 9a). Our larger values of cloud fraction can be explained by our additional use of the radar while Wall et al. (2017) will miss more low-level clouds due to lidar signal extinctions. Nonetheless
the absolute values are in agreement by <10 %.

In spring, the 11 % difference in the SLW fraction between open water and sea ice is larger than the one reported by Frey et al. (2018) over the Southern Ocean (4 %). This difference is even larger (17 %) when considering our low-level CLD fraction, mainly because of a slight difference (5 %) also seen in the all-ice clouds. Our low-level CLD fraction in spring is 81 % over open water and 64 % over sea ice. Their low-level cloud fraction is 68 % over open water and 64 % over sea ice. While our



respective low-level cloud fractions are the same over sea ice (64 %), they are strikingly different over open water (81 % vs. 68 %). Since this attenuation will not affect the radar and that deeper clouds will form over open water than over sea ice (Wall et al., 2017; Frey et al., 2018; Morrison et al., 2018), this may explain the difference between our and their cloud detections over open water only. It is unclear, though, what else could contribute to this difference over open water. In summer, a similar

difference occurs between our value (81 %) and their value (68 %) over both open water and sea ice. The occurrences of lidar attenuation are actually larger than in spring and particularly between 2000m and 3000m asl (Frey et al., 2018, their Figure 8). It is beyond the scope of the paper to further compare lidar-only and radar-lidar products. However, this should be addressed in future studies.

Breaking the SLW fraction down to its MPC and USLW components, we showed a stronger correlation with sea ice fraction

for the MPC fraction than for the USLW fraction (Figure 15). Interestingly in summer the USLW fraction shows no anticorrelation with sea ice fraction while the MPC fraction does to some extent (Figure 15a). It is in spring and summer that both fractions differ the most (Figure 15a and d) in both their values and anticorrelation strength with sea ice. This can be related to the different monthly variability of the MPC and USLW layers (Figure 10g and j). In order to discuss in more details these differences we plot the monthly time-series for each marine region separately (Figure 17). We also add the whole Antarctic

seas and the ocean (Figure 17a) and - as element of comparison - the Antarctic continent for surface height above 1.5-3km asl, i.e. away from the coasts (Figure 17f). The monthly time-series of the sea ice fraction is also represented.

Looking at the differences between monthly time-series of MPC fractions and USLW fractions as well as the time-series of the sea ice fraction, it can be understood why the MPC fractions weakly anticorrelates with the sea ice fraction in summer while the USLW does not at all in Figure 15a. This is related to the maximum of the MPC fraction by February or March which

is concomitant of the minimum in sea ice fraction (Figure 17a-d). In contrast, the USLW fractions decreases from January onwards. This difference between MPC and USLW is not observed on the continent, away from the coast (Figure 17f. It is observed when considering all the Antarctic marine regions, though (Figure 17a). On the continent, however, both seasonal cycles of the MPC and USLW fractions have the same pattern. The further away from the coast, the clearer the increasing phase, and the more marked the maximum, of the MPC fraction is by the end of summer and beginning of autumn (compare

ARsec and Wsec with WS and RS). Wsec and ARsec are areas where the clear increase in the sea ice fraction starts later (March-April) than in WS and RS (February). Also, the monthly evolution of the MPC fractions has a larger amplitude for larger monthly sea ice variations (compare Wsec and ARsec, in Figure 17b and c) as well as - consistently - a larger amplitude in the surface static stability monthly variation (not shown). In autumn, the MPC fraction decreases more strongly than it does in ARsec and gets below ARsecs values by the end of that season. In winter, the MPC fraction in Wsec is smaller by 5-10 %

than in ARsec (Figure 10g). In parallel, the sea ice fraction changes from 0.05 to 0.95 in Wsec while only from 0 to 0.6 in ARsec. Overall, these observations strongly points to a role of the sea ice breaking or melting in the seasonality of glaciation process leading to MPC layers. The "surplus" of anticorrelation with sea ice found for MPC compared to USLW (Figure 15a) hints towards an additional factor playing in the link between sea ice and the formation of a mixed-phase. It would come on top of the role of sea ice melting in the strengthening of sea surface-atmosphere coupling and the release of moisture that drives

the formation of SLW as a whole.



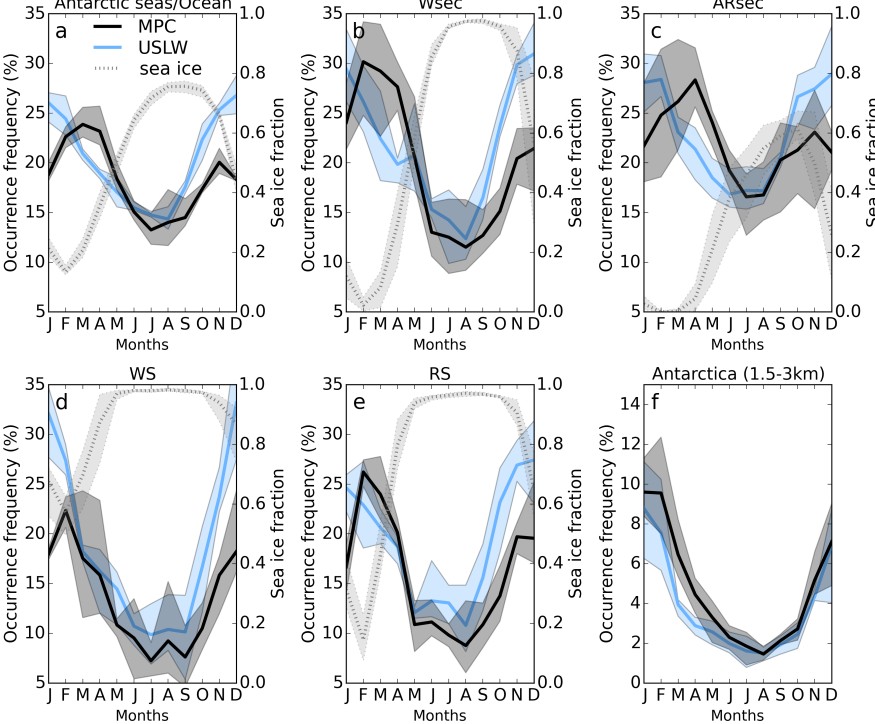

**Figure 17.** Four-year (2007-2010) average monthly time series of the low-level MPC fraction (black line) and the low-level USLW fraction (blue line), for the whole Antarctic waters (a) and for Antarctica (f) for surface heights between 1.5 km and 3 km asl, as well as for the marine regions defined in Figure 4c. In each plot, the monthly time-series of the sea ice fraction (dotted line, labelled on the right y-axis) is also plotted when relevant. The shaded areas indicate the amplitude of variation between the maximum and the minimum monthly averaged fractions recorded over 2007-2010.

The near-surface conditions (temperature and water vapour) are the same for the MPC and USLW layers above waters (Figure 11c and d). Consistently, the surface static stability below either MPC or USLW occurrences is the same (not shown). This suggests that it is not a stronger coupling of the sea surface and the atmosphere that triggers the glaciation of liquid layers, notably by adiabatic cooling caused by the enhanced upward motions. The average altitude of our detected MPC layers is

5   higher than the USLW layer, and the former are logically associated to colder temperatures (hence more glaciation process). In our observations, differences in surface atmospheric states cannot explain the stronger correlation of MPC fraction with sea ice fraction than with USLW fraction. It is not the temperature seasonal cycle that creates the different monthly variation of the MPC fraction from the one of the USLW fraction since the temperature measured at the top of these clouds shows a simpler and similar seasonal evolution (Figure 11a). Recall also that it cannot be an artefact caused by the radar signal loss since the

10   radar clutter occurrences do not show any seasonality (Figure A2a).





### 5.3 On the marine origin of the (biological) INPs

The remaining option is to consider the aerosol, and more particularly the availability of ice nucleating particles (INPs). The open ocean is a documented source of INP via sea spray emissions (Burrows et al., 2013; Wilson et al., 2015; DeMott et al., 2015). INPs are most probably of organic origin (phytoplankton exudates) (according to studies performed in the Arctic seas

Wilson et al., 2015), and their emission is possibly favoured by blooming events (from laboratory experiment DeMott et al., 2015). The marine MPC layers that we detected formed at an average temperature range of -15°C- -5°C (Figure 8A). At these temperatures samples of collected particles were found to be active INPs in the immersion freezing mode (Wilson et al., 2015; DeMott et al., 2015).

The Antarctic aerosols number concentration - of which the INP will be a subset - has been shown to have a seasonal cycle

that is well documented. For instance, Weller et al. (2011) and Kim et al. (2017) documented the monthly variation of the total aerosol concentration on the coast in the Weddell Sector (Neumayer Station) and at the northern tip of the Antarctic Peninsula (King Sejong Station), respectively. They found that the maximum concentrations occurred in February (Kim et al., 2017, their Figure 6) or March (Weller et al., 2011, their Figure 3). Moreover, the increase in aerosols at the end of the year in November-December is much reduced or paused (Weller et al., 2011). Interestingly, in our monthly time-series by the end of spring,

while the sea ice has already started melting and the USLW is still increasing, the MPC fraction stops increasing as well in November-December (Figure 17a). In these studies, the authors relate the seasonal cycle to the enhanced biological activity that prevails with increased solar radiations and sea ice retreat at the sea surface, where phytoplanktons grow. They further highlight the possible role of new particle formation (NPF) in the variations of the total aerosol number concentration. Enhanced NPF caused by biological emissions could be an indication of enhanced direct emission of primary biological aerosols as well.

Hara et al. (2011) measured the aerosol seasonal cycle at different altitudes using a tethered balloon system in Syowa station (coastal East Antarctica). They demonstrated similar seasonal cycles to those found by Kim et al. (2017). Interestingly, above 1km asl - ie the typical altitudes where we detected MPC layers - the absolute maximum in aerosol concentration occurred at the beginning of February with a local maximum at the start of November (at least in the range 1000m-1500m asl). Hence, the measured seasonal cycle of the aerosols on the coasts show striking similarities with the one of our MPC fractions. Hara

et al. (2011) trace back the highest concentrations with airmasses coming from above the Southern Ocean, emphasising the role of biological activity as an emitter of aerosols or aerosol precursors, of which INP should happen to be a subset. In fact, the melting sea ice initiates phytoplankton blooms and primary production of phytoplankton happens to maximise in January on average (Petrou et al., 2016), and open water reaches its maximum area at the start of February. At smaller scales, though, the production of phytoplankton may appear much more complex, with different local seasonal cycles in a given region (Park

et al., 2010). However, and overall the aerosol monthly variability observed in Antarctica and related to the biological activity at the sea surface, matches the one of the MPC fraction. Previous studies demonstrated a link between the number of cloud droplets and Chlorophyl-a concentration (a proxy for phytoplankton biomass) at the sea surface (McCoy et al., 2015). But this is the first time that an indirect signature of the role of biological activity - or say marine aerosols - in glaciation process is pointed at in the observed monthly evolution of mixed-phase cloud occurrences.



Interestingly, the Weddell Sector (Wsec, Fig 4c) is the area with the highest Antarctic SLW fraction in summer (Figure 5e) in all years but 2009, when parts of the Amundsen-Ross sector (Fig 4c) have equally large values (it is also the area with the largest relative contribution of SLW to low-level clouds as shown by Figure 13a). This regional increase of SLW fractions in summer in mainly due to an increase in MPC fractions (Figure 10g) rather than in USLW fraction (Figure 10j). Interestingly,

the Weddell Sector sits in a part of the Southern Ocean where Chlorophyl-a concentration reaches the highest values and has the largest average values throughout the year compared to anywhere else around the continent (Blondeau-Patissier et al., 2014). This observation is in line with our hypothesis, of the marine MPC fraction monthly evolution being driven by the bioaerosol emissions.

South of Wsec, the Weddell Sea shows up as a particularly favourable area for supercooled liquid formation in summer

(Figure 7Bb). More particularly, the eastern part of it shows up as a hotspot for maintaining USLW layers in that season (Figure 6i and Figure 8Bb) up to 1km asl. No other coastal area shows a similar pattern. This is consistent with recent aircraft measurements during the Microphysics of Antarctic Clouds campaign (MAC) in the summer 2015 in the eastern Weddell Sea, where persistent supercooled liquid layers were observed at around 1 km altitude, with only rare and very localised occurrences of patches of ice (O'Shea et al., 2017). A plausible explanation for this is the dominant easterly circulation there, which brings

more airmasses with more continental influences, hence decreasing the effect of marine INPs. The liquid-dominated layer clouds probed during MAC had almost systematically a cloud top in the range 500-1500 m, and their in-cloud temperature was between -5°C and -15°C (except from one frontal cloud) (O'Shea et al., 2017, their Table 1). This is consistent with the altitude ranges and temperatures reported for the SLW fractions (Figure 8). Comparisons between the MAC aircraft measurements and summer satellite observations will be carried out in a separate study. Importantly, (O'Shea et al., 2017) demonstrated the

importance of secondary ice production in the formation of the localised patches of ice in the eastern Weddell Sea and this mechanism was also evidenced in the Antarctic Peninsula region in (Grosvenor et al., 2012; Lachlan-Cope et al., 2016). Given the relatively warm temperatures at which this ice multiplication process is thought to occur (Hallett-Mossop process between -3°C and -8°C; Hallett and Mossop, 1974), we can expect it to happen mainly in the lowest clouds, where the radar may not be always able to detect the presence of ice (thereby distinguishing between USLW and MPC layers). In the Weddell Sea, our

SLW occurrences are mainly USLW occurrences at temperatures larger than -10°C and altitudes higher than 500m asl (Figure 7Bb and Figure 8Bb) and the rare patches of ice (O'Shea et al., 2017) may be too small for the radar footprint to resolve them.

It is possible that USLW or MPC layers are more prevalent than suggest by our study at low altitudes since the detection of an SLW layer at - say - 1000m asl will prevent the detection of another SLW layer, and possibly a MPC, below. To this respect, year-long ground-based measurements of microphysical properties of the mixed-phase and the primary ice production appear

as a needed complement. Note that (O'Shea et al., 2017) found no trend of an increased number of crystals with airmasses originated from the Southern Ocean. This is contrary to our hypothesis of marine INP driving the glaciation process. However, the limited number of localised patches of ice probed by the aircraft on a restrained period of time, in an area relatively more influenced by continental airmasses, might have rendered this task difficult. Note that another area with higher USLW occurrences than MPC occurrences was the interior of the WAIS (Figure 8Bc), an environment relatively more influenced





by continental airmasses devoid of marine INPs. Our results suggest that satellite observations and their large statistics may indirectly help answer the question of the primary ice production - the first ice - in Antarctic clouds.

Finally, biological activity can also create cloud condensation nuclei (CCN). Why do not we then observe a similar pattern in the USLW seasonal cycle? Recall that we observe larger regional relative differences between the MPC fractions than

between the USLW fractions (Figure 9d and e). This suggests a regional dependence on INP availability and much less on CCN availability. While CCN emission can at least partly be the result of biological activity (McCoy et al., 2015) as initially put by the well-known CLAW hypothesis (Charlson et al., 1987), they are also provided by sea salt emission via bubble-bursting (Quinn and Bates, 2011) or blowing snow from over the sea ice (Yang et al., 2008; Legrand et al., 2016). Hence the availability of CCN is probably much less dependent on biological activity (Quinn and Bates, 2011). Sea salt is not an INP at

warm temperatures (Burrows et al., 2013) like the ones at which we detect MPC layers ($T< -15°C$). Another open question remains regarding the aerosol seasonal cycle over the Plateau (Fiebig et al., 2014, and references therein) and its outskirts where it is argued that the aerosol baseline originates from the free troposphere and the lower stratosphere in the descent of air happening at these locations (Fiebig et al., 2014), and how these aerosols might or not affect cloud formation and, or, glaciation process in the continental clouds.

**6    Conclusions**

We demonstrated the geographical, vertical and seasonal distribution of the supercooled liquid water (SLW) in the Antarctic region (60° S-82° S) using the radar-lidar DARDAR-MASK v2 products. We described it in comparison to the total cloud (CLD) fraction, and notably to clouds involving only ice microphysics (all-ice). The combination of the radar and the lidar signals allowed to further distinguish between mixed-phase clouds (MPC) and unglaciated (pure) supercooled liquid water

layers (USLW) at the pixel level. The Antarctic-wide average total CLD fraction - derived with DARDAR for altitudes above 500m above the surface - is around ~70 % and has little internannual variability (≤5 % absolute variation). The Antarctic-wide total SLW fraction varies from ~50 % is summer to ~20 % in winter, while it is the opposite for the all-ice fraction. The Antarctic-wide total MPC and USLW fractions have distinct seasonal cycles. While the USLW fraction is maximum in December- January (~20 %) and minimum in August (~10 %), the MPC fraction maximises at the end of summer (February,

~30 %) or in autumn, and it is minimum in July-August (~10 %).

The total CLD fraction has the largest monthly variability on the Plateau while its evolution on the WAIS shows an almost constant fraction from February to October (65-70 %). However the continental all-ice fractions maximise in winter and are largest on the WAIS (60 %) than on the Plateau (50 %), and the monthly variability is larger over the WAIS (+40 % absolute difference from summer to winter) than over the Plateau (+30 %). This results from the WAIS being in closer contact to the

ASL, which leads to orographic (all-ice) cloud formation. The geographical and vertical distribution of the all-ice fraction is shaped by the orography and its interactions with the permanent low-pressure systems, which are located around the continent, more particularly on the West Antarctic Ice Sheet (WAIS) and south of the Amery Ice Shelf (AIS). This is particularly evident in winter. In all marine and continental regions the all-ice fractions maximise in winter, when cyclonic activity increases and



storms are more numerous over seas. On the continent the CLD fraction monthly variability at mid- and high-level is the largest of the whole Antarctic region and it is exclusively driven by all-ice clouds. Over the Antarctic seas and the Ocean, the total CLD fraction monthly variability is driven by the low-level SLW fraction variability. Conversely, the mid- and high-level CLD fraction are driven by the mid- and high-level all-ice fractions.

The geographical distribution and seasonality of the SLW fraction is shaped by the temperature and the sea ice fraction seasonal evolution, which drives the amount of water vapour released into the atmosphere. On the continent the SLW fraction decreases polewards as a results of decreasing temperatures. It is minimum on the Plateau, where it reaches almost zero ($<1$ %) by winter. We validated our observations of SLW layers by comparing them to ground-based measurements from a micropulsed lidar made at South Pole Station in 2009. We demonstrated the representativity of South Pole for lower latitudes on the Plateau,

in terms of supercooled liquid water measurements. On the WAIS, where the largest continental occurrences of SLW fractions are found, the detected SLW layers in summer are mainly in the form of pure supercooled liquid water and not mixed-phase layers, which reminds of the characteristics of lenticular mountain wave clouds. In marine regions, MPC layers are principally detected between 1km and 1.5km asl in marine regions, while USLW layers dominate between 0 and 1km asl. The temperature range characteristic of marine MPC layers is -15°C to -5°C.

The low-level SLW fraction is responsible for the anticorrelation of the low-level CLD fraction with the sea ice fraction, while the all-ice fraction does not show a clear dependency on the sea ice variability throughout the year. The strongest anti-correlations of SLW with sea ice fraction occur in autumn and winter, when the surface static stability (sea-surface atmosphere coupling) also shows the strongest response to varying sea ice fraction. The SLW fraction decreases by 22 % and 18 % from open water to sea ice, in autumn and winter, respectively. In summer, little anticorrelation is found between the SLW fraction

and the sea ice fraction, in agreement with a lack of correlation between surface stability and sea ice fraction. Our results are in agreement with recent studies investigating links between sea ice in winter, spring and summer using lidar-only products.

    The monthly time series of the MPC fraction shows a distinct maximum by the end of summer or beginning of autumn, which is absent for the USLW fraction, which maximises at the end of December. This difference is observed over marine areas and not over the continent and it is more marked for the Weddell and Amundsen-Ross Sectors than for the Weddell and

the Ross seas, which are closer to the coast. Importantly, the monthly variations of the MPC fraction matches the documented seasonal cycle of aerosols in Antarctica, which are driven by the biological activity in spring and summer when sunlight increases. To our knowledge, this is the first time that a link is made between the seasonal cycle of aerosols (of which the INPs are a subset), and the seasonal cycle of mixed-phase clouds over the Antarctic waters. Based on the literature, our results point to the signature of INP emissions from biological activity at the sea surface in our mixed-phase cloud monthly evolution.

Using satellite products we provided constraints on the distribution of supercooled liquid water Antarctic-wide, and its monthly or seasonal evolution. The radar-lidar synergy also appears as a promising tool for pinning down some fundamental links between polar clouds glaciation process, supercooled liquid water and biological activity at the sea surface, in association with sea ice variability. We plan to extend our investigation with DARDAR products to more recent years as well as to compare with more recent field campaigns measuring supercooled liquid water. Surface-based and aircraft measurements of cloud



microphysical properties and of the nature of the aerosols, on the coast and off-shore, would help test the hypothesis of the MPC evolution being modulated by the release of marine (biological) INPs.

*Data availability.* The DARDAR v2 are available on the AERIS/ICARE Datacenter (http://www.icare.univ-lille1.fr/)).

### Appendix A: Radar clutter and lidar extinction

Contamination of the radar signal by surface echoes and lidar extinctions or attenuations due to optically thick ice clouds, and supercooled liquid layers, reduce our statistics as we near the ground. Figure A1 illustrates this for the summer season (when the lidar signal extinctions are the more numerous, because of higher occurrences of SLW layers). Transects are derived by averaging the occurrence frequency of each of the signal contamination, over the the three latitudinal bands presented in Fig. 4d. The lidar is considered as extinguished when the surface is not detected. Then all the pixels below the last one detected 10 with a signal are flagged accordingly (Ceccaldi et al., 2013). The lidar is considered attenuated when the surface is detected but some features detected by the radar are not seen by the lidar, and the corresponding pixels are flagged accordingly (Ceccaldi et al., 2013). The radar signal is contaminated below 1 km above the ground, with a ∼40% loss of valid observations at 500m above the surface (Figure A1a, b and c). In the coastal areas the contamination occurs more often and affect higher altitudes over terrains with steep slopes (Figure A1b, compare 0° E-40° E and 50° E-150°E). The loss in the lidar signal are more 15 pronounced as the lidar gets extinguished or attenuated at higher altitude also because of thick clouds. The lidar extinctions-attenuations depend on the season and more (less) loss will occur at high (low) altitudes in winter over the interior of the WAIS (the ocean) because of more (less) numerous thick clouds (SLW layers).

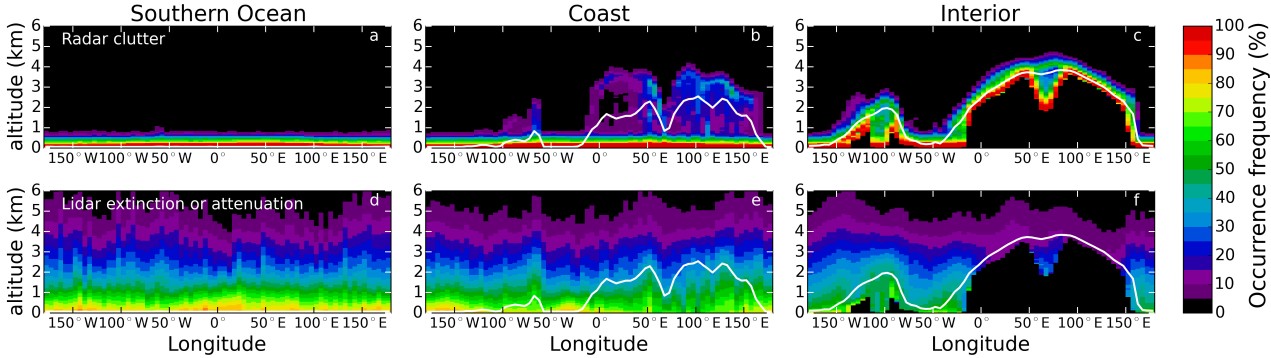

**Figure A1.** Vertical transects (2007-2010 average) showing the occurrence frequency (%) of the radar clutter contamination (top row) and the lidar signal extinctions or attenuations (bottom row). The transects are averaged over the three latitudinal bands presented in Figure 4d, namely the Southern Ocean (a and d), the Coast (b and e) and the interior (c and f). The data is plotted for the summer season.





The seasonality of the occurrences of the instruments obstruction are shown in Figure A2. The only type of signal obstruction showing a clear seasonality are the lidar extinctions and this is caused by the seasonality of the SLW fractions as discussed in the main text. The radar clutter is responsible of a loss of ∼40% of the data at ∼500m above the surface (the occurrences of the clutter are negligible above 1000m above the surface).

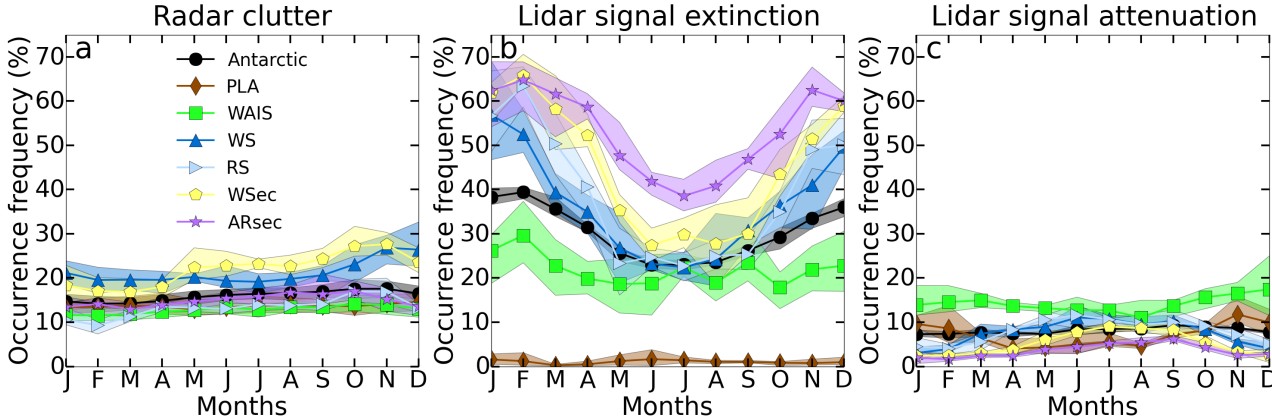

**Figure A2.** Monthly time-series of the low-level occurrences of the radar clutter (a), the lidar extinctions (b) and the lidar attenuations (c) for the areas defined in Figure 4c and averaged over 2007-2010. The shaded areas correspond to the interannual variability.

## 5 Appendix B: Lower altitude cut-off

The lower altitude cut-off chosen for deriving the geographical distribution of clouds and phases will affect this distribution at low-levels. This cut-off is used in the present study to avoid low altitudes where the statistics is significantly reduced because of the radar blind zone and the lidar signal extinctions. A 500m value was chosen in the present paper. Figure A2 shows the impact of changing this cut-off on the values of the low-level cloud and phases occurrences, and on the monthly time-series patterns.

Changing from a 500m cut-off to 1000m cut-off impacts the SLW fraction (Figure B1b), and mostly its USLW component (Figure B1e). There is a 10% difference between the USWL fraction >500m (∼20% yearly average) and the USLW fraction >1000m (∼10% yearly average) over seas, where their occurrences are larger than over the continent. Indeed, a large part of the USLW layers are detected below 1000m. However the monthly relative variations are not impacted and the seasonal cycle shows similar pattern for the three cut-offs. The difference between the MPC fractions >500m and >1000m is ∼2% (Figure

B1d). It is also for the USLW fractions that the difference is the largest between the USLW fraction taken without cut-off and >500m. However, the difference for the MPC fractions is null (Figure B1d) since these clouds are mainly detected above 500m above the surface. An additional ∼5-8% occurrences is added to the USLW fractions when suppressing the cut-off. The all-ice clouds fractions are only marginally affected in winter over the continent (∼5% difference between no cut-off and a 1000m cut-off, Figure B1c ). Overall changing the cut-off does not change the monthly relative variability of the cloud fractions and

the different phases, and this only significantly impacts the low-level USLW occurrences over seas. Below 500m it is difficult





to distinguish between MPC and USLW because of the radar blind zone. Hence, all additional SLW detections are added to the USLW fractions.

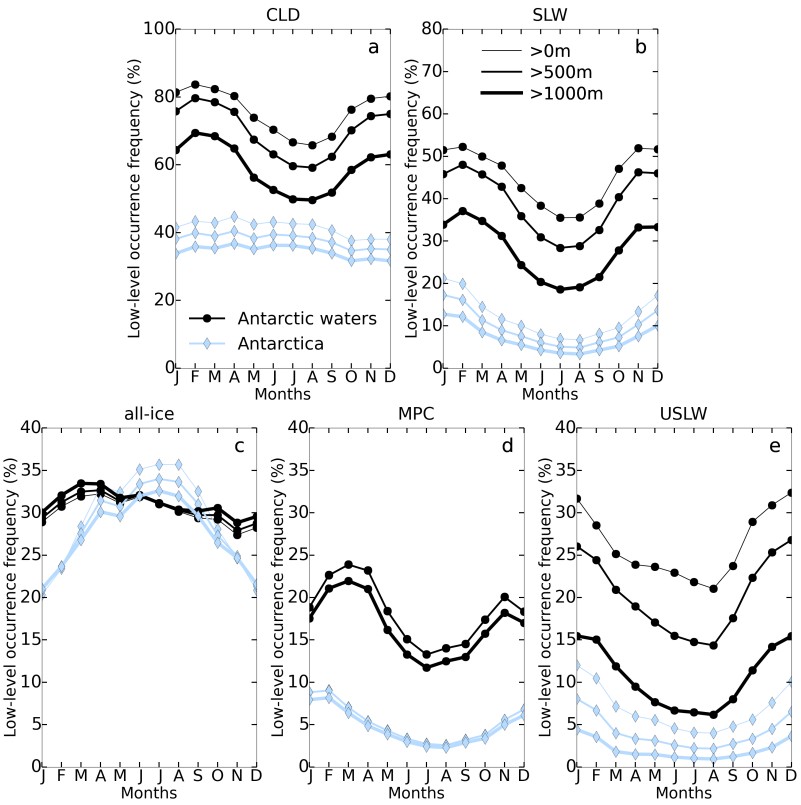

**Figure B1.** Monthly time series of the CLD fraction (a), the SLW fraction (b), the all-ice fraction (c), the MPC fraction (d), and USLW fraction (e), for three different lower-altitude cut-offs: 0m (thinest solid line), 500m (intermediate), 1000m (thickest solid line). They are plotted by distinguishing between continental (blue lines) and marine (black lines) clouds. See section 3.2 for the definitions of the phases.

*Author contributions.* CL performed the analysis and wrote the manuscript. JD helped design the study, use the data, and write the manuscript. AK provided the ceilometer datasets and the related cloud fractions, and helped write the manuscript. TLC and JK advised in the organisation and the writing of the manuscript.

*Competing interests.* The authors declare that they have no conflict of interest.



*Acknowledgements.* CL thanks CNES for Postdoctoral fellowship funding. We are grateful to the AERIS/ICARE Datacenter for generating and storing the DARDAR products (http://www.icare.univ-lille1.fr/)). We thank the NASA CloudSat Project and NASA/LaRC/ASDC for making available the CloudSat and CALIPSO products, respectively, which are used to build the synergetic DARDAR products. We thank NOAA and NSIDC for making available the Sea ice data climate record. We thank the mesocenter ESPRI facility for distributing ERAI

5  products in netcdf format, and which is supported by CNRS, UPMC, Labex L-IPSL, CNES and Ecole Polytechnique.



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
