# Peer review of "Antarctic clouds, supercooled liquid water and mixed-phase investigated with DARDAR: geographical and seasonal variations"

_Atmospheric Chemistry and Physics, 2018_

## Referee Comment (RC1) · Anonymous Referee #1 · 23 Jan 2019

Review of the manuscript "Antarctic clouds, supercooled liquid water and mixed phase investigated with DARDAR: geographical and seasonal variations" by Listowski, C., J. Delanoë, A. Kirchgaessner, T. Lachlan-Cope and J. King

Summary

The study uses DARDAR product (combined satellite-based radar/lidar product) in order to characterize Antarctic tropospheric clouds - distinguishing between supercooled liquid clouds, ice-only clouds, and mixed-phase clouds. A detailed description of spatial distribution, regional behavior and seasonal differences is given. The authors also show the vertical distribution of cloud phase (averaged over 4 years for various transects) and discuss their relationship to temperature. Relationship to sea ice fraction is also explored. Further, a detailed discussion is included (based on literature) about the role of sea ice and snow in IN and CCN production important for cloud phase.

This is a very comprehensive and thorough study bringing both important results and a very useful database to the Antarctic atmospheric community and I recommend the paper for publication. Below I list my minor remarks to improve the readability of the paper.

Minor remarks:

Title: In its present form - I think there a comma needed after "mixed phase": "Antarctic clouds, supercooled liquid water and mixed phase, investigated with DARDAR: geographical and seasonal variations"

However, the title now sounds somewhat restrictive as the authors investigate also ice-only clouds. They authors may consider modifying the title to make it more inclusive of the presented results

page 1:

Abstract:

L 3: "It is the largest over water" => ... over water surface compared to...

L 3-5: please rewrite to make it clearer

In the abstract a clearer distinction has to be made between SLW, USLW, and MPC.

Introduction:

L 21 "Down to the Antarctic Seas" - please rephrase

L 24: "dilemma can be solved" (instead of will)

page 2:

L 7: remove -> removes

L 12: "... track the formation of SLW and the mixed-phase clouds Antarctic-wide" I suggest to add a reference to Lawson and Gettelman 2014 (already in the reference list) who showed the importance of the Antarctic SLW on radiative fluxes both in observations and models, and rephrase the sentence above to emphasize more the existing efforts "to track the formation of SLW and the mixed-phase clouds" from ground-based observations, eg the papers already in the reference list (Lawson and Gettelman 2014, Van Tricht et al 2014, Gorodetskaya et al 2015, Silber et al 2018) and others.

L 16: "cloud science" -> "observations" L 17: I suggest deleting "As a matter of fact"

page 3: L24: "finally sea ice exerts control over the moisture and heat transported..." - the authors can also mention the importance of sea ice and drifting snow in providing cloud nuclei

Section 3.2 Methodology

page 10:

L 14: "above the surface" - please specify which surface - ground level or mean sea level, and implications for the defined cloud levels.

3.3 Ceilometers dataset page 11, L 18: "with a horizontal resolution of 50ft" - I suppose the authors meant vertical range resolution. Please say in meters

As the authors correctly note, there are problems with Vaisala algorithm's identification of cloud base heights in Polar Regions, especially for ice clouds (Van Tricht et al 2014). I suggest adding a justifying sentence that in the present manuscript only cloud fractions derived from ceilometers are compared to the DARDAR product (and no distinction is made for CBH for ice or liquid-containing clouds), and for this application Vaisala algorithm should be enough.

Section 4 Results

page 12, L 13-14: "The deepening of the ASL in winter (Fig 2c) consistently leads to an increase in the CLD fraction over the WAIS (Fig. 5c)." - I suggest replacing "consistently leads" with eg "associated with" as the authors do not show the direct causality between ASL and WAIS cloud fraction, and other factors can be playing a role. My concern is also about the distinction between the "ice-only clouds" and blowing snow. More frequent and more intense storms in winter can be associated with increased blowing snow events - can those be interpreted as "ice clouds" by DARDAR algorithm and contribute to the increase of the "ice clouds" in winter?

The connection between the deepening of the ASL and increased cloud cover in winter over WAIS has to be supported by literature discussing the increased inland moisture flux associated with increased synoptic activity in the Amundsen Sea region. See eg, Dufour et al 2018 and references therein: Dufour, A., Charrondière, C., and Zolina, O.: Analysed and observed moisture transport as a proxy for snow accumulation in East Antarctica, The Cryosphere Discuss., https://doi.org/10.5194/tc-2018-156, in review, 2018.

page 13, L 26: MBL - please write in full Marie Byrd Land (used only once in text and the abbreviation introduced in figure caption) - same for RIS (used here but introduced only on page 19) and AIS.

I suggest to avoid using so many abbreviations and write the geographical region names in full, as there are already many abbreviations used for cloud properties. It becomes difficult to read the text at some point. Also better avoid using abbreviations in section titles.

Figure 6 caption: "...distribution of the total all-ice cloud fraction"

page 15, L 1: The MPC fraction and the USLW cloud fraction..."

The above two comments about using the word "cloud" concerns the entire article:

mentioning of the cloud phase shall be accompanied by the word "cloud" - sometimes it is obvious, but in some places can be confusing. This can be included directly in the abbreviations, Eg, MPC already includes "clouds" (mixed-phase clouds), while SLW doesn't = > changing to SLWC or using "SLW cloud" and similarly for other abbreviations concerning cloud phase.

page 15, L 13: isothermes => isotherms

Section 4.2 page 15, line 26: "The reduced statistics due to the radar blind zone and lidar signal extinction [delete and] across the Antarctic"

Section 4.3, page 20, L7 (over the whole Antarctic region), L11 (for the Antarctic as a whole) - please specify if this means the entire region of the study (Antarctic ice sheet and Southern Ocean and until which latitude) or only the Antarctic ice sheet. As the average value is rather high (68%) I suppose it is the former (Antarctic ice sheet +SO).

Figure 9 caption: please provide the abbreviations also in the caption

Section 4.3.2, page 22, L 26: "The monthly evolution of continental clouds is essentially driven by ice clouds, notably large frontal systems devoid of SLW, as shown in the example in Figure 3b." - I am not sure what the authors mean by "notably large frontal systems devoid of SLW"? Frontal systems associated with extra-tropical cyclones can also bring liquid-containing clouds and the example on Fig 3b shows only one winter case dominated by ice.

Section 4.4.2, page 28, L 21-22: ".. the ice phase in the DARDAR-MASK products include both, cloud ice and precipitating ice" - does it also include blowing snow lifted from the ground in the absence of precipitation?

Section 5.1, page 35 L 5: "... the Antarctic-wide SLW fraction decreases 5 from 47 % in summer to 23 % in winter (Figure 9b)"

[Figure]

2018.

---

## Referee Comment (RC2) · Anonymous Referee #2 · 4 Feb 2019

Review of 'Antarctic clouds, supercooled liquid water and mixed-phase investigated with DARDAR: geographical and seasonal variations' by C. Listowski et al., submitted to ACPD

Listowski et al. use the merged satellite lidar-radar data product 'DARDAR' to characterise cloud phase and variability south of 60S (covering Antarctica and the surrounding oceans) and through the seasons. In doing so, Listowski et al. provide a thorough and comprehensive treatment of the data analyses, and, just as importantly, its limitations (especially with respect to radar ground clutter and lidar attenuation/extinction).

[Figure]

The subject matter addressed is currently highly topical given the known uncertainties with high southern latitude clouds (and radiation) fields in GCMs. The authors demonstrate the links to orography and oceanic-centred cyclones of the ice cloud distribution. Particularly interesting is the differences in seasonality of the MPC and USLW cloud fractions and the links which the authors draw to marine biological activity as a source of INPs, and the links to sea-ice.

This is an excellent manuscript of high interest to the ACP readership. I certainly recommend the publication of this manuscript by ACP following the authors' consideration of the following points as they prepare their revised manuscript.

Minor Comments (references given at the end)

1) There are so many acronyms in the text. I suggest a table in an Appendix listing them all to ease the reader's need to refer back or memorise them.

2) Figure 1: A gap in the contour (black and white) topography is evident south of 82S. No doubt this is due to the fact that CALIOP / CloudSat don't see south of here. However, you really ought to fill in the contour levels between here and the Pole.

3) Page 10, lines 14-16. The authors define low-level clouds as those between 0.5-3km above the surface; mid-level as 3-6km; and high-level clouds >6km. This strikes me as somewhat arbitrary, especially since no rationale for these altitude cutoffs is given. Do you have a convincing argument for choosing fixed levels? While in the tropics a fixed altitude cutoff may be appropriate (such as 4.5km for the fairly-constant freezing level at low latitudes, e.g. Protat et al., 2014, JAMC), in the extra-tropics it is better to work on pressure levels. I suggest the authors change these limits to match those from the ISCCP cloud low/mid/high definitions (Low cloud top pressure >680hPa; middle cloud top pressure>440 hPa etc.). This would facilitate more direct comparisons with previous studies, especially over the Southern Ocean, where pressure levels are regularly used (for example Haynes et al., JClim, 2011; Mason et al., JClim, 2014).

[Figure]

4) Regarding comparisons between surface-based instruments and satellite, this is, as the authors note, a challenge given different temporal / spatial sampling, and indeed, DARDAR curtains likely do not pass directly above the surface sites anyway. One additional option to make closer comparisons would be to spatially average DARDAR and temporally average the ceilometers. For example I found that in a recent Southern Ocean DARDAR/surface lidar study, DARDAR data were horizontally averaged based on typical wind speeds at cloud height (Alexander & Protat, JGR, 2018). I wonder whether applying something similar in Section 4.4.1 above Rothera & Halley would be of benefit, despite the simplicity of this averaging?

5) The authors note that ceilometers detect cloud base heights (page 28, line 5), specifically the Vaisala CBH (Section 3.3). It is known that these are not accurate CBHs in the polar regions, especially for ice (e.g. van Tricht et al., AMT, 2014). Some comment ought to be made about this additional source of uncertainty comparing ceilometer and DARDAR in Section 4.4.1 in the context of your minimum altitude cutoffs.

6) Figure 17: I think that it would be much clearer to interpret if you flipped the sea-ice scale

7) Finally, I suggest a careful, thorough re-read of the manuscript to correct several minor spelling and grammar issues.

References;

Haynes et al., 2011; Journal of Climate, 'Major characteristics of Southern Ocean cloud regimes and their effects on the energy budget, doi:10.1175/2011JCLI4052.1

Mason et al., 2014; Journal of Climate,' Characterizing observed midtopped cloud regimes associated with Southern Ocean shortwave radiation biases', doi: 10.1175/JCLI-D-14-00139.1

Protat et al., 2014: J. Appl. Met. Clim, 'Reconciling ground-based and space-based estimates of the frequency of occurrence and radiative effect of clouds around Darwin,

Australia', doi: 10.1175/JAMC-D-13-072.1

Alexander & Protat, 2018; J. Geophy Res, 'Cloud Properties Observed From the Surface and by Satellite at the Northern Edge of the Southern Ocean', doi: 10.1002/2017JD026552

Van Tricht et al, 2014; Atmos Mes Tech, 'An improved algorithm for polar cloud-base detection by ceilometer over the ice sheets', doi: 10.5194/amt-7-1153-2014
* * *

---

## Author Comment (AC1) · 13 Mar 2019

Please find attached, in the zip file, the (colour-coded) answers to the comments by R1 and the updated version of the manuscript.

Thank you Regards

C. Listowski

Please also note the supplement to this comment:

[Figure]

https://www.atmos-chem-phys-discuss.net/acp-2018-1222/acp-2018-1222-AC1-supplement.zip

---

## Author Comment (AC2) · 13 Mar 2019

Please find attached, in the zip file, the (colour-coded) answers to the comments by R2 and the updated version of the manuscript.

We also report below the answers to the comments by R2.

Thank you Regards

C. Listowski

Please also note the supplement to this comment:
https://www.atmos-chem-phys-discuss.net/acp-2018-1222/acp-2018-1222-AC2-supplement.zip

---

## Author Response (AR1)

P=page of the pdf file

We thank the reviewer #1 for the interesting remarks that lead us to detail some aspects of this work – e.g. related to the use of the ceilometers or the blowing snow considerations.

We first answer the reviewer's questions or comments and then list some changes made by us in the text to improve clarity (sometimes motivated by the other reviewer's comments).

Response to reviewer #1

Minor remarks:

Title: In its present form - I think there a comma needed after "mixed phase": "Antarctic clouds, supercooled liquid water and mixed phase, investigated with DARDAR: geo- graphical and seasonal variations"

Done.

However, the title now sounds somewhat restrictive as the authors investigate also ice- only clouds. They authors may consider modifying the title to make it more inclusive of the presented results

The all-ice category was conveniently introduced to contrast with the fraction of the liquid-bearing clouds (its distributions and the relation to sea ice fraction). However, ice-only clouds may be present above liquid-bearing clouds and as such not be considered by the all-ice category. All-ice is an interesting way to capture ice-only microphysics when it is exclusively present on the entire considered atmospheric column (low, mid, high, all) but it does not include all ice-only clouds. We think "Antarctic clouds…" at the beginning of the title may be enough to implicitly announce the ice cloud considerations of this paper.

Abstract:
Please note that we edited the abstract in order to be more quantitative and not only qualitative. We also added a sentence pointing to the interesting comparisons between ground-based observations at South Pole and our observation at 82°S (on the Plateau).

L 3: "It is the largest over water" => ... over water surface compared to...

Changed.

L 3-5: please rewrite to make it clearer

It is clearer now. The comment related to:

"It is the largest over water. In East Antarctica, the SLW fraction decreases sharply polewards. It is two to three times higher in West Antarctica. The all-ice cloud geographical distribution is shaped by the interaction of the main low-pressure systems surrounding the continent and the orography, with little links with sea ice fraction "

Now this reads:

*"The low-level (< 3 km above surface level) SLC fraction is larger over seas (20-60%), where it varies according to sea ice fraction, than over continental regions (0-35%). The total SLC fraction is much larger over West Antarctica (10-40 %) than it is over the Antarctic Plateau (0-10 %). In East Antarctica the total SLC fraction - in summer for instance - decreases sharply polewards with increasing surface height (decreasing temperatures) from 40% on the coast to <5% at 82°S on the Plateau. The geographical distribution of the continental total all-ice fraction is shaped by the interaction of the main low-pressure systems surrounding the continent and the orography, with little association with the sea ice fraction."*

In the abstract a clearer distinction has to be made between SLW, USLW, and MPC.

In relation to your comment about page 15 L1 we decided to slightly change the way we referred to the different clouds. Now SLC designates Supercooled Liquid water-containing Clouds, while USLC designates SLC where no ice is present. In doing so we are more consistent with the acronym MPC (where C stands for "clouds" – as noted in your comment). MPC remains unchanged. We believe these acronyms are now clearly defined in the abstract (and also introduced in the main text and in Table 1 as request by the other reviewer, as well as changed everywhere in the manuscript, figures, and figure captions.)

The following sentence was added

P 10 L12-13:

*"Table 1 recalls the acronyms used for the various cloud types (as well as the ones for specific Antarctic regions) "*

**Table 1.** Acronyms used in the text to designate some cloud phase or cloud types and some Antarctic places or areas.

| Acronym | Meaning |
| --- | --- |
| SLW | supercooled liquid water |
| SLC* | SLW-containing cloud |
| MPC* | mixed-phase cloud |
| USLC* | unglaciated SLW cloud |
| WA | West Antarctica |
| EA | East Antarctica |
| SO | Southern Ocean |
| WSS** | Weddell Sea sector |
| ARS** | Amundsen-Ross sector |
| WS** | Weddell Sea |
| RS** | Ross Sea |
| WAIS** | West Antarctic Ice Sheet |
| AP** | Antarctic Plateau |
| AIS | Amery Ice Shelf |
| RIS | Ross Ice Shelf |

(*) Their fractions are linked by equation 2.
(**) Geographical areas defined in Figure 4c, and whose names relate to the regions in which they are located.

Edited abstract:

*"Antarctic tropospheric clouds are investigated using the DARDAR (raDAR/liDAR)-MASK products between 60°S and 82°S. The cloud fraction (occurrence frequency) is divided into the supercooled liquid water-containing clouds (SLC) fraction and its complementary part called the all-ice cloud fraction. A further distinction is made between SLC involving ice (mixed-phase clouds, MPC) or not (USLC, for unglaciated SLC). The low-level (< 3 km above surface level) SLC fraction is larger over seas (20-60%), where it varies according to sea ice fraction, than over continental regions (0-35%). The total SLC fraction is much larger over West Antarctica (10-40 %) than it is over the Antarctic Plateau (0-10 %). In East Antarctica the total SLC fraction - in summer for instance - decreases sharply polewards with increasing surface height (decreasing temperatures) from 40% on the coast to <5% at 82°S on the Plateau. The geographical distribution of the continental total all-ice fraction is shaped by the interaction of the main low-pressure systems surrounding the continent and the orography, with little association with the sea ice fraction. Opportunistic comparisons with published ground-based supercooled liquid water observations at South Pole in 2009 are made with our SLC fractions at 82°S in terms of seasonal variability, showing good agreement. We demonstrate that the largest impact of sea ice on the low-level SLC fraction (and mostly through the MPC) occurs in autumn and winter (22% and 18% absolute decrease of the fraction between open water and sea ice-covered regions, respectively), while it is almost null in summer and intermediate in spring (11%). Monthly variability of the MPC fraction over seas shows a maximum at the end of summer and a minimum in winter. Conversely, the USLC fraction has a maximum at the beginning of summer. However, monthly evolutions of MPC and USLC fractions do not differ on the continent. This suggests a seasonality in the glaciation process in marine liquid-bearing clouds. From the literature, we identify the pattern of the monthly evolution of the MPC fraction as being similar to that of the aerosols in coastal regions, which is*

*related to marine biological activity. Marine bioaerosols are known to be efficient Ice Nucleating Particles (INPs). The emission of these INPs into the atmosphere from open waters would add to the temperature and sea ice fraction seasonalities as factors explaining the MPC fraction monthly evolution."*

Introduction:

L 21 "Down to the Antarctic Seas" - please rephrase

We opted for deleting this bit and write instead at the beginning of the sentence:

P2 L4:

*"In the Southern Ocean (SO) and Antarctic seas…"*

L 24: "dilemma can be solved" (instead of will)

Done. (P2 L7)

page 2:

L 7: remove -> removes

Done. (P2 L14)

L 12: "... track the formation of SLW and the mixed-phase clouds Antarctic-wide" I suggest to add a reference to Lawson and Gettelman 2014 (already in the reference list) who showed the importance of the Antarctic SLW on radiative fluxes both in obser- vations and models, and rephrase the sentence above to emphasize more the existing efforts "to track the formation of SLW and the mixed-phase clouds" from ground-based observations, eg the papers already in the reference list (Lawson and Gettelman 2014, Van Tricht et al 2014, Gorodetskaya et al 2015, Silber et al 2018) and others.

We changed the sentence by adding the part in bold:

P2 L19-22

*"Hence, being able to track the formation of SLW and the mixed-phase Antarctic-wide **and adding to the efforts of ground-based observation studies (Lawson and Gettelman, 2014; Scott and Lubin, 2014; Van Tricht et al., 2014; Gorodetskaya et al., 2015; Silber et al., 2018)** appear as a necessary step to improve cloud microphysics modeling in the Antarctic…"*

Note however, that it appears to us that Lawson and Gettelman 2014, who indeed show the importance of SLW on the Antarctic radiative budget, do show it only in the CESM not through observations. They do investigate - observationally – SLW, but not the radiative budget and the CRE (model only). However, Scott and Lubin 2014 (10.1002/2013JD021132, now added to the reference list from which it was initially missing) very interestingly show the impact of ice fraction in MPC on their radiative impact observed in the NIR at the surface and we acknowledge this by adding their reference into the list.

L 16: "cloud science" -> "observations"

Done. (P2 L25)

L 17: I suggest deleting "As a matter of fact"

P2 L26:

Done. We replaced it by "For instance,…"

page 3: L24: "finally sea ice exerts control over the moisture and heat transported..." - the authors can also mention the importance of sea ice and drifting snow in providing cloud nuclei

We added after this sentence the part in bold (using two references already cited elsewhere in the manuscript, in the discussion):

*P5 L3-8:*

*"Finally, the sea ice exerts control over the moisture and heat transported into the lower atmosphere, and therefore will affect the cloud cover and their properties, as evidenced in the Arctic (Kay and Gettelman, 2009; Taylor et al., 2015; Morrison et al., 2018), and over the Southern Ocean in winter (Wall et al., 2017), and spring and summer (Frey et al., 2018).* ***The sea ice can also impact cloud formation by acting as a source of cloud condensation nuclei (for sea salt coming from blowing snow, see e.g. Yang et al., 2008; Legrand et al., 2016) although this direct link between sea ice and clouds has been much less investigated in the literature so far."***

Section 3.2 Methodology page 10:

L 14: "above the surface" - please specify which surface - ground level or mean sea level, and implications for the defined cloud levels.

We now specify ground level and explain that it seems more appropriate as the large variety of ground levels across our region of interest (from 0m over seas to 4000m over the Plateau) would lead to artificially large variations and depletion of low-level clouds occurrences from the coast to the interior (see also our answer to the other reviewer asking about the reason for not using ISCCP levels). We edited the initial sentence and added the bold part (also motivated by the other reviewer's comment):

*P11 L1-13:*

*"Following Mioche et al. (2015), a distinction is made between low-level clouds (at altitudes between 500 m and 3 km above ground level), mid-level clouds (3 km-6 km above ground level), and high-level clouds (more than 6 km above ground level). When no restriction to a particular altitude level is considered, we will speak about the total cloud fraction or, simply, the cloud fraction.* ***We choose to use ground level and not mean sea level as a reference for altitudes and, similarly, altitude levels rather than pressure levels in order to remain consistent in our description of clouds across the Antarctic region, where ground levels between 0 km and 4 km above mean sea level are found. Using a mean sea level reference or pressure levels to discriminate between clouds of different height would artificially lead to an empty low-level cloud category as looking closer to the pole. Thus we do not make use of the International Satellite Cloud Climatology Project (ISCCP, Rossow and Schiffer, 1999) pressure levels (680 hPa and 440 hPa, which approximately correspond to 3 km and 6 km above mean sea level) as this was done for studies over the SO only. Our goal is to describe the marine and continental clouds of the Antarctic and their differences rather than comparing our observations to the numerous characterisations made over part of, or the whole of the SO using A-Train (and sometimes DARDAR v1) and, or, ISCCP observations (e.g. Hu et al., 2010; Haynes et al., 2011; Huang et al., 2012; Mason et al., 2014; Bodas-Salcedo et al., 2014; Huang et al., 2015)."***

Note the new references added:

Hu et al. 2010: 10.1029/2009jd012384
Haynes et al. 2011 : 10.1175/ 2011jcli4052.1
Huang et al. 2012 : 10.1029/2012jd017800,
Mason et al. 2014 : 10.1175/ jcli-d-14-00139.1,
Huang et al. 2015 : 10.1175/jcli-d-14-00169.1,

3.3 Ceilometers dataset

page 11, L 18: "with a horizontal resolution of 50ft" - I suppose the authors meant vertical range resolution. Please say in meters

Done.

As the authors correctly note, there are problems with Vaisala algorithm's identifica- tion of cloud base heights in Polar Regions, especially for ice clouds (Van Tricht et al 2014). I suggest adding a justifying sentence that in the present manuscript only cloud fractions derived from ceilometers are compared to the DARDAR product (and no dis- tinction is made for CBH for ice or liquid-containing clouds), and for this application Vaisala algorithm should be enough.

Actually section 4.4.3 shows comparison of DARDAR results to micropulse lidar (MPL) measurements made at South Pole. Although coming from distinct regions (90°S for the MPL and 82°S for the satellite), these opportunistic comparisons are fruitful and this is explained in the text. So, strictly speaking it is not only ceilometer measurements that are compared to satellite measurements in our paper. However, we still wanted to present comparisons with ceilometers as these have been the most commun continuously-running instrument in Antarctica so far.

As for the Van Tricht et al. 2014 (VT2014) algorithm it is indeed proven to be more reliable than Vaisala's algorithm since it is built for detecting the thin (precipitating) ice part in the lower part of the mixed-phase layer clouds (with SLW at the top and ice below). The algorithm – at least the way it is presented in VT2014 – does not give access to the phase of the detected CBH (although this could certainly be done by varying the detection threshold- but this is a different work).

Using VT2014's PT algorithm would certainly not change our results and the shortcomings of a satellite/ceilometer' CBH comparison since the impact of using VT2014's algorithm is effectively to lower all the CBH by detecting the signal of the thin precipitating ice below the liquid at the cloud top and since most of the detect CBH (with Vaisala's algorithm) on the two sites (Rothera and Halley) are mostly in the low-level category as explained in the text in 4.4.1

Motived by the reviewer's comment we added a sentence already in the presentation of the dataset in 3.3 to make things clearer:

P13 L11 :

*"VanTricht et al. (2014)'s polar-optimised algorithm effectively lower the cloud base height by allowing for the detection of thin precipitating ice below the supercooled liquid layer at the top of the mixed-phase clouds. Since most of the cloud bases detected by the Vaisala's algorithm are already at low-level (see section 4.4.1), in our particular case of (vertically integrated) low-level cloud cover comparisons between the ceilometers and DARDAR we cannot expect a significant change in using the VanTricht et al. (2014)'s algorithm. However, future work will certainly benefit from using the polar-optimised algorithm for characterising the vertical structure of clouds at these stations and improve the comparison between ceilometers and satellites detections."*

Section 4 Results

page 12, L 13-14: "The deepening of the ASL in winter (Fig 2c) consistently leads to an increase in the CLD fraction over the WAIS (Fig. 5c)." - I suggest replacing "consis- tently leads" with eg "associated with" as the authors do not show the direct causality between ASL and WAIS cloud fraction, and other factors can be playing a role.

We agree with the reviewer's comment and made this change adding two references :

P13 L30:

*"The deepening of the ASL in winter (Figure 2c) is associated with an increase in the cloud fraction over the WAIS (Figure 5c), which is consistent with the intense moisture fluxes and higher cloudiness related to the sustained cyclonic activity across the Amundsen and Ross seas (Nicolas and Bromwich, 2011), a process also observed along East Antarctica's coasts (Dufour et al. 2019)"*

*Nicolas and Bromwich, 2011:* 10.1175/2010jcli3522.1
Dufour et al. 2019 : 10.5194/tc-13-413-2019

My concern is also about the distinction between the "ice-only clouds" and blowing snow. More frequent and more intense storms in winter can be associated with increased blowing snow events - can those be interpreted as "ice clouds" by DARDAR algorithm and contribute to the increase of the "ice clouds" in winter?

Blowing snow could in theory be detected by DARDAR as ice clouds provided it reaches high enough altitudes but…

The average height of blowing snow layers according to Palm et al. (2011,2017) (10.1029/2011JD015828, and 10.5194/tc-11-2555-2017) is a few hundred meters, hence below our 500m threshold used to characterise the cloud seasonality.  From Palm et al. 2017's conclusions: "the average blowing snow layer depth as determined from the CALIOP measurements is 120 m. Layers as high as 200–300 m are not uncommon"

Palm et al. studies use a specific algorithm to detect BS layers (see section 2 of Palm et al. 2011) so that CALIOP alone (at least the way its signal is used within DARDAR) is probably not suited to BS detection and might for instance classify it as "aerosols" (a category we are not investigating with DARDAR here) or just miss it.

In any case, Palm et al. studies suggest that BS should marginally be affecting our statistics and conclusions. If you look at Figure B1c in the appendix you will see that the cloud seasonality is not affected by putting the threshold at 1000 m, altitude above which blowing snow is never detected by Palm et al. The removal of this threshold is actually also only marginally affecting the all-ice statistics (Figure B1c).  Hence, the main features of the all-ice fraction seasonality will not be affected by blowing snow.

It is possible that some part of blowing snow events are missed by Palm et al. since lidar signal may be extinguished when optically thick ice (storm) clouds are present, but then we would expect these deep systems to have cloud base below 3 km above ground i.e. at low-level altitudes, thus not affecting the low-level all-ice fraction in our study.

Studies in a mountainous site (Lloyd et al. 2015 10.5194/acp-15-12953-2015 ; Geert et al. 2015 10.1175/mwr-d-15-0241.1) documented the possible effect of blowing snow on increasings number of cloud ice particles. It appears that this could indeed have significant contribution in situations of cloud enveloped surfaces and in any case near the surface (low-levels) also with the help of surface-induced turbulent processes in the Boundary layer (Geert et al. 2015). Note that Lachlan-Cope et al. 2001 also mentioned the possible effect of BS over the Avery Plateau (Antarctic Peninsula) but only for cloud layers in contact with the surface.

For all the reasons presented above, we do not expect our low-level all-ice fraction to be biased by BS.

We added the paragraph below in the discussion. We added it in 5.1 just after saying that the highest occurrences of BS events are evidenced in the Megadune by Palm et al. (2017) where we actually see the lowest cloud fractions.

*"Additionally, we do not expect blowing snow to bias our all-ice fraction. Palm et al. (2017) showed using a dedicated algorithm based on CALIOP signal that the blowing snow layer depth in Antarctica was on average 120 m and almost always smaller than 500 m with on very rare occasions a depth reaching 1000 m (Palm et al., 2011). DARDAR products are not tuned to detect blowing snow and Palm et al. (2011, 2017) demonstrate the need of a specific algorithm for this purpose. However, and interestingly, Lachlan-Cope et al. (2001) mentioned the possible effect of blowing snow on clouds over the Avery Plateau (Antarctic Peninsula) but only for cloud layers in contact with the surface. Lloyd et al. (2015) documented the effect of blowing snow in increasing the number of cloud ice particles in situations of cloud enveloped surfaces during strong wind events at an alpine site, and Geerts et al. (2015) demonstrated the contribution of surface-induced turbulent processes in the Boundary layer over complex terrains, which is certainly relevant to the Antarctic coastal areas. In any case blowing snow is expected not to affect our statistics because of our 500 m threshold and given the little difference between our low-level all-ice cloud statistics by removing the altitude threshold (Figure B1c). Hence, it is very likely that blowing snow does not bias our low-level cloud fraction statistics even in the (unlikely) case of blowing snow being detected as cloud ice by DARDAR."*

The connection between the deepening of the ASL and increased cloud cover in winter over WAIS has to be supported by literature discussing the increased inland moisture flux associated with increased synoptic activity in the Amundsen Sea region. See eg, Dufour et al 2018 and references therein: Dufour, A., Charrondière, C., and Zolina, O.: Analysed and observed moisture transport as a proxy for snow accumulation in East Antarctica, The Cryosphere Discuss., https://doi.org/10.5194/tc-2018-156, in review, 2018.

The connection between low pressure systems and moisture injection over the WAIS has been investigated e.g. by Nicolas and Bromwhich 2011

As pointed above we added this sentence with a reference related to the WAIS and the Amundsen-Ross low-pressure system.

"*The deepening of the ASL in winter (Figure 2c) is associated with an increase in the cloud fraction over the WAIS (Figure 5c), which is consistent with the intense moisture fluxes and higher cloudiness related to the sustained cyclonic activity across the Amundsen and Ross seas (Nicolas and Bromwich, 2011), a process also observed along East Antarctica's coasts (Dufour et al. 2019)*"

*Nicolas and Bromwich, 2011:* 10.1175/2010jcli3522.1
*Dufour et al. 2019 :* 10.5194/tc-13-413-2019

page 13, L 26: MBL - please write in full Marie Byrd Land (used only once in text and the abbreviation introduced in figure caption) - same for RIS (used here but introduced only on page 19) and AIS.

I suggest to avoid using so many abbreviations and write the geographical region names in full, as there are already many abbreviations used for cloud properties. It becomes difficult to read the text at some point. Also better avoid using abbreviations in section titles.

Done for MBL and PEL (Princess Elisabeth Land). As suggested by reviewer 2 we added a Table to help with the acronyms (see table in the answer related to the abstract). However, we kept RIS and AIS acronyms since both names are often used in some specific paragraphs and it appeared very convenient to keep these acronyms in the text. Also, we removed abbreviations from section titles.

Figure 6 caption: "...distribution of the total all-ice cloud fraction"

Done.

page 15, L 1: The MPC fraction and the USLW cloud fraction..."

We changed the USLW acronym to USLC (and SLW to SLC) as explained in our answer to your comment about the abstract "clearer distinction has to be made between SLW, USLW, and MPC."

The above two comments about using the word "cloud" concerns the entire article: mentioning of the cloud phase shall be accompanied by the word "cloud" - sometimes it is obvious, but in some places can be confusing. This can be included directly in the abbreviations, Eg, MPC already includes "clouds" (mixed-phase clouds), while SLW doesn't = > changing to SLWC or using "SLW cloud" and similarly for other abbrevia- tions concerning cloud phase.

As explained above, we clarified this.

page 15, L 13: isothermes => isotherms

Done (and the other instances too)

Section 4.2 page 15, line 26: "The reduced statistics due to the radar blind zone and lidar signal extinction [delete and] across the Antarctic"

Done. P17 L20

Section 4.3, page 20, L7 (over the whole Antarctic region), L11 (for the Antarctic as a whole) - please specify if this means the entire region of the study (Antarctic ice sheet and Southern Ocean and until which latitude) or only the Antarctic ice sheet. As the average value is rather high (68%) I suppose it is the former (Antarctic ice sheet +SO)

It is indeed the continent + SO. We specify in brackets:

*"(60°S-82°S)"*

We added at the beginning of section 2 in the first paragraph a sentence recalling that "Antarctica" relates to the continent while "Antarctic" refers to Antarctica and its seas/the SO (here poleward of 60°S).

We added the sentence:

P3 L18-19

*"Recall that "Antarctica" refers to the continent while "Antarctic" refers to the whole region including the ocean (60°S-82°S in this study)"*

Figure 9 caption: please provide the abbreviations also in the caption

Done.

Section 4.3.2, page 22, L 26: "The monthly evolution of continental clouds is essentially driven by ice clouds, notably large frontal systems devoid of SLW, as shown in the example in Figure 3b." - I am not sure what the authors mean by "notably large frontal systems devoid of SLW"? Frontal systems associated with extra-tropical cyclones can also bring liquid-containing clouds and the example on Fig 3b shows only one winter case dominated by ice.

We deleted the part of the sentence "notably large frontal systems devoid of SLW, as shown in the example in Figure 3b" as it had nothing to do here, being misleading, while we are mainly describing results here:

P24 L19:

"The monthly evolution of continental clouds is essentially driven by the all-ice clouds."

Section 4.4.2, page 28, L 21-22: ".. the ice phase in the DARDAR-MASK products include both, cloud ice and precipitating ice" - does it also include blowing snow lifted from the ground in the absence of precipitation?

Please refer to our extensive answer above about the blowing snow.

Section 5.1, page 35 L 5: "... the Antarctic-wide SLW fraction decreases 5 from 47 % in summer to 23 % in winter (Figure 9b)"

Indeed. Done.

============================================================================

OTHER MINOR CHANGES/ADDITIONS/REFORMULATIONS (possibly motivated by comments of the other reviewer – if so this is explicitly said.)

============================================================================

A careful reading of the manuscript was operated, which allowed to correct several minor spelling and grammar issues (as pointed out by reviewer 1). Below we report further edits and reformulation of sentences that were made in the manuscript to improve the text and the presentation of the science more generally.

→ P1-L24 to P2-L2:

Reformulation:

*"Clouds' contribution to Antarctica's ice mass balance via precipitation, and to the Antarctic surface energy budget are poorly constrained. However, it has been shown that they exert a warming effect on the ice sheet (Scott et al., 2017; Nicolas et al., 2017) "*

→ P2 L16:

A recent reference added for secondary ice multiplication process modelling in Antarctica. Initially, no reference was given here.

*"… and how processes like secondary ice multiplication, observed in that region (Grosvenor et al., 2012; Lachlan-Cope et al., 2016; O'Shea et al., 2017) can be correctly accounted for (Young et al., 2019). "*

10.1029/2018gl080551

→ P2 L27:

Palerme et al. 2019 added as it is an important addition to the 2014 paper.

*"For instance, Palerme et al. (2014) used satellite radar products to build the first climatology of snowfall rates across the Antarctic continent (updated by Palerme et al. 2019) "*

10. 1109/lgrs.2018.2875007

→ P2 L31

Genthon et al. 2018 added as reference to the published data of their ground-based campaign.

10.5194/essd-10-1605-2018

→ P4 Figure 1

We added the location of Amundsen-Scott South Pole station in order to have all stations used in section 4 (comparison with ground-based measurements) presented on this map.

→ P3 L33 – P5 L1-2

Addition of the bold part:

 *"A cyclonic circulation dominates above the surface with a strong permanent low above the Ross Ice Shelf area, as illustrated **by the 500 hPa geopotential height contour lines plotted in** Figures 2e-h"*

→ P5 L8

We put the sentence in bold before the two other sentences (it was initially put after these sentences).

***"Figure 2i-l show the average seasonal sea ice fraction over 2007-2010 plotted using the passive microwave sea ice concentration data record (Cavalieri and Zwally, 1996, updated yearly) archived by the National Snow and Ice Data 10 Center (NSIDC), and projected onto the grid used to map the cloud fraction (see section 3.2).*** *The largest extent of sea ice occurs in September, and the smallest in February. The westernmost part of the Weddell Sea shows a persistent, and dense, sea ice coverage throughout the year."*

→ P6 L10

We added the reference Cazenave et al. 2018 as it is a recent piece of work updating the work by Delanoe and Hogan 2008 on the DARDAR-CLOUD products:

10.5194/amt-2018-397

→ P8 L12-13

For clarity, "**per gridbox**" was added in the following sentence:

*« The SO limit at 60° S shows one overpass every two days (~45 per season) **per gridbox**, while the southern most limit at 82° S shows on average more than 2.5 overpasses per day (~250 per season) **per gridbox** »*

→ *P8 L 31*

*Subsection title simplified: "Cloud and phase fraction mapping" to "Cloud fraction mapping"*

→ P10 L15-26 .

We reorganised the following paragraph for more clarity – adding also a reference to Korolev et al. 2017 (for the typical structure of boundary layer mixed-phase clouds described here).

*"The fraction of SLW-containing clouds is called the SLC fraction, $F_{SLC}$. Table 1 recalls the acronyms used for the various cloud types (as well as the ones for specific Antarctic regions). The DARDAR-MASK includes a mixed-phase category ("SLW with ice" – first type), and we extend this category by adding the clouds where a pure SLW layer is detected with at least three adjacent vertical pixels containing ice below (second type), following Mioche et al. (2015). In practice, most of the detected mixed-phase clouds are of the first type, but pure SLW layers with an ice phase immediately below are clearly detected. We interpret these as occurrences of a mixed-phase since the ice below is immediately in contact with the liquid layer; their microphysics must be interacting. Note that cloud where ice crystals are too small and/or too few to be detected by the radar in the top SLW layer of the cloud is also possible (recall that in the upper atmosphere, for instance, the CPR cannot detect thin cirrus). A cloud top made out of SLW with ice precipitating below is characteristic of boundary layer mixed-phase clouds (e.g. Korolev et al., 2017) and, in practice, cloud layers flagged by DARDAR as actual mixed-phase (and not pure SLW) come systematically along with ice below. The mixed-phase clouds (first and second types) are described by the MPC fraction, $F_{MPC}$. Supercooled liquid water-containing clouds (SLC) that are not part of any mixed-phase clouds as defined above (hence being pure liquid) are categorised as unglaciated supercooled liquid clouds (USLC), whose fraction is $F_{USLC}$"*

→ P15 L25-29

Parts in bold added to the sentence to be more explicit:

*"It can be already noticed that the spatial pattern of the sea ice fraction spatial distribution in winter is not similar to the one of the all-ice cloud fraction distribution, **contrary to what is observed for the SLC fraction** (for instance, compare the winter patterns of sea ice in Figure 2o with the winter cloud **and SLC** distributions in Figure 5c and **g, on one side**, and the winter all-ice distribution in Figure 6c, on the other side)."*

→ P15 L34-P16 L1

Parts in bold added to the sentence to be more explicit:

*"As for the SLC fraction, the MPC and USLC fractions are lower on the continent and particularly more in EA, where they decrease polewards. Interestingly, these fractions show no significant differences between each other **on the continent, in contrast to what is observed over seas. This will be investigated and discussed further below** "*

→ P20 L9-11

New formulation for the sentence below, for clarity:

"Hence, higher cloud occurrences occur at higher altitudes in winter, and this is consistent with the deeper ASL and the contraction of the westerly circulation towards the coast (Figure 4c)"

→ P21 L13

Reformulation:
*"In EA, the presence of SLC is evidenced in summer (Figure 7Bc) while no SLC is detected over the Plateau in winter, except where the depression of the land south of the AIS is (Figure 7Bi). There, poleward intrusion of moisture and cloudiness from the coastal areas would cause enhanced SLC fractions (Figure 7Ai)."*

→ P29 L29: Following a remark by the other reviewer about SO clouds observed in Australia we added this.

*"Another study comparing the DARDAR v2 dataset with ground-based measurements of SO clouds at Cape Grim, Australia (Alexander and Protat, 2018) used a space-averaging technique based on typical wind speeds for the DARDAR observations. However, the high occurrence of ceilometer*

*signals obscuration (Figure 12b) makes it unlikely that using more sophisticated averaging techniques will improve our comparisons at both stations"*

10.1002/ 2017jd026552

→ P30 L2

Addition of the part in bold to recall the brief discussion on the polar-optimsed algorithm by VT2014 added in 3.3 already (see answer to reviewer's comment about section 3.3)

*"We have shown qualitative agreements between DARDAR and ceilometer observations at both stations. **The use of a polar-optimised algorithm for ceilometer observations (VanTricht et al., 2014) for further cloud vertical distributions comparisons is needed but it would not have affected our conclusions in the present study, as explained in section 3.3.** More generally, there is a need for more systematic comparisons of ground-based measurements of cloud occurrences with combinations of lidars and cloud profiling radars in Antarctica. This will be the topic of a future study using the DARDAR products for more recent years, when such ground instruments were 35 deployed. "*

→ P36 Figure 16

Caption extended as some explanations were missing.

*"Whisker plots showing the potential temperature difference $\theta_{850hPa}$-$\theta_{SLP}$, as a function of the sea ice fraction for each season. The potential temperatures are derived from the ECMWF temperatures and pressure profiles collocated with the satellite overpasses, at the DARDAR footprint level. **The Spearman's rank correlation coefficient between $\theta_{850hPa}$-$\theta_{SLP}$ and the sea ice fraction is derived over the whole sample that is used to then compute the whisker plots, and it is indicated at the top right corner of each subplot. The p-values for the correlation coefficients are always <0.01 except for summer (p-value=0.03). The absolute difference (in K) of near-surface static stability between open water (sea ice fraction <5 %) and sea ice (sea ice fraction >95 %) is indicated below each correlation coefficient. "***

*Also note Figure 16c and d were (marginally) updated with slightly different amplitude differences between open water and sea ice for the potential temperature difference (1K difference only)*

→ P39 L34

Reformulation:

*"This difference between MPC and USLC is observed for all the Antarctic seas taken together (Figure 17a). On the continent, however, both seasonal cycles of the MPC and USLC fractions have the same pattern (Figure 17f). The further away from the coast, the more marked the maximum of the MPC fraction is by the end of summer and beginning of autumn (compare ARS and WSS with WS and RS)"*

→ P40 L7

Reformulation. We toned down and reformulate to the following sentence.

"Overall, these observations point to a possible role of the sea ice in modulating the seasonality of glaciation processes leading to MPC"

→ P40 – Figure 17:

We flipped the sea ice scale as suggested by the other reviewer, in order to ease the interpretation. We notify this in the figure caption and in the main text. We changed the colours of the lines.

For clarity and in order to better present the arguments the following paragraph has been reformulated:

*"The near-surface conditions (atmospheric temperature and water vapour mass mixing ratio) are similar below MPC and USLC over seas (Figure 11c and d). Consistently, the near surface static stability below either MPC or USLC is the same (not shown). This suggests that it is not a stronger coupling of the sea surface with the atmosphere at particular places that overall drives the difference between MPC and USLC seasonalities, notably by the adiabatic cooling caused by enhanced upward motions where the coupling is stronger. Hence, differences in surface atmospheric states cannot explain the stronger correlations of the MPC fraction with the sea ice fraction, compared to the USLC fraction. The average altitude of MPC is higher than the one of USLC, and the former are logically associated to colder temperatures than the latter (hence more glaciation processes are favoured). However, it is not the temperature seasonal cycle alone that is responsible for the differences between the MPC and the USLC seasonal cycles, since the temperatures measured at the top of both cloud types show a similar and simpler seasonal evolution (Figure 11a). Importantly, Figure B1 (used for testing the sensitivity of our results to the choice of the lower altitude threshold) shows that above 1000m asl, where MPC dominate compared to USLC (Figure 8), the seasonal cycle of USLC still differs from the one of MPC in a similar way than it does with the 500 m asl threshold. Finally, the difference between the MPC and the USLC seasonal cycles cannot be an artefact caused by the radar signal loss since the radar clutter occurrences do not show any seasonality (Figure A2a). "*

Addition of two sentences. Possible reason to help understand the time-lag between peak of primary biological production at the sea surface, and peak in MPC fraction, backing up our hypothesis:

*"The time-lag between the month of the maximum primary production (January, on average) and the month with the largest MPC fractions (February, on average) could be related to the life cycle of phytoplankton blooms, the demise of which eventually triggers the release of organic material (e.g. exudates) in about three weeks to a month (O'Dowd et al., 2015). This would then come along with the surface area of emission of biological INP via sea spray (i.e. open water) being the largest in February"*

10.1038/srep14883

We thank the reviewer #2 for the interesting comments that lead us to detail some aspects of the method – e.g. related to the use of the ceilometers or the choice of altitude levels and altitude reference (e.g. different from the ISCCP levels)

We first answer the reviewer's questions or comments and then list some changes made by us in the text to improve clarity (sometimes motivated by the other reviewer's comments).

Response to reviewer #2

Please note that we edited the abstract in order to be more quantitative and not only qualitative. We also added a sentence pointing to the interesting comparisons between ground-based observations at South Pole and our observation at 82°S (on the Plateau).

Edited abstract:
*"Antarctic tropospheric clouds are investigated using the DARDAR (raDAR/liDAR)-MASK products between 60°S and 82°S. The cloud fraction (occurrence frequency) is divided into the supercooled liquid water-containing clouds (SLC) fraction and its complementary part called the all-ice cloud fraction. A further distinction is made between SLC involving ice (mixed-phase clouds, MPC) or not (USLC, for unglaciated SLC). The low-level (< 3 km above surface level) SLC fraction is larger over seas (20-60%), where it varies according to sea ice fraction, than over continental regions (0-35%). The total SLC fraction is much larger over West Antarctica (10-40 %) than it is over the Antarctic Plateau (0-10 %). In East Antarctica the total SLC fraction - in summer for instance - decreases sharply polewards with increasing surface height (decreasing temperatures) from 40% on the coast to <5% at 82°S on the Plateau. The geographical distribution of the continental total all-ice fraction is shaped by the interaction of the main low-pressure systems surrounding the continent and the orography, with little association with the sea ice fraction. Opportunistic comparisons with published ground-based supercooled liquid water observations at South Pole in 2009 are made with our SLC fractions at 82°S in terms of seasonal variability, showing good agreement. We demonstrate that the largest impact of sea ice on the low-level SLC fraction (and mostly through the MPC) occurs in autumn and winter (22% and 18% absolute decrease of the fraction between open water and sea ice-covered regions, respectively), while it is almost null in summer and intermediate in spring (11%). Monthly variability of the MPC fraction over seas shows a maximum at the end of summer and a minimum in winter. Conversely, the USLC fraction has a maximum at the beginning of summer. However, monthly evolutions of MPC and USLC fractions do not differ on the continent. This suggests a seasonality in the glaciation process in marine liquid-bearing clouds. From the literature, we identify the pattern of the monthly evolution of the MPC fraction as being similar to that of the aerosols in coastal regions, which is related to marine biological activity. Marine bioaerosols are known to be efficient Ice Nucleating Particles (INPs). The emission of these INPs into the atmosphere from open waters would add to the temperature and sea ice fraction seasonalities as factors explaining the MPC fraction monthly evolution."*

Minor Comments (references given at the end) :

1) There are so many acronyms in the text. I suggest a table in an Appendix listing them all to ease the reader's need to refer back or memorise them.

Based also on the comments by reviewer #1 – and for homogeneity - we changed:
SLW to SLC for Supercooled Liquid water-containing Clouds.
USLW to USLC for Unglaciated… Clouds.
(We kept MPC.) We removed the CLD designation as it is confusing, since it is not an acronym. We just use "cloud" instead.

Note that:
Wsec was changed to WSS for Weddell Sea sector
ARsec was changed to ARS for Amundsen-Ross sector
PLA was changed to AP for Antarctic Plateau
(in the text AP referred to Antarctic Peninsula and this was corrected for.)
The above changes are meant to use proper acronyms only.

We added a Table at the end of section 3.2.2 (not in the appendix) as requested to facilitate the reading of the manuscript.

P 10 L12-13: "Table 1 recalls the acronyms used for the various cloud types (as well as the ones for specific Antarctic regions) "

**Table 1.** Acronyms used in the text to designate some cloud phase or cloud types and some Antarctic places or areas.

| | |
|---|---|
| SLW | supercooled liquid water |
| SLC* | SLW-containing cloud |
| MPC* | mixed-phase cloud |
| USLC* | unglaciated SLW cloud |
| WA | West Antarctica |
| EA | East Antarctica |
| SO | Southern Ocean |
| WSS** | Weddell Sea sector |
| ARS** | Amundsen-Ross sector |
| WS** | Weddell Sea |
| RS** | Ross Sea |
| WAIS** | West Antarctic Ice Sheet |
| AP** | Antarctic Plateau |
| AIS | Amery Ice Shelf |
| RIS | Ross Ice Shelf |

(*) Their fractions are linked by equation 2.

(**) Geographical areas defined in Figure 4c, and whose names relate to the regions in which they are located.

2) Figure 1: A gap in the contour (black and white) topography is evident south of 82S. No doubt this is due to the fact that CALIOP / CloudSat don't see south of here. However, you really ought to fill in the contour levels between here and the Pole.

We updated Figure 1.

3) Page 10, lines 14-16. The authors define low-level clouds as those between 0.5- 3km above the surface; mid-level as 3-6km; and high-level clouds >6km. This strikes me as somewhat arbitrary, especially since no rationale for these altitude cutoffs is given. Do you have a convincing argument for choosing fixed levels? While in the tropics a fixed altitude cutoff may be appropriate (such as 4.5km for the fairly-constant freezing level at low latitudes, e.g. Protat et al., 2014, JAMC), in the extra-tropics it is better to work on pressure levels. I suggest the authors change these limits to match those from the ISCCP cloud low/mid/high definitions (Low cloud top pressure >680hPa; middle cloud top pressure>440 hPa etc.). This would facilitate more direct comparisons with previous studies, especially over the Southern Ocean, where pressure levels are regularly used (for example Haynes et al., JClim, 2011; Mason et al., JClim, 2014).

We see your point. The explanations below hopefully demonstrate why we think it makes sense to stick to our initial choice of using altitude levels.

We are investigating a region where the ground level altitude changes from 0 m above mean sea level to >3500 m above mean sea level, in the 60°S-82°S region. Using pressure levels as a reference would artificially reduce the number of low-level clouds from the coastal regions to the interior of the continent in relation to the surface pressure decrease with altitude (for instance at Dome C – approx. 3200m altitude – the surface pressure is < 660 hPa – the ISCCP low-level cloud category would be empty). We ought to keep a consistent framework to describe low-level clouds, especially that for our distinction between MPC and USLC (former USLW) we focus on low-level clouds to show that both categories display a different seasonality over marine areas but not over land (Figure 17).

Regarding any comparison to ISCCP, recall that our method does not discriminate between cloud levels based on cloud top altitudes. We do not detect cloud tops here or say we only do it for the liquid-containing clouds (to get the cloud top temperature in Figure 11). So, even by changing our altitude levels to the ISCCP pressure levels our method would still show discrepancies since our way to use DARDAR-MASK does not involve an automated retrieval and classification according to cloud tops. Note that the 680 hPa and 440 hPa levels actually correspond to the ~3 km and ~6 km altitudes (above sea level), but, again, we are not discriminating cloud levels using cloud tops, rather the altitude of the cloud phase pixels themselves (as this was done by Mioche et al. 2011 for the Arctic using DARDAR v1).

As a final note, Figure 10 d, e, f demonstrate that liquid-bearing clouds are largely driven by low-level clouds, which largely dominate below 3 km (Figure 7Ba, d, g and j) and even below 2 km. This for saying that changing the limit from 3 km to 2 km would not significantly affect our results.

As we think it might indeed be useful for the reader to have this emphasised we added two sentences in 3.2.2 (along with references recommended by the reviewer, and others):

*P11 L4-13:*

*"We choose to use ground level and not mean sea level as a reference for altitudes and, similarly, altitude levels rather than pressure levels in order to remain consistent in our description of clouds across the Antarctic region, where ground levels between 0 km and 4 km above mean sea level are found. Using a mean sea level reference or pressure levels to discriminate between clouds of different height would artificially lead to an empty low-level cloud category as looking closer to the pole. Thus we do not make use of the International Satellite Cloud Climatology Project (ISCCP Rossow and Schiffer, 1999) pressure levels (680 hPa and 440 hPa, which approximately correspond to 3 km and 6 km above mean sea level) as this was done for studies over the SO only. Our goal is to describe the marine and continental clouds of the Antarctic and their differences rather than comparing our observations to the numerous characterisations made over part of, or the whole of the SO using A-Train (and sometimes DARDAR v1) and, or, ISCCP observations (e.g. Hu et al., 2010; Haynes et al., 2011; Huang et al., 2012; Mason et al., 2014; Bodas-Salcedo et al., 2014; Huang et al., 2015)."*

Note the new references added:

Hu et al. 2010: 10.1029/2009jd012384
Haynes et al. 2011 : 10.1175/ 2011jcli4052.1
Huang et al. 2012 : 10.1029/2012jd017800,
Mason et al. 2014 : 10.1175/ jcli-d-14-00139.1,
Huang et al. 2015 : 10.1175/jcli-d-14-00169.1,

4) Regarding comparisons between surface-based instruments and satellite, this is, as the authors note, a challenge given different temporal / spatial sampling, and indeed, DARDAR curtains likely do not pass directly above the surface sites anyway. One additional option to make closer comparisons would be to spatially average DARDAR and temporally average the ceilometers. For example I found that in a recent Southern Ocean DARDAR/surface lidar study, DARDAR data were horizontally averaged based on typical wind speeds at cloud height (Alexander & Protat, JGR, 2018). I wonder whether applying something similar in Section 4.4.1 above Rothera & Halley would be of benefit, despite the simplicity of this averaging?

In section 4.4.1 we horizontally averaged the DARDAR data to increase the statistics, in addition to considering the gridbox of the station only. As explained in the text, for Rothera we average over the upwind area (Bellingshausen sea) and for Halley, over the Weddell Sea. The resulting time-series have expectedly a "less noisy" interannual variability but overall resemble the gridbox-only time-series (Figure 12a) (similar to the differences seen between DARDAR-250 and DARDAR-1000 in the reference you cite). Ceilometer observations are averaged over a month and, as we say it in the text, using only the times closer to satellite overpasses does not significantly affect the time-series. From figure 12b we can see the high occurrences of obscuration of the ceilometers signals (presumably due to fog or blowing snow), which may help explain a large part of the discrepancies between satellite and ceilometer. In their study using a UV lidar at 355 nm Alexander & Protat, JGR, 2018 do use an averaging technique based on typical wind speeds but – given the issues with obscurations when measuring CBH – we cannot expect significant improvement of our comparisons using this technique alone.

In the end, to contrast our investigation with a different context for SO clouds we added this sentence in 4.4.1 where we present the comparisons ceilometers/DARDAR, using the reference you cite:

P29 L29:

*"Another study comparing the DARDAR v2 dataset with ground-based measurements of SO clouds at Cape Grim, Australia (Alexander and Protat, 2018) used a space-averaging technique based on typical wind speeds for the DARDAR observations. However, the high occurrence of ceilometer*

*signals obscuration (Figure 12b) makes it unlikely that using more sophisticated averaging techniques will improve our comparisons at both stations"*

5) The authors note that ceilometers detect cloud base heights (page 28, line 5), specifically the Vaisala CBH (Section 3.3). It is known that these are not accurate CBHs in the polar regions, especially for ice (e.g. van Tricht et al., AMT, 2014). Some comment ought to be made about this additional source of uncertainty comparing ceilometer and DARDAR in Section 4.4.1 in the context of your minimum altitude cutoffs.

We had a similar remark from the other reviewer and we report the relevant part of our answer here:

The Van Tricht et al. 2014 (VT2014) algorithm it is indeed proven to be more reliable than Vaisala's algorithm since it is built for detecting the thin (precipitating) ice part in the lower part of the mixed-phase layer clouds (with SLW at the top). Using VT2014's PT algorithm would very likely not change our results and the shortcomings of our satellite/ceilometers' CBH comparison since the impact of using VT2014's algorithm is effectively to lower the detected CBH by capturing the signal of the thin precipitating ice below the liquid at the cloud top, and since most of our detected CBH (with the Vaisala's algorithm) on the two sites (Rothera and Halley) are mostly in the low-level category as explained in the text in 4.4.1.

These three sentences were added at the end of section 3.3:

P13 L11 :

*"VanTricht et al. (2014)'s polar-optimised algorithm effectively lower the cloud base height by allowing for the detection of thin precipitating ice below the supercooled liquid layer at the top of the mixed-phase clouds. Since most of the cloud bases detected by the Vaisala's algorithm are already at low-level (see section 4.4.1), in our particular case of (vertically integrated) low-level cloud cover comparisons between the ceilometers and DARDAR we cannot expect a significant change in using the VanTricht et al. (2014)'s algorithm. However, future work will certainly benefit from using the polar-optimised algorithm for characterising the vertical structure of clouds at these stations and improve the comparison between ceilometers and satellites detections."*

6) Figure 17: I think that it would be much clearer to interpret if you flipped the sea-ice scale

We flipped the scale and changed the colour-code and this does seem to help the interpretation and we specify this in the text and in the figure caption. (We changed the colours of the lines.)

7) Finally, I suggest a careful, thorough re-read of the manuscript to correct several minor spelling and grammar issues.

We corrected many minor spelling and grammar issue.

====================================================================

OTHER MINOR CHANGES/ADDITIONS/REFORMULATIONS (possibly motivated by comments of the other reviewer – if so this is explicitly said.)

====================================================================

A careful reading of the manuscript was operated. Below we report further edits and reformulation of sentences that were made in the manuscript to improve the text and the presentation of the science more generally.

→ P1-L24 to P2-L2:

Reformulation:

*"Clouds' contribution to Antarctica's ice mass balance via precipitation, and to the Antarctic surface energy budget are poorly constrained. However, it has been shown that they exert a warming effect on the ice sheet (Scott et al., 2017; Nicolas et al., 2017)* "

→ P2 L16:

A recent reference added for secondary ice multiplication process modelling in Antarctica. Initially, no reference was given here.

*"… and how processes like secondary ice multiplication, observed in that region (Grosvenor et al., 2012; Lachlan-Cope et al., 2016; O'Shea et al., 2017) can be correctly accounted for **(Young et al., 2019).** "*

10.1029/2018gl080551

→ P2 L27:

Palerme et al. 2019 added as it is an important addition to the 2014 paper.

*"For instance, Palerme et al. (2014) used satellite radar products to build the first climatology of snowfall rates across the Antarctic continent (**updated by Palerme et al. 2019**) "*

10. 1109/lgrs.2018.2875007

→ P2 L31

Genthon et al. 2018 added as reference to the published data of their ground-based campaign.

10.5194/essd-10-1605-2018

→ P4 Figure 1

We added the location of Amundsen-Scott South Pole station in order to have all stations used in section 4 (comparison with ground-based measurements) presented on this map.

→ P3 L18-19

*"Recall that "Antarctica" refers to the continent while "Antarctic" refers to the whole region including the ocean (60°S-82°S in this study)"*

→ P3 L33 – P5 L1-2

Addition of the bold part:

*"A cyclonic circulation dominates above the surface with a strong permanent low above the Ross Ice Shelf area, as illustrated **by the 500 hPa geopotential height contour lines plotted in** Figures 2e-h"*

→ P5 L8

We put the sentence in bold before the two other sentences (it was initially put after these sentences).

***"Figure 2i-l show the average seasonal sea ice fraction over 2007-2010 plotted using the passive microwave sea ice concentration data record (Cavalieri and Zwally, 1996, updated yearly) archived by the National Snow and Ice Data 10 Center (NSIDC), and projected onto the grid used to map the cloud fraction (see section 3.2).*** *The largest extent of sea ice occurs in September, and the smallest in February. The westernmost part of the Weddell Sea shows a persistent, and dense, sea ice coverage throughout the year."*

→ P6 L10

We added the reference Cazenave et al. 2018 as it is a recent piece of work updating the work by Delanoe and Hogan 2008 on the DARDAR-CLOUD products:

10.5194/amt-2018-397

→ P8 L12-13

For clarity, "**per gridbox**" was added in the following sentence:

*« The SO limit at 60° S shows one overpass every two days (~45 per season) **per gridbox**, while the southern most limit at 82° S shows on average more than 2.5 overpasses per day (~250 per season) **per gridbox** »*

*→ P8 L 31*

*Subsection title simplified: "Cloud and phase fraction mapping" to "Cloud fraction mapping"*

→ P10 L15-26 .

We reorganised the following paragraph for more clarity – adding also a reference to Korolev et al. 2017 (for the typical structure of boundary layer mixed-phase clouds described here).

*"The fraction of SLW-containing clouds is called the SLC fraction, $F_{SLC}$. Table 1 recalls the acronyms used for the various cloud types (as well as the ones for specific Antarctic regions). The DARDAR-MASK includes a mixed-phase category ("SLW with ice" – first type), and we extend this category by adding the clouds where a pure SLW layer is detected with at least three adjacent vertical pixels containing ice below (second type), following Mioche et al. (2015). In practice, most of the detected mixed-phase clouds are of the first type, but pure SLW layers with an ice phase immediately below are clearly detected. We interpret these as occurrences of a mixed-phase since the ice below is immediately in contact with the liquid layer; their microphysics must be interacting. Note that cloud where ice crystals are too small and/or too few to be detected by the radar in the top SLW layer of the cloud is also possible (recall that in the upper atmosphere, for instance, the CPR cannot detect thin cirrus). A cloud top made out of SLW with ice precipitating below is characteristic of boundary layer mixed-phase clouds (e.g. Korolev et al., 2017) and, in practice, cloud layers flagged by DARDAR as actual mixed-phase (and not pure SLW) come systematically along with ice below. The mixed-phase clouds (first and second types) are described by the MPC fraction, $F_{MPC}$. Supercooled liquid water-containing clouds (SLC) that are not part of any mixed-phase clouds as defined above (hence being pure liquid) are categorised as unglaciated supercooled liquid clouds (USLC), whose fraction is $F_{USLC}$"*

→ P13 L30:

Justification to the link between a more intense ASL and a higher cloud fraction (as asked by the other reviewer).

*"The deepening of the ASL in winter (Figure 2c) is associated with an increase in the cloud fraction over the WAIS (Figure 5c), which is consistent with the intense moisture fluxes and higher cloudiness related to the sustained cyclonic activity across the Amundsen and Ross seas (Nicolas and Bromwich, 2011), a process also observed along East Antarctica's coasts (Dufour et al. 2019)"*

*Nicolas and Bromwich, 2011:* 10.1175/2010jcli3522.1
Dufour et al. 2019 : 10.5194/tc-13-413-2019

→ P15 L25-29

Parts in bold added to the sentence to be more explicit:

*"It can be already noticed that the spatial pattern of the sea ice fraction spatial distribution in winter is not similar to the one of the all-ice cloud fraction distribution, **contrary to what is observed for the SLC fraction** (for instance, compare the winter patterns of sea ice in Figure 2o with the winter cloud and **SLC** distributions in Figure 5c and **g, on one side**, and the winter all-ice distribution in Figure 6c, on the other side)."*

→ P15 L34-P16 L1

Parts in bold added to the sentence to be more explicit:

*"As for the SLC fraction, the MPC and USLC fractions are lower on the continent and particularly more in EA, where they decrease polewards. Interestingly, these fractions show no significant differences between each other **on the continent, in contrast to what is observed over seas. This will be investigated and discussed further below** "*

→ P20 L9-11

New formulation for the sentence below, for clarity:

"Hence, higher cloud occurrences occur at higher altitudes in winter, and this is consistent with the deeper ASL and the contraction of the westerly circulation towards the coast (Figure 4c)"

→ P21 L13

Reformulation:
*"In EA, the presence of SLC is evidenced in summer (Figure 7Bc) while no SLC is detected over the Plateau in winter, except where the depression of the land south of the AIS is (Figure 7Bi). There, poleward intrusion of moisture and cloudiness from the coastal areas would cause enhanced SLC fractions (Figure 7Ai)."*

→ P30 L2

Addition of the part in bold to recall the brief discussion on the polar-optimsed algorithm by VT2014 added in 3.3 already (see answer to reviewer's comment about section 3.3)

*"We have shown qualitative agreements between DARDAR and ceilometer observations at both stations. **The use of a polar-optimised algorithm for ceilometer observations (VanTricht et al., 2014) for further cloud vertical distributions comparisons is needed but it would not have affected our conclusions in the present study, as explained in section 3.3.** More generally, there is a need for more systematic comparisons of ground-based measurements of cloud occurrences with combinations of lidars and cloud profiling radars in Antarctica. This will be the topic of a future study using the DARDAR products for more recent years, when such ground instruments were 35 deployed."*

→ P36 Figure 16

Caption extended as some explanations were missing.

*"Whisker plots showing the potential temperature difference $\theta_{850hPa}$-$\theta_{SLP}$, as a function of the sea ice fraction for each season. The potential temperatures are derived from the ECMWF temperatures and pressure profiles collocated with the satellite overpasses, at the DARDAR footprint level. **The Spearman's rank correlation coefficient between $\theta_{850hPa}$-$\theta_{SLP}$ and the sea ice fraction is derived over the whole sample that is used to then compute the whisker plots, and it is indicated at the top right corner of each subplot. The p-values for the correlation coefficients are always <0.01 except for summer (p-value=0.03). The absolute difference (in K) of near-surface static stability between open water (sea ice fraction <5 %) and sea ice (sea ice fraction >95 %) is indicated below each correlation coefficient.** "*

*Also note Figure 16c and d were (marginally) updated with slightly different amplitude differences between open water and sea ice for the potential temperature difference (1K difference only)*

→ P37 L17-28:

Motivated by the other reviewer's comments, we added the paragraph below about blwoing snow in the discussion. We added it in 5.1 just after saying that the highest occurrences of BS events are evidenced in the Megadune by Palm et al. (2017) where we actually see the lowest cloud fractions.

*"Additionally, we do not expect blowing snow to bias our all-ice fraction. Palm et al. (2017) showed using a dedicated algorithm based on CALIOP signal that the blowing snow layer depth in Antarctica was on average 120 m and almost always smaller than 500 m with on very rare occasions a depth reaching 1000 m (Palm et al., 2011). DARDAR products are not tuned to detect blowing snow and Palm et al. (2011, 2017) demonstrate the need of a specific algorithm for this purpose. However, and interestingly, Lachlan-Cope et al. (2001) mentioned the possible effect of blowing snow on clouds over the Avery Plateau (Antarctic Peninsula) but only for cloud layers in contact with the surface. Lloyd et al. (2015) documented the effect of blowing snow in increasing the number of cloud ice particles in situations of cloud enveloped surfaces during strong wind events at an alpine site, and Geerts et al. (2015) demonstrated the contribution of surface-induced turbulent processes in the Boundary layer over complex terrains, which is certainly relevant to the Antarctic coastal areas. In any case blowing snow is expected not to affect our statistics because of our 500 m threshold and given the little difference between our low-level all-ice cloud statistics by removing the altitude threshold (Figure B1c). Hence, it is very likely that blowing snow does not bias our low-level cloud fraction statistics even in the (unlikely) case of blowing snow being detected as cloud ice by DARDAR."*

→ P39 L34

Reformulation:

*"This difference between MPC and USLC is observed for all the Antarctic seas taken together (Figure 17a). On the continent, however, both seasonal cycles of the MPC and USLC fractions have the same pattern (Figure 17f). The further away from the coast, the more marked the maximum of the MPC fraction is by the end of summer and beginning of autumn (compare ARS and WSS with WS and RS)"*

→ P40 L7

Reformulation. We toned down and reformulate to the following sentence.

*"Overall, these observations point to a possible role of the sea ice in modulating the seasonality of glaciation processes leading to MPC"*

→ P41 L1-14

For clarity and in order to better present the arguments the following paragraph has been reformulated:

*"The near-surface conditions (atmospheric temperature and water vapour mass mixing ratio) are similar below MPC and USLC over seas (Figure 11c and d). Consistently, the near surface static stability below either MPC or USLC is the same (not shown). This suggests that it is not a stronger coupling of the sea surface with the atmosphere at particular places that overall drives the difference between MPC and USLC seasonalities, notably by the adiabatic cooling caused by enhanced upward motions where the coupling is stronger. Hence, differences in surface atmospheric states cannot explain the stronger correlations of the MPC fraction with the sea ice fraction, compared to the USLC fraction. The average altitude of MPC is higher than the one of USLC, and the former are logically associated to colder temperatures than the latter (hence more glaciation processes are favoured). However, it is not the temperature seasonal cycle alone that is responsible for the differences between the MPC and the USLC seasonal cycles, since the temperatures measured at the top of both cloud*

*types show a similar and simpler seasonal evolution (Figure 11a). Importantly, Figure B1 (used for testing the sensitivity of our results to the choice of the lower altitude threshold) shows that above 1000m asl, where MPC dominate compared to USLC (Figure 8), the seasonal cycle of USLC still differs from the one of MPC in a similar way than it does with the 500 m asl threshold. Finally, the difference between the MPC and the USLC seasonal cycles cannot be an artefact caused by the radar signal loss since the radar clutter occurrences do not show any seasonality (Figure A2a). "*

Addition of two sentences. Possible reason to help understand the time-lag between peak of primary biological production at the sea surface, and peak in MPC fraction, backing up our hypothesis:

[revised manuscript text omitted]